# A fluorogenic probe for predicting treatment response in non-small cell lung cancer with *EGFR*-activating mutations

Hui Deng [1,2,3,5] ✉, Qian Lei[1,2,3,5], Chengdi Wang[1,5], Zhoufeng Wang [1,3], Hai Chen[2,3], Gang Wang[2], Na Yang[2], Dan Huang[4], Quanwei Yu[2], Mengling Yao[3], Xue Xiao[3], Guonian Zhu[3], Cheng Cheng[3], Yangqian Li[3], Feng Li [4], Panwen Tian[1] ✉ & Weimin Li [1,2,3] ✉

Therapeutic responses of non-small cell lung cancer (NSCLC) to epidermal growth factor receptor (EGFR) - tyrosine kinase inhibitors (TKIs) are known to be associated with *EGFR* mutations. However, a proportion of NSCLCs carrying *EGFR* mutations still progress on EGFR-TKI underlining the imperfect correlation. Structure-function-based approaches have recently been reported to perform better in retrospectively predicting patient outcomes following EGFR-TKI treatment than exon-based method. Here, we develop a multicolor fluorescence-activated cell sorting (FACS) with an EGFR-TKI-based fluorogenic probe (HX103) to profile active-EGFR in tumors. HX103-based FACS shows an overall agreement with gene mutations of 82.6%, sensitivity of 81.8% and specificity of 83.3% for discriminating *EGFR*-activating mutations from wild-type in surgical specimens from NSCLC patients. We then translate HX103 to the clinical studies for prediction of EGFR-TKI sensitivity. When integrating computed tomography imaging with HX103-based FACS, we find a high correlation between EGFR-TKI therapy response and probe labeling. These studies demonstrate HX103-based FACS provides a high predictive performance for response to EGFR-TKI, suggesting the potential utility of an EGFR-TKI-based probe in precision medicine trials to stratify NSCLC patients for EGFR-TKI treatment.

Lung cancer is considered one of the leading causes of cancer-related deaths worldwide, accounting for 1.79 million deaths (18.4% of total cancer deaths) in 2020[1,2]. Histologically, about 85% of diagnosed lung cancers are non-small cell lung cancer (NSCLC) with a 5-year survival of less than 15%[3]. Most NSCLC are considered locally advanced or metastatic at the time of diagnosis, and the patients with epidermal growth factor receptor (*EGFR*) mutations account for a clinically significant proportion[4]. The most common NSCLC-associated *EGFR* mutations are the point mutation replacing leucine with arginine in exon 21 (L858R) and in-frame deletions in exon 19 (19del), which account for ~85% of all *EGFR* mutations and are associated generally with having adenocarcinoma histology[5,6]. These common mutations, also known as *EGFR*-

[1]Department of Respiratory and Critical Care Medicine, West China Hospital, Sichuan University, Chengdu, Sichuan, China. [2]Targeted Tracer Research and Development Laboratory, Precision Medicine Key Laboratory of Sichuan Province, Precision Medicine Research Center, West China Hospital, Sichuan University, Chengdu, Sichuan, China. [3]Institute of Respiratory Health, Frontiers Science Center for Disease-related Molecular Network, West China Hospital, Sichuan University, Chengdu, Sichuan, China. [4]Key Laboratory of Green Chemistry & Technology of Ministry of Education, College of Chemistry, Sichuan University, Chengdu, Sichuan, China. [5]These authors contributed equally: Hui Deng, Qian Lei, Chengdi Wang. ✉e-mail: huideng0923@hotmail.com; mrascend@163.com; weimi003@scu.edu.cn

activating mutations, can be detected in approximately 10–15% of the US/European population and up to 50% of Asian patients with NSCLC[7–10], occurring at a high frequency in Asian patients who never smoked or were only light smokers[4,11].

Over the last decades, EGFR-tyrosine kinase inhibitors (EGFR-TKIs) such as gefitinib[12] and erlotinib[13] were discovered and produced significant clinical benefits in patients with advanced NSCLC[14–17]. *EGFR*-activating mutations appear to correlate with sensitivity to EGFR-TKIs (e.g., gefitinib and erlotinib)[18,19]. The standard method to detect *EGFR* mutations is based on polymerase chain reaction (PCR) amplified genomic DNA[20–22]. Clinical studies have shown that approximately 70–80% (depending on the trials) of *EGFR* mutation-positive NSCLC patients respond to EGFR-TKI treatment[19,23–32], and a small proportion (~20–30%) of patients carrying *EGFR*-activating mutations still do not show objective response when treated with EGFR-TKI. This observation suggests that there are probably alternative mechanisms conferring EGFR mutant (in)activation. As such, *EGFR* gene mutations may not the sole determinant of EGFR-TKI response. In addition, DNA-based approaches have an inherent limit of detection as well as poor reproducibility. Furthermore, high cost and technically complex also make these methods often inaccessible for widespread application, outside of commercial laboratories and some research centers.

Other predictive beneficial biomarkers have also been proposed for EGFR-TKI treatment, including EGFR expression measured by immunohistochemistry (IHC) with mutation-specific antibodies (L858R and 19del)[33–37], and *EGFR* copy number assessed by fluorescent in situ hybridization (FISH)[38–41]. However, they were not validated as useful biomarkers for EGFR-TKI treatment. Most recently, ref. 42 reported that a structure-function-based approach for defining functional groups of *EGFR* mutations performed better in retrospectively predicting patient outcomes following EGFR-TKI treatment than traditional exon-based groups. Accordingly, there is an urgent need to develop techniques for the prediction of EGFR-TKI sensitivity by comprehensive functional EGFR profiling, and hence for the improvement of targeted therapy efficacy.

EGFR-TKIs are tailored drugs that specifically target the ATP site of the EGFR kinase domain and prevent EGFR phosphorylation. Thus, an EGFR-TKI-based probe that mimics the action of EGFR-TKI can "on-target" detect EGFR with functional activity. Fluorescent probes have recently been extensively studied and widely applied for protein-specific detection[43–47]. In contrast to other methods (e.g., gene sequencing and radioactive tracers), fluorescent probes have unique advantages, such as low cost, high sensitivity, easy operation, fast response, and high spatial and temporal resolution. Despite a variety of fluorescent probes have been developed, the one with a turn-on switch mechanism is of particular interest in terms of high signal-to-background ratios, efficiency, and sensitivity. Among them, environment-sensitive fluorophores such as 4-sulfonamidebenzoxadiazole (SBD) have unique emission properties that are highly sensitive to the immediate environment[48–50]. Generally, very weak fluorescence is observed with these fluorophores in the polar and protic environment, whereas fluorescence turn-on can be activated when they are in hydrophobic surroundings[51]. As most of the binding sites in proteins are hydrophobic, fluorescent turn-on probes thus can be achieved by incorporating an environment-sensitive fluorophore with a protein-specific ligand (e.g., small-molecule inhibitors like EGFR-TKIs). Therefore, the development of such fluorogenic probes targeting functional EGFR may offer new convenient tools to guide clinical choices for patients with *EGFR*-mutant NSCLC, both clinically and economically.

In this work, we report a fluorogenic probe (HX103) and establish a proof-of-concept multicolor fluorescence-activated cell sorting (FACS) assay with HX103 to fully quantify active-EGFR in tumors from NSCLC patients. Specifically, we (i) design and synthesize a fluorogenic probe on the basis of known EGFR-TKIs (e.g., gefitinib); (ii) evaluate the binding specificity of HX103 in the presence of human recombinant EGFR, as well as in living cells harboring distinct *EGFR* mutations; (iii) develop a probe-based multicolor FACS assay to fully quantify active-EGFR; (iv) analyze a total of 46 surgical specimens from NSCLC patients, including 23 tumor tissues and 23 the adjacent normal tissues, to evaluate the correlation between *EGFR*-activating mutations and HX103-based FACS analysis; and (v) translate HX103 to a clinical study and analyze 31 biopsy samples from the enrolled NSCLC patients to identify the individuals who may benefit from EGFR-TKI therapy. Our results show that EGFR-TKI-based probes can detect functional EGFR, holding promise as biomarkers for identifying patients sensitive to EGFR-TKIs in precision medicine trials.

## Results

### Design and synthesis of an EGFR-TKI-based fluorogenic probe

To prove the feasibility of using fluorescent probes for stratifying NSCLC patients for EGFR-TKI therapy, we set out to rational design EGFR-TKI-based probes (Fig. 1a). In general, small-molecule fluorescent probes are composed of three distinct features: (i) a pharmacophore (recognition element that selectively binds to the intended target), (ii) a fluorophore, and (iii) a linker. In view of the reported EGFR-TKI structures, we choose gefitinib as the starting point to develop fluorescent probes. Gefitinib, a 4-anilinoquinazoline derivative, is a selective EGFR inhibitor approved by FDA for locally advanced or metastatic NSCLC therapy[12]. As numerous derivatives have been synthesized based on the scaffold of gefitinib, we thus conducted a careful analysis of the available structure-activity relationship (SAR) data to identify a suitable site for attachment of a fluorophore to the 4-anilinoquinazoline scaffold without disrupting the potency[52]. We found the C-7/C-8 positions of the quinazoline, pointing outside the ATP binding pocket toward solvent, were well modified to include different elements for inhibitor optimization and probe development. Furthermore, inspired by the "off-on" property of environment-sensitive fluorophores (e.g., SBD), we envision that novel fluorogenic probes targeting active-EGFR can be achieved by incorporating environment-sensitive fluorophores on the pharmacophore of EGFR-TKI. It is expected that the binding of EGFR-TKI to the hydrophobic domain of EGFR would bring the environment-sensitive fluorophore closer to the hydrophobic pocket, thereby causing the fluorophore to emit stronger fluorescence. By contrast, in the absence of a target protein, the EGFR-TKI-based probe would remain in the aqueous buffer and show very weak fluorescence. To the end, we developed HX103 by introducing SBD to 4-anilinoquinazoline scaffold at the C-8 position via a suitable linker (Fig. 1b).

To understand its binding mode, we performed a molecular docking study of HX103 with EGFR wild-type, L858R, and L858R/T790M double-mutant. Of note, as the crystal structure of EGFR 19del has not yet been published, we thus haven't performed the docking study of HX103 with EGFR 19del. After extracting EGFR-TKI from the inhibitor-protein complex, HX103 was flexibly docked into the active site, adopting a similar conformation to gefitinib in the binding site (Fig. 1c and Supplementary Fig. 1). The key interactions between the quinazoline ring of HX103 and the surrounding residues remained as the same as gefitinib, including Met 793 and Lys 745 in EGFR L858R. Although the interaction of Asp800 with 6-propylmorphiolino (from gefitinib) was absent in the binding of HX103 with EGFR L858R, an additional lone pair-π interaction between Phe795 and the SBD group of HX103 was observed (Fig. 1c and Supplementary Fig. 1). As for EGFR wild-type and L858R/T790M double-mutant, orientations of the SBD group of HX103 were different from that of EGFR L858R. As shown in Fig. 1c and Supplementary Fig. 1, only one H bond interaction was found between HX103 and Met 793 in the EGFR wild-type, while gefitinib formed another ionic bonding with Asp800 in EGFR wild-type. Despite an additional H bond was found between HX103 and Pro794 in EGFR double-mutant, the interaction was weaker when compared with

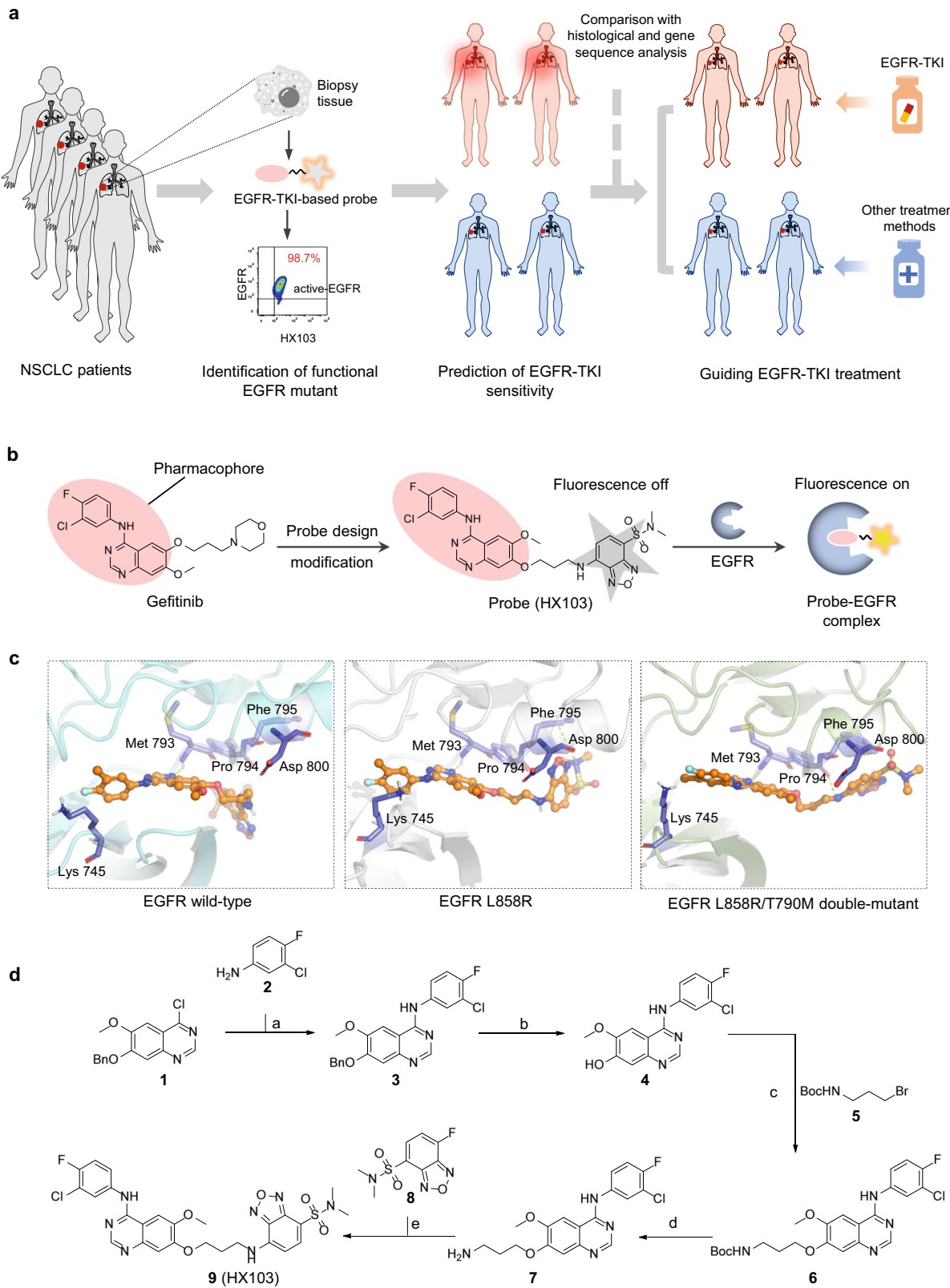

the interaction between gefitinib and Asp800 in double-mutant (the ionic bonding and H bond interaction were both observed, Supplementary Fig. 1). Furthermore, binding energies of the fluorescent probe HX103 and gefitinib to EGFR wild-type and mutants were calculated (Supplementary Table 1). These results provide the theoretical basis for the binding affinity of HX103 toward EGFR wild-type and the mutants.

The synthesis of HX103 (**9**) commenced with the nucleophilic substitution of the chloride in quinazoline derivative **1** by the commercially available 3-chloro-4-fluoroaniline (**2**), followed by removing benzyl group to afford alcohol intermediate **4**, which was further converted into compounds **6** via nucleophilic substitution of building block **5**. Probe precursor **7** was subsequently obtained via N-Boc deprotection in the presence of TFA. Finally, the fluorogenic probe (**9**,

**Fig. 1 | Development of an EGFR-TKI-based fluorogenic probe to quantify active-EGFR for stratification of NSCLC patients. a** Schematic of EGFR-TKI explaining the benefits to NSCLC patients through a stratification strategy using the combination of a fluorogenic probe and DNA sequencing analysis to guide treatment and clinical trial choices for patients with *EGFR*-mutant NSCLC. **b** Structure-based design of a fluorogenic probe by incorporating the environment-sensitive fluorophore (SBD) into the pharmacophore of EGFR-TKI (e.g., gefitinib) and illustration of the probe turn-on property: the probe shows low fluorescence in the absence of EGFR, whereas it can exhibit stronger fluorescence by binding to the kinase domain of EGFR. **c** Predicted binding modes of HX103 with EGFR wild-type (PDB: 4WKQ), EGFR L858R mutant (PBD: 4LQM), and EGFR L858R/T790M (PDB: 5EDP). **d** Synthesis of HX103 (**9**), reagents and conditions: (a) 3-chloro-4-fluoroaniline (**2**), i-PrOH, DCM, reflux, 4 h, 91% (**3**); (b) TFA, 70 °C, ammonium hydroxide, 96% (**4**); (c) tert-butyl (3-bromopropyl)carbamate (**5**), $K_2CO_3$, DMF, 80 °C, 4 h, 78% (**6**); (d) TFA, r.t., 1 h, 70% (**7**); (e) 7-fluoro-*N,N*-dimethylbenzo[c][1,2,5] oxadiazole-4-sulfonamide (**8**), acetonitrile, $Et_3N$, r.t., 1 h, 23% (**9**).

HX103) was furnished by nucleophilic substitution with the SBD derivative 7-fluoro-N,N-dimethylbenzo[c][1,2,5]oxadiazole-4-sulfonamide (**8**) (Fig. 1d).

## Spectroscopic properties and fluorescence turn-on response on EGFR

With the probe in hand, we set out to determine the fluorescent properties, including UV absorption, fluorescence spectra, and quantum yield. The probe presented maximum UV absorption at ~340 and ~440 nm, and the fluorescence emission wavelength was in green-range of 570 nm with excitation wavelength at 440 nm (Stokes shift, 130 nm) (Supplementary Table 1 and Supplementary Fig. 2). The absolute quantum yield of HX103 was 5.8% in PBS (pH 7.2), whereas that in DMSO was significantly increased to 42.4% (Supplementary Table 2). To further investigate the environment-sensitive property, we recorded the fluorescence spectra of HX103 in various solvents with different polarity to mimic the hydrophobic environment of the protein pocket. In the end, HX103 gave remarkable fluorescence enhancement in acetonitrile in contrast to the aqueous solution (PBS or $H_2O$) (Supplementary Fig. 2). In other organic solvents such as DMSO and ethanol, we also observed increased fluorescence intensity, revealing that HX103 possesses environment-sensitive properties with turn-on mechanism.

Next, we decided to evaluate whether HX103 exhibits environment-sensitive properties while binding with the kinase domain of human EGFR. As shown in Fig. 2, HX103 was non-fluorescent in PBS, but exhibited high fluorescence upon the addition of wild-type or mutant EGFR (L858R and 19del). Furthermore, we applied HX103 to the "gatekeeper" point mutation T790M, the most common resistance mutation[53,54], and a relatively low fluorescence was found. These results indicated that HX103 is selective toward EGFR wild-type and primary mutants (L858R and 19del), but less sensitive to the acquired resistance mutation EGFR T790M (Fig. 2c). The enhanced fluorescence was quenched by preincubation of EGFR-TKI (e.g., gefitinib, afatinib) (Fig. 2a, b), confirming the binding specificity of the probe on EGFR-tyrosine kinase. Furthermore, HX103 had a slightly stronger binding affinity to EGFR L858R ($K_d = 0.8 \pm 0.3\,\mu M$) and EGFR 19del ($K_d = 1.1 \pm 0.2\,\mu M$), when compared with EGFR wild-type ($K_d = 2.7 \pm 0.4\,\mu M$) and the acquired resistance mutation T790M ($K_d = 6.6 \pm 4.6\,\mu M$) (Supplementary Fig. 2). These results demonstrate the high fluorescence turn-on response of HX103 on EGFR wild-type and primary mutants (L858R and 19del).

## Molecular pharmacology of HX103 on EGFR

We next set out to determine the inhibitory activity of HX103 to recombinant human EGFR kinase using a fluorescent mobility-shift assay. As shown in Fig. 2e and Supplementary Table 2, HX103 showed a concentration-dependent inhibition to EGFR wild-type with an $IC_{50}$ of 4.0 nM (3.6–4.4 nM; 95% CI, $n = 3$), similarly to gefitinib ($IC_{50} = 1.9$ nM) (Supplementary Fig. 4 and Supplementary Table 3), suggesting a minimal effect by introducing the fluorophore to the pharmacophore of EGFR-TKI. As for EGFR L858R and EGFR 19del, HX103 inhibited the kinase activities with $IC_{50}$ values of 1.5 nM (1.3–1.9 nM; 95% CI, $n = 3$) and 1.3 nM (1.2–1.5 nM; 95% CI, $n = 3$), respectively (Fig. 2e and Supplementary Table 3). As expected, the probe's inhibition toward the resistance mutation T790M was remarkably decreased with an $IC_{50}$ of

977 nM (832–1147 nM; 95% CI, $n = 3$), ~650-fold and ~750-fold decrease when compared to the primary mutants L858R and 19del, respectively (Supplementary Table 3). These results demonstrate that HX103 preserves high binding affinity toward the kinase domain of EGFR wild-type and the primary mutants (L858R and 19del).

Since the *EGFR*-activating mutations reside near the ATP cleft targeted by EGFR-TKI, we thus assessed whether our probe has altered the effects on EGFR activation and the downstream signaling pathways, including PI3K-AKT-mTOR and RAS-RAF-MEK-ERK in several cancer cells. In this study, EGFR activation was quantified by measuring the phosphorylation of the tyrosine residue (Y1068), a marker commonly known for EGFR autophosphorylation[55]. In the absence of serum and associated growth factors, minimal autophosphorylation of EGFR was observed in cells expressing EGFR wild-type (A549, Hela, and A431) (Fig. 2i–k), while cells harboring *EGFR*-activating mutations (HCC827 and H1975) showed obvious EGFR autophosphorylation (Fig. 2g, h). The addition of a ligand (e.g., EGF) significantly induced phosphorylation of EGFR in wild-type cells and doubled the activation of EGFR in mutant cells (HCC827 and H1975). To evaluate the effects of HX103 on EGFR activation and the downstream signaling networks, we determined the protein levels of EGFR, pEGFR, ERK1/2, pERK1/2, AKT, and pAKT in these cells. Remarkably, HX103 inhibited the phosphorylation of EGFR and the downstream proteins (without obviously affecting their total proteins' levels) in HCC827 cells (EGFR 19del) (Fig. 2f, g). In H1975 cells (L858R/T790M), a certain resistance of HX103 was observed (Fig. 2h and Supplementary Fig. 3). Further studies revealed that wild-type EGFR phosphorylation (e.g., A549, A431, and Hela cells) was also significantly inhibited by HX103 (Fig. 2i–k), which is similar to the effects of gefitinib (Supplementary Fig. 4). As an ideal fluorescent probe to detect and visualize biotargets, low cytotoxicity is important. Thus, the cytotoxicity of HX103 in these cells was evaluated and the minimal effect was found on cell growth under the experimental conditions in most cells (Supplementary Table 4).

Taken together, these results demonstrate that HX103 targets the active site of EGFR-tyrosine kinase and inhibits EGFR activation by competing with ATP, thereby holding promise as a useful chemical tool to detect active-EGFR and providing functional information on the tyrosine kinase of EGFR.

## Targeting active-EGFR by HX103 in cancer cells with distinct EGFR mutation status

To evaluate the targeting specificity of HX103 in living cells, a panel of cancer cells (H1975, HCC827, A549, A431, Hela, and MCF-7) were selected and treated with HX103 (5 μM, 30 min, 37 °C). Fluorescence confocal imaging studies showed abundant binding of HX103 to HCC827, A431, and H1975, significantly less binding to A549 and Hela, and neglectable binding to EGFR-negative cells (MCF-7) (Fig. 3a and Supplementary Fig. 5). The binding specificity of HX103 in these cells was evaluated by preincubation with gefitinib. HCC827, H1975, A431, and Hela cells pretreated with gefitinib (50 μM) showed ~50% lower uptake of HX103 than untreated cells (Fig. 3b). We thus applied HX103 to determine the subcellular distributions of EGFR in living cells. Counterstaining experiments for HX103-treated H1975 cells were performed with markers of various subcellular organelles (e.g., ER Tracker, Mito-Tracker, and LysoSensor) (Supplementary Fig. 6). The results showed that HX103-labeled cellular EGFR was internalized and

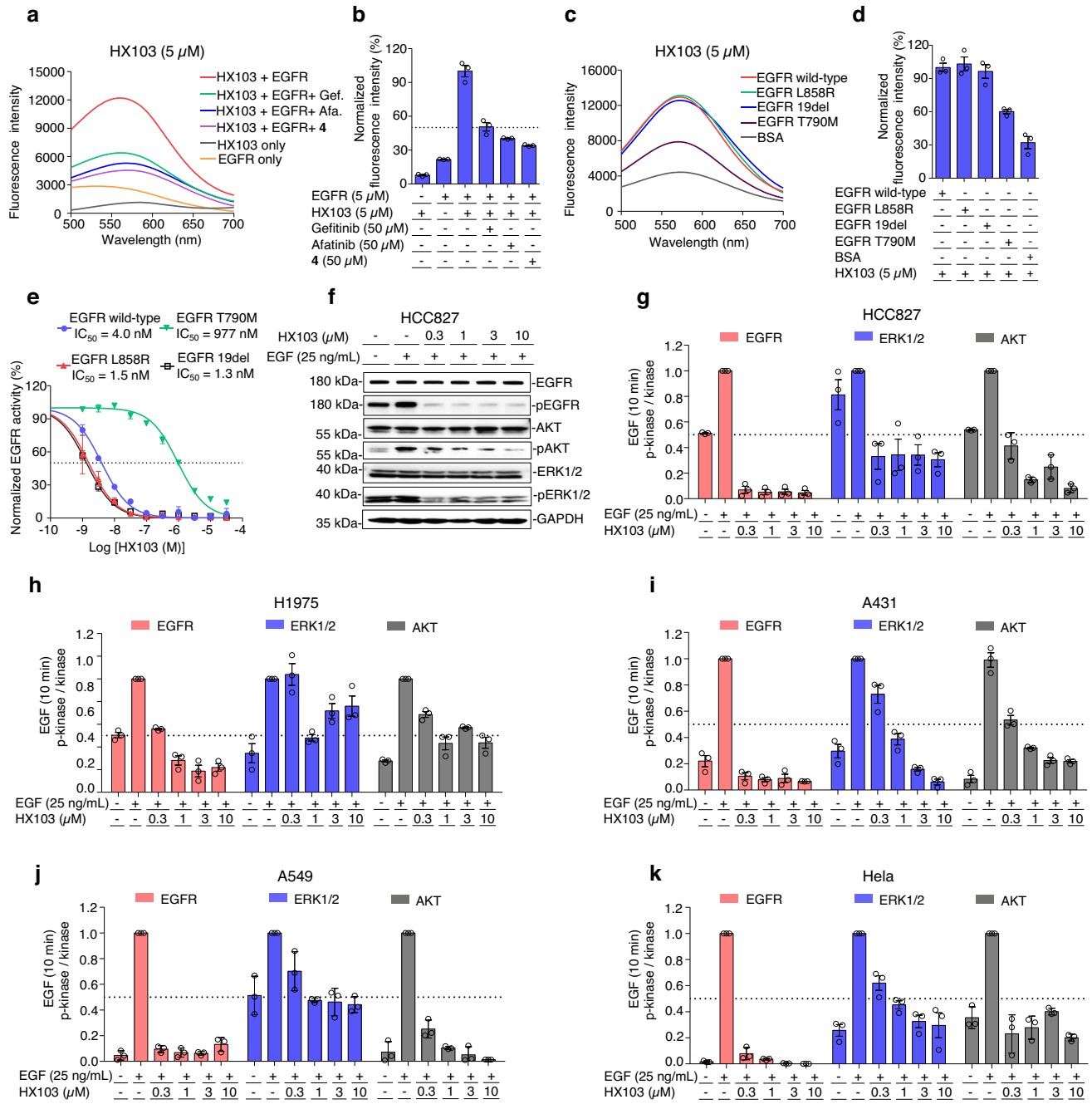

**Fig. 2 | Fluorescence turn-on response and molecular pharmacology of HX103 on EGFR. a** Fluorescence spectra of HX103 (5 μM) in the absence and presence of EGFR (5 μM) in PBS buffer (0.1% DMSO) and after the addition of EGFR-TKI (50 μM, e.g., gefitinib, afatinib, and compound **4**). **b** Quantification of the fluorescence intensity at 570 nm is shown in (**a**). Data represent average values ± SEM, n = 3 independent experiments per group. The excitation wavelength for emission fluorescence spectra was 440 nm. **c** Fluorescence spectra of HX103 (5 μM) in the presence of distinct forms of EGFR (wild-type, L858R, 19del, and T790M) in PBS buffer (0.1% DMSO). **d** Quantification of the fluorescence intensity at 570 nm is shown in (**c**). Data represent average values ± SEM, n = 3 independent experiments

per group. The excitation wavelength for emission fluorescence spectra was 440 nm. **e** Kinase inhibitory activity of HX103 on recombinant human EGFR wild-type, EGFR L858R, EGFR 19del, and EGFR T790M determined by a fluorescent mobility-shift assay. Data represent average values ± SEM, n = 3 independent experiments per group. **f** Western blots were used to evaluate the impacts of HX103 (0 to 10 μM) on EGFR activation with EGF (25 ng/mL) and its downstream AKT and ERK1/2 phosphorylation levels in HCC827. Uncropped blots are in Source Data. **g–k** Quantification of protein bands shown in (**F** and Supplementary Fig. 3). Data represent average values ± SEM, n = 3 independent experiments per group. Source data are provided as a Source Data file.

translocated in the endoplasmic reticulum in living cells, which is in line with previous reports[56,57].

Next, to precisely quantify the accumulation of HX103 in living cells, we turned to FACS analysis in cancer cells with different forms of EGFR. In the presence of HX103, high fractions of H1975 and HCC827 cells demonstrated remarkably increased fluorescence, followed by

A431 and PC-9, whereas a lower amount of probe labeling was detected in A549 and Hela cells, and a significantly less amount was observed in EGFR-negative cells (e.g., MCF-7 and Jurkat) (Fig. 3c). Preincubation with gefitinib showed significant reductions in mean fluorescence intensity (MFI), thereby indicating that EGFR was successfully targeted by HX103 (Supplementary Fig. 7). To evaluate the correlation between

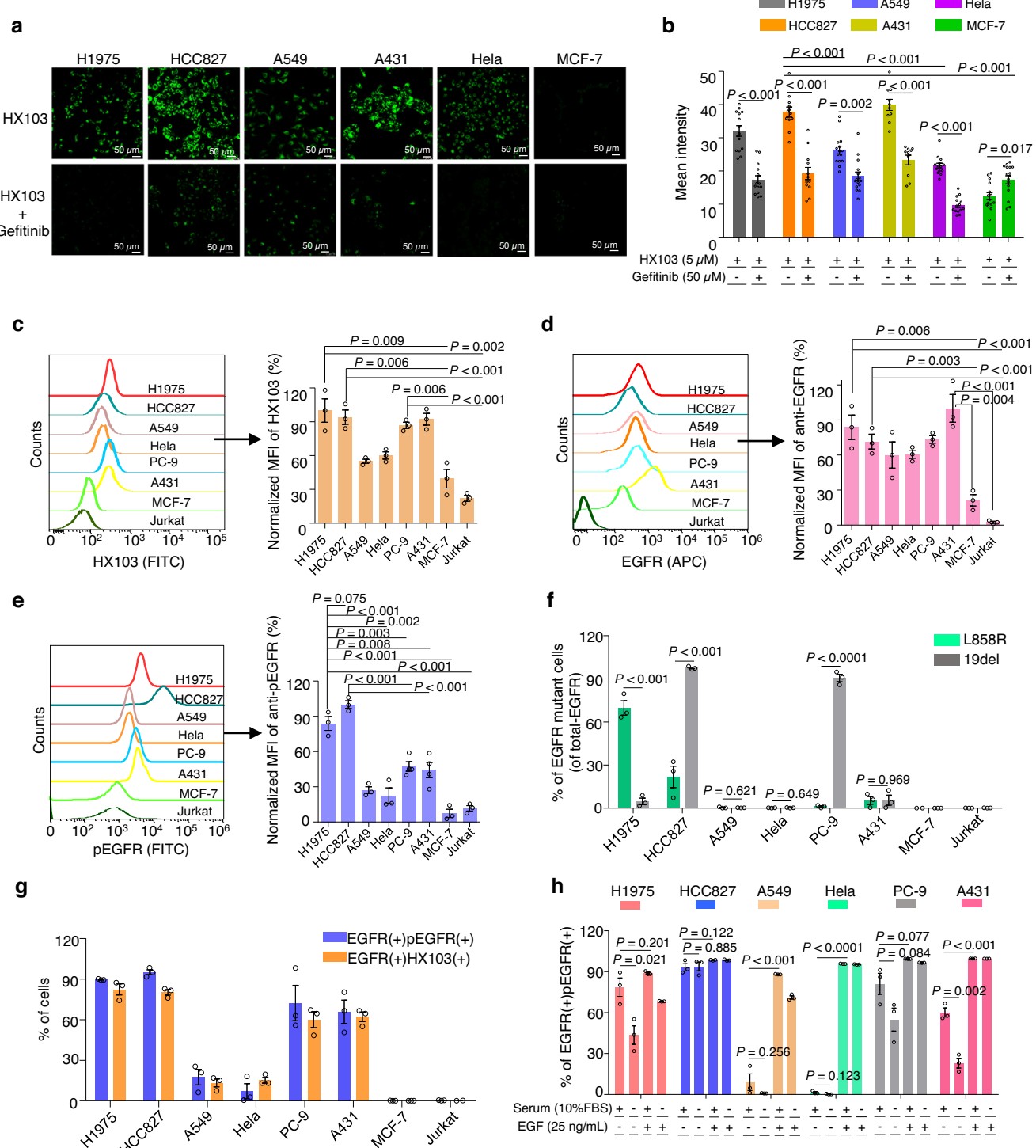

**Fig. 3 | Targeting active-EGFR by HX103 in various cancer cells. a** Visualization of HX103 (5 μM, 30 min) uptake in cells by confocal microscopy. Scale bar, 50 μm. The uptake of HX103 in a panel of cells can be blocked by preincubation with gefitinib (50 μM). **b** Quantification of the probe's mean fluorescence intensity in cell images from (**a**). Data represent average values ± SEM. $n = 14$ for H1975 and Hela (3 biological replicates × 4–5 pictures); $n = 12$ for HCC827 (3 biological replicates × 4 pictures); $n = 15$ for A549 and MCF-7 cells (3 biological replicates × 5 pictures); $n = 9$ for A431 (3 biological replicates × 3-4 pictures). The statistical $P$ values were calculated by the two-tailed Student's test. **c–e** Flow cytometric experiments were performed to quantify the labeling of HX103 (**c**), EGFR antibody (**d**), and pEGFR antibody (**e**) in cancer cells. Bar chart revealing normalized MFI in cells. For HX103 (5 μM, 30 min) and pEGFR, the MFI of HCC827 was set at 100% to normalize the MFI of the remaining cells. For EGFR, the MFI of A431 was set at 100% to normalize the

remaining cells. Data represent average values ± SEM, $n = 3$ independent experiments; note that $n = 4$ independent experiments for PC-9 and A431 in (**e**). The statistical $P$ values were calculated by the two-tailed Student's test. **f** Bar chart showing the percentages of total EGFR that is mutant in cancer cells (Supplementary Fig. 14). Data represent average values ± SEM, $n = 3$ independent experiments. The statistical $P$ values were calculated by the two-tailed Student's test. **g** Bar chart showing percentage values of EGFR(+)pEGFR(+) or EGFR(+)HX103(+) in cells (Supplementary Figs. 9–11). Data represent average values ± SEM, $n = 3$ independent experiments. **h** Bar chart showing the percentage of EGFR(+)pEGFR(+) in the presence and absence of EGF or 10% FBS (Supplementary Figs. 16, 17). Data represent average values ± SEM, $n = 3$ independent experiments. The statistical $P$ values were calculated by the two-tailed Student's test. Source data are provided as a Source Data file.

probe labeling and the actual EGFR expression, we further performed FACS analysis with the antibody of total EGFR in these cells (Fig. 3d). Consistent with the results of probe labeling, low levels of EGFR expression were observed in MCF-7 and Jurkat cells. Interestingly, we found A431 exhibited the highest level of EGFR expression among these cells, followed by H1975 and HCC827. The discrepancy between the labeling of HX103 and EGFR antibody in A431 might be due to the fact that HX103 only detects active-EGFR rather than the amount of total-EGFR expression. A431 is previously reported as a carcinoma cell line with an unusually high number of EGFR[58], and we thus validated the expression of EGFR in A431 cells by quantitative polymerase chain reaction (qPCR). Indeed, an overproduction of *EGFR* mRNA was observed in A431 when compared to other EGFR wild-type cells (e.g., A549 and Hela) (Supplementary Fig. 8). In a normal cellular system, EGFR is known to be autoinhibited (that is only activated on the binding of the cognate ligand such as EGF), indicating a small number of active-EGFR can be detected by HX103 in normal EGFR wild-type cells. To validate this, we quantified EGFR activation levels with the antibody specifically recognizing the phosphorylated form of EGFR. In line with HX103 labeling, the highest level of EGFR autophosphorylation was observed in HCC827 and H1975 cells, followed by PC-9 and A431 (Fig. 3e). The results reveal that the levels of EGFR activation show a similar pattern as the accumulation of HX103. Furthermore, we found the percentage of EGFR antibody and HX103 double-positive cells [EGFR(+)HX103(+)] showed similar percentage values as that of EGFR(+)pEGFR(+) (Fig. 3g and Supplementary Figs. 9–12), again suggesting that HX103 labeling may correlate with EGFR activation. We subsequently substantiated this by using Pearson's correlation analysis [HX103(+) v.s. pEGFR(+), $r = 0.9835$; $P < 0.0001$; EGFR(+)HX103(+) vs EGFR(+)pEGFR(+), $r = 0.9928$; $P < 0.0001$] (Supplementary Fig. 13). Altogether, these studies support the hypothesis that HX103 preferably targets active-EGFR, measuring the levels of EGFR kinase activity instead of protein expression.

With the EGFR mutant-specific antibodies, ~66% of EGFR L858R expression was detected in H1975, while ~97 and ~86% of EGFR 19del expressions were found in HCC827 and PC-9, respectively (Fig. 3f and Supplementary Fig. 15). *EGFR*-activating mutations have been postulated to cause structural alterations that destabilize the autoinhibited confirmation of EGFR in the absence of ligand binding, supporting the results that high level of EGFR autophosphorylation was detected in H1975, HCC827, and PC-9. Of note, the percent of EGFR activation in HCC827 (~90%) is higher than that in PC-9 (~60%) (both cell lines harboring 19del). This might be due to *EGFR* gene amplification in HCC827[34] (an overproduction of *EGFR* mRNA was observed in Supplementary Fig. 8). A431 was demonstrated without EGFR-activating mutations (Fig. 3f and Supplementary Figs. 14, 15), but showing around 60% of EGFR autophosphorylation (Fig. 3g and Supplementary Figs. 10, 12). This can be explained by EGFR overexpression that also induces the activation of EGFR-pathway[59–61]. It gives a clue that higher labeling of HX103 was observed in A431, when compared to A549 and Hela. Importantly, in the absence of serum, we found EGFR autophosphorylation was obviously reduced in A431 (Fig. 3h and Supplementary Fig. 17), whereas serum-free culture conditions showed no significant reduction in EGFR autophosphorylation of HCC827 and PC-9 cells. Notably, in the absence of serum, approximately 50% of EGFR autophosphorylation was reduced in H1975. These results suggest that the culture conditions also play a role in EGFR activation in cancer cells. Furthermore, the addition of EGF significantly increased EGFR phosphorylation in A549, Hela, and A431 cells (Supplementary Fig. 17), but had no significant effects on EGFR mutant cells (e.g., HCC827, PC-9, and H1975) (Fig. 3h and Supplementary Fig. 16), implicating high baseline of EGFR activation in cancer cells carrying EGFR-activating mutations. Besides, no obvious signal of either EGFR autophosphorylation or HX103 labeling was detected in EGFR-negative cells (MCF-7 and Jurkat) (Fig. 3g). Finally, the binding affinities of HX103 in these cell lines were

**Table 1 | Comparison of EGFR mutation detection in surgical specimens (tumors) from NSCLC patients using different methods**

| Sample code | Sex | Stage (AJCC)[a] | Sanger | EGFR(+)HX103(+) (%)[b] | EGFR(+) (%)[c] | HX103(+) (%)[d] |
|---|---|---|---|---|---|---|
| #1 | M | I | 19del | + (37.7) | + (72.7) | + (39.7) |
| #2 | M | I | – | + (62.2) | + (98.9) | + (58.0) |
| #3 | M | II | 19del | – (22.2) | + (93.9) | – (21.9) |
| #4 | F | II | L858R | + (56.2) | + (89.9) | + (56.5) |
| #5 | M | I | 19del | + (40.6) | + (85.2) | + (39.0) |
| #6 | F | I | 19del | + (60.9) | + (90.5) | + (61.8) |
| #7 | F | I | L858R | – (1.69) | – (11.2) | – (2.40) |
| #8 | M | I | – | – (22.6) | – (21.6) | + (90.6) |
| #9 | F | II | –[e] | + (44.4) | + (59.3) | + (54.6) |
| #10 | F | II | – | – (5.35) | – (29.3) | – (13.4) |
| #11 | M | II | – | – (2.39) | – (28.3) | – (4.48) |
| #12 | F | I | L858R | + (98.7) | + (98.9) | + (99.6) |
| #13 | M | III | L858R | + (80.3) | + (98.2) | + (65.7) |
| #14 | F | IV | – | – (0.02) | – (14.7) | – (0.50) |
| #15 | F | I | – | – (0.03) | – (0.26) | – (9.84) |
| #16 | F | II | – | – (0.15) | – (0.78) | – (1.72) |
| #17 | F | I | L858R | + (57.8) | + (64.8) | + (74.7) |
| #18 | M | I | – | – (1.70) | – (2.52) | – (20.1) |
| #19 | F | I | – | – (0.04) | – (5.18) | – (2.48) |
| #20 | F | II | L858R | + (39.8) | + (43.2) | + (50.0) |
| #21 | F | II | – | – (7.26) | – (13.9) | – (22.9) |
| #22 | M | I | L858R | + (59.9) | + (65.1) | + (76.4) |
| #23 | F | I | L858R | + (79.8) | + (79.8) | + (97.5) |

[a]AJCC (American Joint Committee on Cancer).
[b]For EGFR(+)HX103(+), the threshold value of 30.1% was used to differentiate *EGFR*-activating mutations.
[c]For EGFR(+), the threshold value of 36.3% was used to differentiate *EGFR*-activating mutations.
[d]For HX103(+), the threshold value of 31.0% was used to differentiate *EGFR*-activating mutations.
"–" represents no mutations or mutation negative; "+" represents mutation positive.
[e]Droplet digital PCR assay showing *EGFR*-activating mutation positive (*EGFR* 19del).

determined with the apparent $K_d$ values. The highest $K_d$ values of $2.1 \pm 0.4\,\mu M$ and $2.9 \pm 0.5\,\mu M$ were found in HCC827 and H1975, respectively, followed by PC-9, A431, Hela, and A549 cells (Supplementary Fig. 18).

**Association between HX103 labeling and *EGFR* mutation status in surgically resected samples**

*EGFR*-activating mutations are known to be associated with response to EGFR-TKI therapy, and we thus set out to evaluate the correlation between HX103 labeling and *EGFR*-activating mutations in surgical specimens from NSCLC patients. In this study, a total of 46 clinical specimens (tumor and the paired adjacent normal tissues) were obtained from 23 untreated NSCLC patients from stage I to IV (adenocarcinoma subtype) who underwent lung lobe resection (Table 1 and Supplementary Table 7). Following resection, the tissues were obtained, rapidly digested to single-cell suspensions, and analyzed by FACS. To obtain an unbiased quantification of probe labeling, we developed an HX103-based multicolor FACS assay with a panel of fluorescently labeled markers. Briefly, a depletion of CD45(+) leukocytes was carried out by adding an anti-CD45 antibody to exclude nonspecific labeling of the probe in infiltrating immune cells, followed by the addition of HX103 and EGFR antibody to obtain the percent of HX103 labeling in EGFR-positive tumor cells. The following gating strategy was applied for HX103-based FACS assay (Fig. 4a): (i) EGFR(+) cells were determined with total-EGFR antibody in comparison with

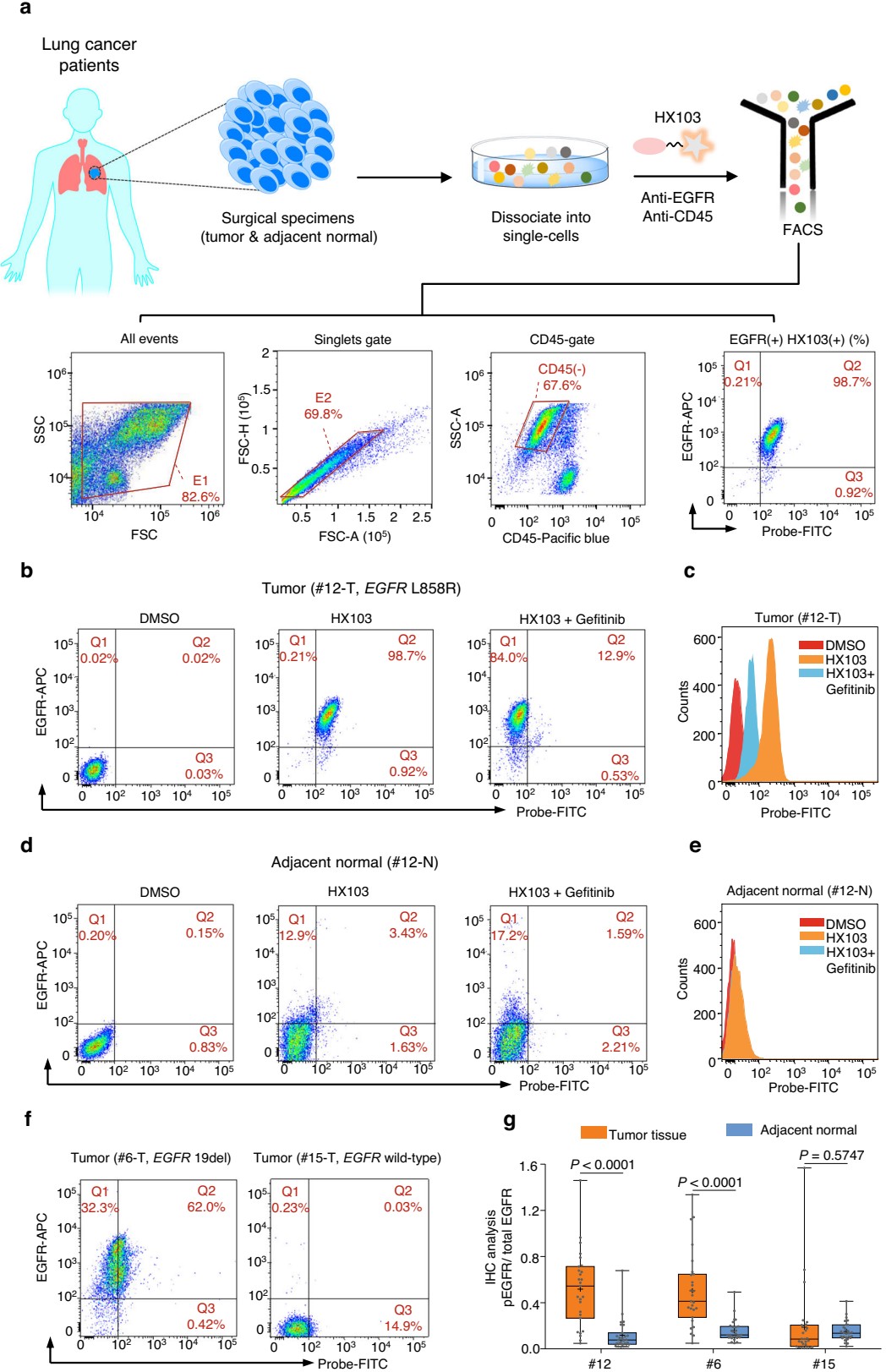

isotype control, (ii) the gates were set to have no positive events above the thresholds in DMSO-treated samples, (iii) EGFR-positive, HX103-positive and HX103-labeled EGFR-positive cells were calculated as the percentage values of EGFR(+) (Q1 + Q2), HX103(+) (Q2 + Q3) and EGFR(+)HX103(+) (Q2) cells in single-cell suspensions, respectively, (iv) specific ratios of these labeling parameters were determined by background correction with the corresponding DMSO-treated samples, and (v) the probe labeling was further validated by performing the same experiment in parallel with preincubation of gefitinib. For example, as shown in Fig. 4b, a total of 98.7% of EGFR(+)HX103(+) cells were detected in a tumor tissue (#12-T) from a patient with *EGFR* L858R (identified by Sanger sequencing, Supplementary Fig. 29), and the

**Fig. 4 | Application of HX103 to differentiate *EGFR*-activating mutations in surgically resected samples from NSCLC patients. a** General workflow of multi-color FACS assay for quantitative detection of active-EGFR with HX103. The ratio of probe-labeled EGFR-positive cells were differentiated by FACS using the following strategy: (i) the cell debris and cell clumps were gated out in the forward/side scatter plot (FSC/SSC gate); (ii) the CD45(+) cells were excluded by CD45-gate; (iii) the gates were set to have no positive events above the thresholds in DMSO-treated samples (e.g., left panel in **b**); (iv) HX103-labeled EGFR-positive cells were calculated as the percentage of EGFR(+)HX103(+) tumor cells in single-cell suspensions; (v) the labeling of HX103 was determined by background correction with DMSO-treated samples; (vi) the specific labeling of HX103 was examined by performing the same experiment in parallel with preincubation of gefitinib (e.g., the right panel in **b**). **b**, **d** Representative dual-parameter dot plots (HX103/EGFR antibody) showing the successful application of HX103 for detecting active-EGFR in tumor tissue (#12-T)

from an NSCLC patient with *EGFR* L858R mutation (identified by Sanger sequencing) (**b**), whereas low HX103 labeling was observed in the paired adjacent normal tissue (#12-N) (**d**). **c**, **e** Histograms showing fluorescence intensity differences between untreated sample (DMSO), HX103-treated sample, and sample with gefitinib preincubation. **f** Representative dual-parameter dot plots (HX103/EGFR antibody) showing the labeling of active-EGFR by HX103 in tumor tissues #6-T (*EGFR* 19del) and #15-T (*EGFR* wild-type). **g** Box and whisker plots indicate the ratio of pEGFR/total EGFR in NSCLC tumor tissues and the adjacent normal tissues determined from IHC analysis (Supplementary Fig. 19). Average values are shown as "+", *n* = 27 per group (3 biological replicates × 9 ROIs per picture). The line in the box corresponds to the median. The boxes go from the upper to the lower quartiles of the data. Whiskers represent Min to Max, shown with all data points. The statistical *P* values were calculated by the two-tailed Student's test. *P* values are two-sided. Source data are provided as a Source Data file.

percentage values of EGFR(+) and HX103(+) cells were 98.9 and 99.6%, respectively (Supplementary Fig. 26). Preincubation with gefitinib showed significant reductions in the ratio of EGFR(+)HX103(+) (98.7 vs. 12.9%), whereas the ratio of EGFR(+)HX103(−) was increased (0.21 vs. 84.0%) (Fig. 4b, right panel). Meanwhile, the mean fluorescence intensity of HX103 was also significantly decreased in the presence of gefitinib, demonstrating the specific labeling of HX103 in tumor sample #12-T (Fig. 4c). A minimal population (3.43%) of EGFR(+)HX103(+) cells were found in the paired adjacent normal tissue (#12-N, Fig. 4d), indicating low level of EGFR activation in normal tissue (confirmed by IHC analysis with pEGFR antibody, Fig. 4g). Furthermore, a ratio of 62.0% of EGFR(+)HX103(+) cells was found in tumor #6-T, where *EGFR* 19del was identified (Fig. 4f and Supplementary Fig. 28). For a tumor sample without *EGFR*-activating mutation (#15-T), a neglectable population (0.03%) of EGFR(+)HX103(+) cells were observed (Fig. 4f). To verify the expression of mutant EGFR and phosphorylated EGFR in these human tissues, IHC analysis was carried out with specific antibodies (Supplementary Fig. 19). Staining with EGFR L858R antibody was clearly seen in tumor sample #12-T (~56% of total EGFR is mutant), and the significant staining of 19del-specific antibody was only found in tumor sample #6-T (~50% of total EGFR is mutant) (Supplementary Fig. 19b). As expected, tumor sample #15-T was negative for both EGFR mutation-specific antibodies. Importantly, high levels of EGFR phosphorylation were detected in tumor tissues carrying *EGFR*-activating mutations (#12-T and #6-T), whereas minimal pEGFR was detected in the adjacent normal tissues and the tumor tissue without *EGFR*-activating mutation (#15-T) (Fig. 4g and Supplementary Fig. 19). These results are well consistent with probe labeling determined from HX103-based multicolor FACS analysis.

Next, we decided to analyze all the surgical specimens with an HX103-based FACS assay. Initially, the 23 adjacent normal tissues were studied and minimal (<13%) probe labeling was found [evaluated by HX103(+) and EGFR(+)HX103(+)]. However, as for EGFR(+) cells, the ratios were relatively higher when compared to HX103(+) and EGFR(+)HX103(+) (Supplementary Fig. 20 and Supplementary Table 5). Particularly, a significantly high level of EGFR expression (73.2%) was observed in normal sample #20-N, whereas the percent of HX103(+) cells was less than 10% (Supplementary Fig. 20 and Supplementary Table 5). This may be explained by the fact that EGFR is mostly autoinhibited in normal tissues. Subsequently, we applied an HX103-based FACS assay to analyze the 23 tumor tissues (Supplementary Figs. 26, 27). As shown in Fig. 5 and Supplementary Table 5, 12 tumor samples showed significantly high ratios of EGFR(+)HX103(+), while one sample (#3-T) with high EGFR expression showed low HX103 labeling. Notably, tumor sample #8-T showed a low amount of EGFR expression, but high HX103 labeling. Preincubation with gefitinib showed a significant reduction in mean fluorescence intensity, confirming the specific labeling of HX103 in tumor tissue #8-T (Supplementary Fig. 21 and Supplementary Table 6).

To evaluate the correlation between probe labeling and *EGFR* mutation status, Sanger sequencing (DNA-based approach) was carried out to verify *EGFR*-activating mutation status (L858R and 19del) (Supplementary Figs. 28, 29). The diagnosis data of *EGFR* mutation status for the 23 tumor samples is summarized in Table 1, including *EGFR* gene mutation status and ratios of EGFR(+)HX103(+), EGFR(+), and HX103(+) determined from HX103-based FACS analysis. Furthermore, the receiver operating characteristic (ROC) curves were obtained by comparing *EGFR* mutations with HX103-based FACS analysis (Supplementary Fig. 23). EGFR(+)HX103(+) showed an overall performance of an area under the curve (AUC) of 0.85 [95% CI = 0.682–1.02], while EGFR(+) and HX103(+) achieved AUC values of 0.86 [95% CI = 0.688–1.04] and 0.79 [95% CI = 0.592–0.983], respectively. We next attempted to use a specific threshold value to discriminate between *EGFR*-activating mutations (L858R or 19del) and wild-type. For EGFR(+)HX103(+), the ratio of 30.1% was used as an optimal cut-off point, and the results were considered mutation positive if ≥30.1% population of cells were identified. It can be seen that 10 out of 12 NSCLC surgical tumor samples with *EGFR*-activating mutations (L858R or 19del) were correctly identified as positive by EGFR(+)HX103(+) with a sensitivity of 83.3% (95% CI = 51.6%–97.9%), while 2 of the 11 *EGFR* wild-type samples were misdiagnosed to *EGFR*-activating mutations (Fig. 5b). The total accuracy of EGFR(+)HX103(+) for discrimination of *EGFR*-activating mutations in surgical specimens was 82.6%. When using 36.3% as the optimal cut-off point for EGFR(+), 11/12 (91.7%) cases of *EGFR* mutation-positive tumor samples showed positive for the ratio of EGFR(+), and 9/11 (81.8 %) cases of these *EGFR* mutation-negative tumor tissues were negative for the ratio of EGFR(+) (Supplementary Fig. 24). Using 31.0% cut-off point for HX103(+), 10/12 (83.3%) cases of *EGFR* gene mutation-positive samples were probe labeling positive, and 8/11 (72.7%) cases of *EGFR* gene mutation-negative cases were probe labeling negative (Supplementary Fig. 25). The comparative analysis revealed that the labeling parameters [EGFR(+), HX103(+) or EGFR(+)HX103(+)] showed a statistically significant difference between tumor samples with *EGFR*-activating mutations and wild-type (Supplementary Figs. 23–25). As expected, there were no associations between the labeling parameters and *EGFR* 19del or L858R mutation (Supplementary Figs. 23–25).

Subsequently, we carefully analyzed the cases with discordant results. Among them, two cases (#2-T and #9-T) were positive for FACS analysis with all three labeling parameters, but *EGFR*-activating mutation negative. Considering the detection limit of Sanger sequencing, we further evaluated the DNA samples of #2-T and #9-T by droplet digital PCR (ddPCR) (which is known as highly sensitive). Indeed, ddPCR assay revealed that tumor sample #9-T was *EGFR* 19del mutation-positive (Fig. 5c), which is in line with the results of our FACS analysis. Sample #2-T was *EGFR*-activating mutation negative in both ddPCR and Sanger sequencing tests (Supplementary Fig. 22), but showed 62.2% of EGFR(+)HX103(+) labeling, as well as a significant

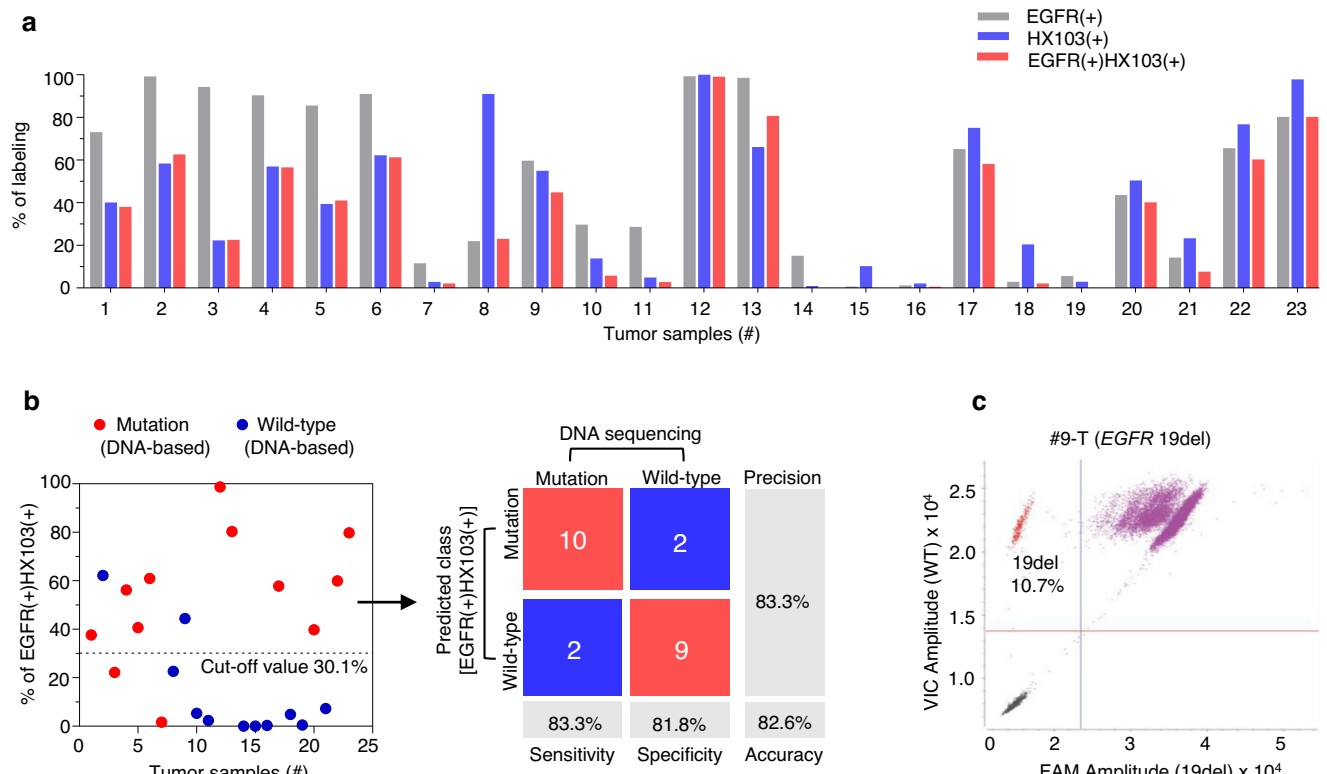

**Fig. 5 | Correlations between HX103-based FACS and *EGFR*-activating mutations in surgically resected samples from NSCLC patients. a** Bar chart showing the percentage of HX103(+), EGFR(+), and EGFR(+)HX103(+) in 23 surgical tumor samples (Table 1 and Supplementary Table 4). **b** Discriminating *EGFR*-activating mutations of 23 surgical tumor samples from NSCLC patients by the percent of EGFR(+)HX103(+). Confusion matrix analysis was performed by using the threshold of 30.1%, which is determined from ROC curves in Supplementary Fig. 23. It should be noted that the tumor tissues were from the patients diagnosed without any treatment. **c** Two-dimensional scatterplots of droplet digital PCR assay showing *EGFR* 19del positive for tumor sample #9-T. Source data are provided as a Source Data file.

response to gefitinib (Supplementary Fig. 20). This case might be explained by A431 with EGFR wild-type, ~60% of EGFR activation was detected by EGFR(+)HX103(+) in a similar vein (Fig. 3h). By contrast, #7-T was identified as *EGFR* L858R mutation-positive, but FACS negative for all labeling parameters. This might be due to the low level of EGFR expression (11.2%) in #7-T, where limited EGFR activation can be detected by HX103. A high level of EGFR expression (93.9%) was observed in #3-T with *EGFR* 19del, while the ratios of EGFR(+)HX103(+) and HX103(+) were low, indicating inactivation of EGFR mutant might be happened via a non-EGFR-dependent mechanism. Of note, heterogeneity of the tumor tissues can not be excluded during the tests, as tumor samples used for DNA extraction and single-cell suspension preparation were not the same tissue. Lastly, for tumor sample #8-T, although a high ratio of HX103(+) was observed, the percentage values for EGFR(+)HX103(+) and EGFR(+) were both low. This might be due to the shared protein target(s) of HX103 and gefitinib (Supplementary Fig. 21).

Taken together, these studies demonstrate that HX103-based FACS is able to discriminate between *EGFR*-activating mutations and wild-type by quantitively detecting active-EGFR. Although the correlation between FACS analysis and *EGFR*-activating mutation status is imperfect to some extent in surgical tissues, it is mainly due to myriad factors associated with EGFR activation, rather than only *EGFR* mutation status. Considering HX103 is an EGFR-TKI-based probe and mirrors the binding of the drugs to the EGFR kinase domain, it was then asked whether HX103-based FACS would show predictive value for drug response in patients with *EGFR*-mutant NSCLC.

## Correlation of HX103 labeling with response to EGFR-TKI therapy

We next performed an observational clinical study to evaluate the correlation of HX103 labeling to response to EGFR-TKI therapy. A total of 35 NSCLC patients from stage II to IV (did not receive any treatment) were enrolled in this study, while sufficient biopsy samples were obtained from 31 patients (Fig. 6a). After *EGFR* gene mutation analysis, 15/31 (48.4%) cases of the biopsy samples were *EGFR*-activating mutation-positive (L858R or 19del) (Table 3 and Supplementary Figs. 40 and 41). Single-cell suspensions were prepared from the 31 biopsy samples, followed by assessment with HX103-based FACS. First of all, the correlations between *EGFR* gene mutation status and HX103-based FACS analysis were studied. The results revealed that HX103(+) and EGFR(+)HX103(+) with AUC values of 0.93 (95% CI = 0.841–1.02) and 0.93 (95% CI = 0.838–1.02), respectively, showed better performance than EGFR(+) [AUC = 0.87 (95% CI = 0.733–1.01)] (Supplementary Fig. 31). Using the previous cut-off points, 11/15 (73.3%) cases of *EGFR* mutation-positive biopsy samples showed positive for EGFR(+)HX103(+), while 16/16 (100%) cases of *EGFR* mutation negative samples were all negative for EGFR(+)HX103(+) (Fig. 6c). As for the correlation between EGFR(+) and *EGFR* mutation status, the sensitivity and specificity were lower than that of HX103(+) and EGFR(+)HX103(+) in biopsy samples (Supplementary Fig. 32). For example, gene analysis as well as EGFR(+)HX103(+) and HX103(+) all revealed Patient 17 was *EGFR* mutation negative, but not EGFR(+) (39.4% ≥ the cut-off point of 36.3%) (Table 2). This can be explained by the low level of EGFR activation in this tumor sample. In a similar vein, surgical adjacent normal sample #20-N (without any *EGFR* mutations), also showed high EGFR

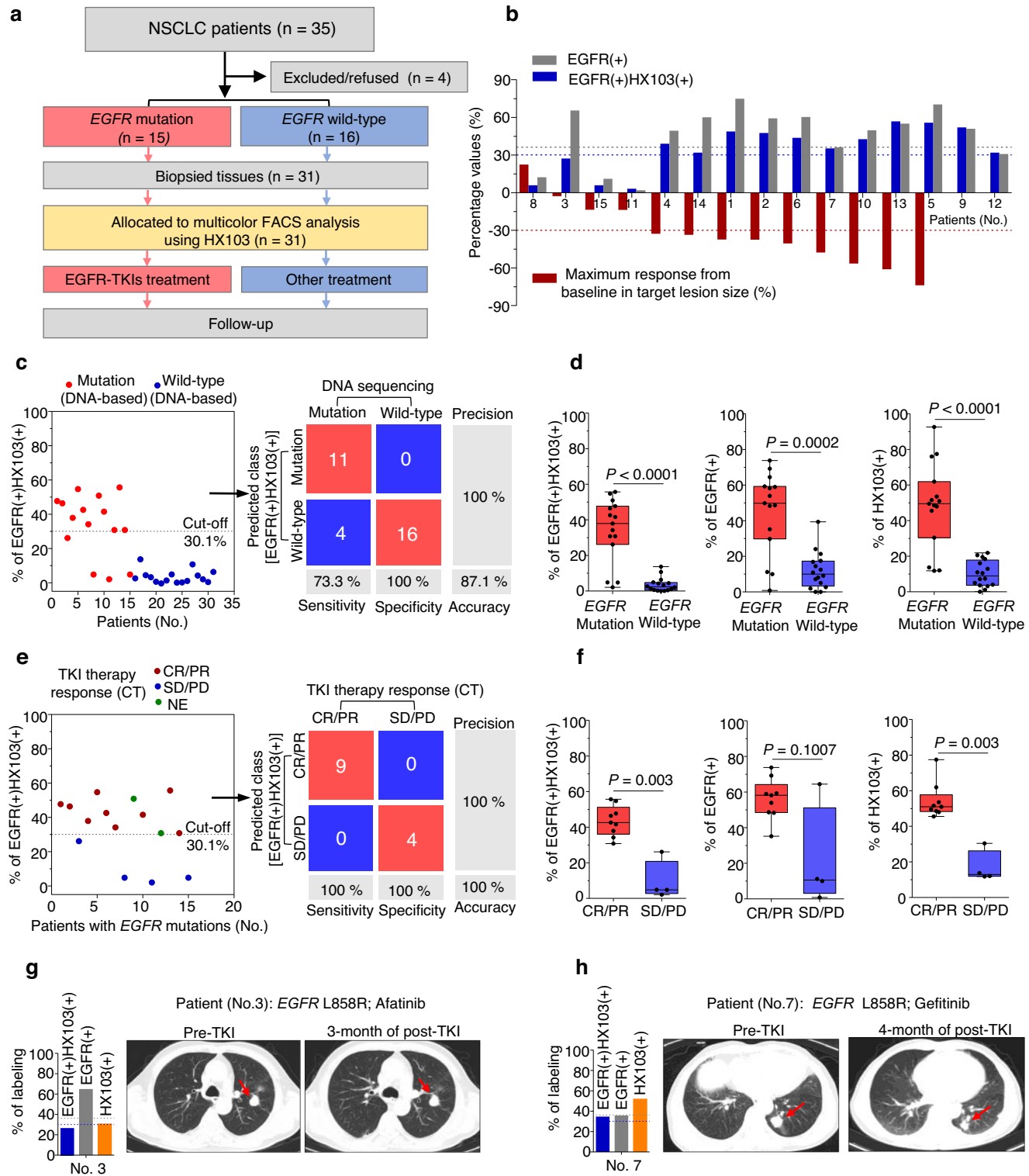

expression, but low probe labeling (Supplementary Table 5). Nevertheless, the mean percentage values for all the labeling parameters [HX103(+), EGFR(+), and EGFR(+)HX103(+)] were significantly higher in the *EGFR*-activating mutation-positive group than in *EGFR* mutation-negative group (Fig. 6d).

Regarding the imperfect correlation between *EGFR*-activating mutations and EGFR-TKI therapy response (e.g., ~20–30% of *EGFR*-activating mutation-positive NSCLC patients do not respond to EGFR-TKIs), we performed an observational clinical study to evaluate the relationship between HX103-based FACS analysis and response to

EGFR-TKI in *EGFR*-mutant NSCLC patients (Fig. 6a). In this study, the 15 patients carrying *EGFR*-activating mutations were selected to receive EGFR-TKI therapy according to the guidelines for the clinical use of targeted drugs in lung cancer patients. Therapy response to EGFR-TKI was examined by computed tomography (CT) imaging in 13 patients (CT scans of Patients 9 and 12 after EGFR-TKIs were not obtained) following the Response Evaluation Criteria for Solid Tumors (RECST) criteria: progressive disease (PD), stable disease (SD), partial response (PR), and complete response (CR)[62]. The 69.2% (9 of 13) of patients with *EGFR*-activating mutations demonstrated objective responses (CR or

**Fig. 6 | Prediction of EGFR-TKI sensitivity for NSCLC patients with *EGFR*-activating mutations by using the HX103-based FACS approach. a** Study design and patient allocation. The collected biopsy samples (*n* = 31) from NSCLC patients were divided into two groups according to *EGFR* mutation status, followed by the analysis of HX103-based FACS. Fifteen patients received EGFR-TKIs and therapy responses to EGFR-TKI were examined by CT imaging. **b** Waterfall plot for best percentage change in target lesion size (red), the labeling for EGFR(+) (gray) and EGFR(+)HX103(+) (blue) are shown for NSCLC patients carrying EGFR-activating mutations who received EGFR-TKIs (*n* = 15). **c** Analysis of the prediction for *EGFR*-activating mutations by the percentage of EGFR(+)HX103(+) in 31 biopsy samples from NSCLC patients. **d** Box plots of EGFR(+)HX103(+), EGFR(+) and HX103(+) in biopsy samples with *EGFR*-activating mutations and wild-type. The line in the box corresponds to the median. The boxes go from the upper to the lower quartiles of the data. Whiskers represent Min to Max, shown with all data points. For *EGFR*-activating mutation, *n* = 15; for *EGFR* wild-type, *n* = 16. Statistics was performed using Mann–Whitney test (two-sided). **e** Comparison of responsiveness to EGFR-TKI therapy with the percentage of EGFR(+)HX103(+) in 15 NSCLC patients carrying *EGFR*-activating mutations. Of note, the therapy responses of two patients (9 and 12) were evaluated. **f** Box plots of the percentage of EGFR(+)HX103(+), EGFR(+), or HX103(+) with responsiveness to EGFR-TKI. CR or PR cases were categorized as responders, and SD or PD cases, as non-responders. The line in the box corresponds to the median. The boxes go from the upper to the lower quartiles of the data. Whiskers represent Min to Max, shown with all data points. For CR/PR, *n* = 9; for SD/PD, *n* = 4. Statistics was performed using Mann–Whitney test (two-sided). **g, h** Representative CT scans comparisons of pre-TKI and post-TKI treatment showed the change in the size of tumor from NSCLC patients with *EGFR*-activating mutations, as well as the corresponding labeling parameters of EGFR(+)HX103(+) (blue), HX103(+) (orange) and EGFR(+) (gray). Source data are provided as a Source Data file.

PR) to EGFR-TKIs, while three patients showed SD and 1 patient (Patient 8) experienced PD after receiving EGFR-TKIs (Tables 2, 3 and Supplementary Table 8). By contrast, in the unselected patients, the response rate to other treatments was 36.4% (4 of 11) (Table 3 and Supplementary Table 8). Next, we analyzed the correlation between HX103-based FACS and responsiveness to EGFR-TKI. Interestingly, we found the labeling of EGFR(+)HX103(+) and HX103(+) showed high correlations with the maximum response in target lesion size to EGFR-TKIs (Fig. 6b). The ROC curves obtained from comparing therapy response to EGFR-TKI with HX103-based FACS analysis achieved AUC values of 1.0 for both EGFR(+)HX103(+) and HX103(+), whereas the AUC of 0.81 (95% CI = 0.463–1.15) was found for EGFR(+) (Supplementary Fig. 31). When the previous cut-off points were applied, the sensitivity and specifically were both 100% for EGFR(+)HX103(+) or HX103(+), while the sensitivity and specificity for EGFR(+) were 88.9 and 75.0%, respectively (Fig. 6e and Supplementary Fig. 32). Specifically, the four patients (Patients 3, 8, 11, and 15) with *EGFR*-activating mutations who did not respond to EGFR-TKIs, also showed negative for EGFR(+)HX103(+) and HX103(+) (Fig. 6b, Table 2, and Supplementary Figs. 34–37), suggesting high correlation between therapy response to EGFR-TKI and labeling of EGFR(+)HX103(+) or HX103(+). As for EGFR(+), a high percentage value (≥36.3%) was found in Patient 3, who was diagnosed as a non-responder to EGFR-TKI therapy (Fig. 6g), while Patient 7 showing negative for EGFR(+) (<36.3%) was diagnosed as a responder to gefitinib (Fig. 6h). This observation indicates an imperfect correlation between EGFR(+) and therapy response to EGFR-TKI, supporting the previous notion that the level of EGFR protein expression is not associated with EGFR-TKI response[63]. Moreover, despite a small decrease in the mean ratio of EGFR(+) was observed in patients who were non-responders (SD or PD), these differences did not reach statistical significance (non-responders: 21.6 ± 28.9% vs. responders: 56.2 ± 11.5%, *P* = 0.1007; Fig. 6f). However, the mean ratio of EGFR(+)HX103(+) or HX103(+) was significantly higher in the group of responders as compared with the group of non-responders [EGFR(+)HX103(+): non-responders: 9.5 ± 5.6% vs. responders: 43.5 ± 2.8%; HX103(+): non-responders: 17.0 ± 4.5% vs. responders: 54.1 ± 3.3%; Fig. 6f). As expected, no associations were found between therapy response to EGFR-TKI and *EGFR* 19del or L858R mutation (Supplementary Fig. 33).

These findings, taken together, demonstrate that the percent of active-EGFR determined by HX103-based FACS assay correlates well with tumor sensitivity to EGFR-TKI, highlighting the potential utility of an EGFR-TKI-based probe in precision medicine trials to improve lung cancer patient management.

## Cost-benefit analysis for HX103-based FACS analysis

To evaluate under which conditions upfront *EGFR* gene mutation testing with HX103-based FACS would be economically beneficial, a decision model was made to visualize the decision process (Supplementary Fig. 42). According to the model, there are three different situations: (1) testing the patients only with DNA-based method (e.g., NGS) (standard of practice), and the cost is that of NGS alone; (2) the other two possibilities involve upfront testing with HX103-based FACS and further, depending on the outcome, if it is probe labeling positive determined by HX103-based FACS, the patient will be proceeded directly to EGFR-TKI therapy without further NGS testing, and the cost is just of FACS alone; (3) if the probe labeling is negative, the additional NGS testing will be needed, and the cost is that of FACS and NGS combined.

Having calculated the cost of the HX103-based-FACS approach (~14$ per biopsy sample, Supplementary Table 9), we speculated that FACS would cost much less than NGS. Thus, the second situation would save money and the third one is the most expensive one. Therefore, the key to the cost-benefit analysis is to determine the probability that an NSCLC patient will be HX103-based FACS [EGFR(+)HX103(+)] positive. The key factors associated with this probability are the sensitivity and specificity of HX103-based FACS analysis, and the percentage of patients who are *EGFR* mutation-positive (~30–50% in Asia, ~10–15% in the United States and Europe). In the end, it is important to know how many times less expensive when FACS is compared with NGS, thereby allowing us to estimate which option is economically preferable. However, the cost of NGS varies from different countries, we thus made a cost analysis with different cost ratios between FACS and NGS, as well as various percentages of patients who are *EGFR* mutation-positive. Firstly, the sensitivity and specificity of EGFR(+)HX103(+) were assumed to be 73.3 and 100% (Fig. 6c), respectively, as determined in our biopsy study (comparing with *EGFR* mutations). As shown in Supplementary Fig. 42b, we can see that for the Asian population, with *EGFR* mutation-positive probability of ~30–50%, the cost of FACS analysis needs to be ~3 times less expensive than NGS that upfront FACS analysis would be economically justified. However, for U.S./European population, with a lower rate of *EGFR* mutation-positive population (~10–15%), the cost of FACS needs to be ~9–13 times less expensive than NGS. Furthermore, the results for different sensitivity of HX103-based FACS [EGFR(+)HX103(+)] from 50 to 99% were also evaluated. For the Asian population, it can be seen that if the sensitivity reaches above 85% (100% specificity of FACS was assumed), the cost of FACS needs to be approximately three times less expensive than NGS (Supplementary Fig. 42c). For the US/European population, if the sensitivity reaches above 85%, the cost of FACS needs to be approximately seven times less expensive than NGS (Supplementary Fig. 42d). Altogether, regarding the low cost of FACS analysis (~14$ per biopsy sample), HX103-based FACS may serve as a cost-effective test in precision medicine trials to stratify NSCLC patients for EGFR-TKI treatment.

## Discussion

EGFR-TKIs, tyrosine kinase inhibitors, have become widely used therapeutic agents for advanced NSCLC patients. The diagnostic approach

**Table 2 | Comparison of the prediction value for EGFR-TKI therapy response in NSCLC patients using different methods (*EGFR* genotype and different labeling parameters of HX103-based FACS).[a–d]**

| Patient (No.) | *EGFR* mutations | First-line treatment[a] | Therapy response[b] | EGFR(+)HX103(+) (%)[c] | EGFR(+) (%)[d] | HX103(+) (%)[e] |
|---|---|---|---|---|---|---|
| 1 | 19del | Afatinib | PR | +(47.7) | + (73.8) | + (48.0) |
| 2 | 19del | Afatinib | PR | +(46.4) | + (58.2) | + (53.5) |
| 3 | L858R | Afatinib | SD | – (26.2) | + (64.5) | – (30.4) |
| 4 | L858R | Dacomitinib | PR | +(38.0) | + (48.3) | + (49.5) |
| 5 | 19del | Gefitinib | CR | +(54.7) | + (69.2) | + (61.9) |
| 6 | L858R | Dacomitinib | PR | +(42.6) | + (59.3) | + (48.6) |
| 7 | L858R | Gefitinib | PR | +(34.2) | – (35.2) | + (51.5) |
| 8 | 19del | Dacomitinib | PD | – (4.84) | – (11.2) | – (13.7) |
| 9 | L858R | Icotinib | NE | +(50.9) | + (49.8) | + (92.6) |
| 10 | 19del | Icotinib | CR | +(41.6) | + (48.7) | + (51.0) |
| 11 | L858R | Icotinib | SD | – (2.07) | – (0.83) | – (11.8) |
| 12 | L858R | Erlotinib | NE | +(30.8) | – (29.8) | + (76.0) |
| 13 | 19del | Gefitinib | PR | +(55.7) | + (53.9) | + (77.5) |
| 14 | 19del | Gefitinib | PR | +(30.8) | + (58.9) | + (45.5) |
| 15 | L858R | Afatinib | SD | – (4.81) | – (10.0) | – (12.1) |
| 16 | – | Other | SD | – (2.61) | – (9.10) | – (5.01) |
| 17 | – | Other | PR | – (13.7) | + (39.4) | – (21.9) |
| 18 | – | Other | PR | – (4.44) | – (16.9) | – (11.0) |
| 19 | – | Other | SD | – (3.27) | – (12.5) | – (7.41) |
| 20 | – | Other | SD | – (0.60) | – (10.9) | – (3.16) |
| 21 | – | None | NE | – (−0.3) | – (21.3) | – (−1.8) |
| 22 | – | Other | NE | – (1.35) | – (−3.51) | – (3.59) |
| 23 | – | Other | NE | – (4.89) | – (7.13) | – (12.9) |
| 24 | – | Other | SD | – (0.10) | – (1.92) | – (4.01) |
| 25 | – | Other | NE | – (0.24) | – (14.5) | – (1.16) |
| 26 | – | Other | SD | – (1.08) | – (17.5) | – (7.12) |
| 27 | – | Other | SD | – (10.8) | – (24.0) | – (18.4) |
| 28 | – | None | NE | – (4.24) | – (2.70) | – (19.5) |
| 29 | – | Other | PD | – (2.10) | – (5.06) | – (10.5) |
| 30 | – | Other | PR | – (0.45) | – (−3.7) | – (20.6) |
| 31 | – | Other | PR | – (6.32) | – (8.10) | – (16.0) |

[a]NSCLC patients with *EGFR* mutations were selected for EGFR-TKI therapy, while patients without any *EGFR* mutations were treated with other methods, including chemotherapy, radiotherapy, and immunotherapy.

[b]Response to treatment was categorized into four categories: complete response (CR), partial response (PR), stable disease (SD), and progressive disease (PD). NE indicates not evaluable.

[c]For EGFR(+)HX103(+), the threshold of 30.1% was used to determine *EGFR*-activating mutations.

[d]For EGFR(+), the threshold of 36.3% was used to determine *EGFR*-activating mutations.

[e]For HX103(+), the threshold of 31.0% was used to determine *EGFR*-activating mutations. "–" represents no mutations or mutation negative; "+" represents mutation positive.

and management of lung cancer patients have changed significantly with the advent of these TKIs. A clinical challenge for physicians treating patients with *EGFR*-mutant NSCLC is to precisely identify individuals who may benefit from EGFR-TKI therapy. Although the identification of *EGFR* gene mutations has become standard of practice in the classification of individuals into subpopulations with different sensitivity to EGFR-TKI, some patients carrying *EGFR* mutations still do not respond to EGFR-TKI. Regarding this, structure-function-based approaches detecting mutant EGFR with functional activity may constitute better outcomes in the prediction of drug sensitivity to targeted therapies. Thus, there is considerable interest in developing robust methods to precisely quantify active-EGFR in tumors. Previously, ref. 64 developed an EGFR-TKI-based positron emission tomography (PET) probe ($^{18}$F-MPG) with high specificity to *EGFR*-activating mutant kinase. However, the widespread use of radiotracers is always limited by the requirement of a specific facility for radiolabeling. To address this, fluorescent probes with the virtue of simplicity and economy have occurred in medical diagnosis. Yet, to our knowledge, there is a limited description of EGFR-TKI-based fluorescent probes and none of them has been used in the clinical prediction of drug sensitivity. Here, we

have shown the application of an EGFR-TKI-based fluorogenic probe (HX103) in fully quantifying active-EGFR to predict drug sensitivity in NSCLC patients with *EGFR*-activating mutations.

The design and synthesis of HX103 was on the basis of a 4-anilinoquinazoline derivative. In this molecule, we functionalized the quinazoline ring to attach a fluorophore SBD via a small linker. Molecular docking studies and biochemical assays demonstrated the high targeting specificity of HX103 toward the kinase domain of EGFR by adopting a similar conformation to gefitinib in the active site. Furthermore, we found the accumulation of HX103 in living cells showed a high correlation with the levels of EGFR autophosphorylation. The preferential targeting of HX103 was prevented by the preincubation of gefitinib, confirming its specificity in the cellular system. HX103 thus can be used to identify the presence of EGFR activation, which is difficult to detect by other means. Based on HX103, we established a multicolor FACS assay with single-cell suspensions isolated from patients' tumors to achieve an unbiased quantification of active-EGFR in human tissues. In contrast to the DNA-based method, the interpretation of HX103-based FACS results depends on the number of stained live-cells rather than on results obtained from a tissue lysate,

**Table 3 | Summary of the clinicopathological and treatment characteristics**

| Characteristics | Total | EGFR L858R | EGFR 19del | EGFR (WT) |
|---|---|---|---|---|
| Patients, n (%) | 31 | 8 (26%) | 7 (23%) | 16 (52%) |
| **Ages (years)** | | | | |
| Median | 66 | 66 | 66 | 67 |
| Range | 49–87 | 52–86 | 49–76 | 53–87 |
| Sex (male: female) | 21:10 | 5:3 | 3:4 | 4:3 |
| **Stage (AJCC)** | | | | |
| II | 1 | 0 | 0 | 1 |
| III | 6 | 0 | 0 | 6 |
| IV | 24 | 8 | 7 | 9 |
| **EGFR(+)HX103(+) %** | | | | |
| ≥30.1 | 10 | 5 | 5 | 0 |
| <30.1 | 21 | 3 | 2 | 16 |
| EGFR-TKI therapy | 15 | 8 | 7 | 0 |
| **TKI therapy response** | | | | |
| CR/PR | 9 | 3 | 6 | - |
| SD | 3 | 3 | 0 | - |
| PD | 1 | 0 | 1 | - |
| Unknown (NE) | 2 | 2 | 0 | - |

and its evaluation is also fully quantitative and less subjective. Therefore, tumor samples with a low percentage of cancer cells carrying *EGFR* mutations that are often missed by the DNA-based method, can be detected by HX103-based FACS (e.g., surgical sample #9-T). In addition, one particular merit of HX103-based FACS is that the fluorogenic probe targets the active site of the EGFR kinase domain by competing with ATP, thereby providing valuable functional information on EGFR mutants, which cannot be obtained when the DNA-based method is used. This information is important because we intended to predict responsiveness to EGFR-TKI rather than only *EGFR* gene mutations. Overall, our FACS assay with the EGFR-TKI-based probe is a simple, sensitive, and reliable approach to quantitatively identify the presence of active-EGFR in living cells and human tissues.

Accordingly, HX103-based FACS offers an alternative opportunity to detect NSCLC patients potentially responsive to EGFR-TKI. In our clinical study, 31 biopsy samples were analyzed to identify patients who may benefit from EGFR-TKI therapy, and 15 of them were identified with *EGFR*-activating mutations (L858R or 19del). After evaluation of therapy response to EGFR-TKI, 69.2% (9/13) of patients with *EGFR*-activating mutations demonstrated objective responses (CR or PR), while 30.8% (4/13) of the *EGFR*-mutant patients showed SD or experienced PD. After HX103-based FACS analysis, we found the labeling of EGFR(+)HX103(+) or HX103(+) showed a high correlation with therapy response to EGFR-TKI, but not that of EGFR(+). These results demonstrate the ability of our FACS approaches to predict EGFR-TKI sensitivity via recognizing active-EGFR, which is difficult to detect by gene sequencing analysis. However, it is worth noting that the cut-off values here for the prediction of EGFR-TKI sensitivity should be considered as a pilot study, because we only included a very limited number of patients in this clinical study. Besides, the number of live-cells isolated from biopsy samples (~30 mg per tissue, obtained ~$10^4$ live-tumor cells) were often less than that from surgical specimens (~300 mg per tissue, obtained ~$10^6$ live-tumor cells), which may result in different cut-off points in biopsy samples (Supplementary Fig. 43).

Our study has several limitations, which still need to be addressed before the optimal translation of our approach into clinical use. First, as the results with clinical samples in this study was performed non-interventional prospectively, it might induce the potential bias. Second, our method should be improved and validated using a large-scale and multicentre clinical cohort. Furthermore, it is noteworthy that amplified wild-type *EGFR* was also associated with EGFR-TKI responses in some patients (~11%)[59,65], and HX103-based FACS measures EGFR activation not only in EGFR-mutant cells but also wild-type EGFR cells. As such, this method may also have utility in patients with EGFR overexpression/amplification to select individuals who may respond to EGFR-TKI. Further studies are therefore required to evaluate the predictive value of this method and to determine whether it can predict EGFR-TKI sensitivity in patients with EGFR overexpression/amplification, independent of DNA-based *EGFR* mutation diagnosis. Besides, although the less bulky fluorogenic benzofurazan fluorophore (e.g., SBD) is incorporated in the pharmacophore of EGFR-TKI, the probe still shows certain nonspecific labeling, which possibly leads to a potential false positive diagnosis. In the future, HX103 should be optimized with various fluorophores and linkers (e.g., PEG) to increase the sensitivity. Finally, for the clinical study, probe concentration, incubation time, as well as tumor cell isolation and live-cell collection during FACS measurements need to be further investigated and optimized. Additionally, whether other factors, such as optimal incubation temperature and other parameters for FACS analysis, might affect the percentage values of probe labeling should also be considered.

In conclusion, HX103-based FACS adds valuable functional information in selecting patients who may respond to EGFR-TKI, showing high potential to effectively personalize EGFR-TKI therapy in combination with gene sequencing analysis. This method provides immediate possibilities in performing in situ diagnosis for EGFR-TKI sensitivity with advantages such as simplicity, speed, low cost, and a low tendency for error. Importantly, this method using an EGFR-TKI-based chemical probe provides fully quantitative measurements of EGFR expression and functional activity, showing fairly good sensitivity and specificity for predicting EGFR-TKI sensitivity in *EGFR*-mutant NSCLC. These findings support the notion that a structure-function-based approach may improve the prediction of drug sensitivity to targeted therapies in oncogenes. Projecting forward, we are acquiring a larger number of biopsy samples from NSCLC patients to identify individuals who may benefit from EGFR-TKI with our approach.

## Methods
### Materials
All buffers and solutions were prepared using analytical grade reagents and solvents, as well as Millipore water (deionized using a Millio A10 Biocel™, with a 0.22 μm filter). Recombinant human EGFR (amino acids 668–1210, #PR7295B), EGFR L858R (amino acids 668–1210, #PR7447A), EGFR 19del (E746-A750) (amino acids 672–1210, amino acids 746–750 deleted, #PV6179) and EGFR T790M (amino acids 668–1210, #PR8052A) were purchase from Thermo Fisher, aliquoted, stored at −80 °C and thawed only once for one experiment. The following antibodies were purchased from Cell Signaling Technology (CST): EGFR wild-type (#4267), EGFR (L858R mutant, #3197), EGFR (E746-A750del, #2085), phospho-EGFR (Tyr1068, #3777), AKT (#4685), phospho-AKT (Ser473, #4060), ERK1/2 (Tyr1068, #4695), phospho-ERK1/2 (Thr202/Tyr204, #4370), GAPDH (#5174), Anti-rabbit IgG (HRP-linked antibody, #7074) and Rabbit mAb IgG isotype control (#3900). Goat anti-rabbit IgG (H + L) highly cross-adsorbed secondary antibody (Alexa Fluor Plus 647, #A32733), Goat anti-rabbit IgG (H + L) cross-adsorbed secondary antibody (Alexa Fluor 448, #A11008) and CD45 antibody eFluor 450 (#48-0459-42) were purchased from Invitrogen (Thermo Fisher). Alex Fluor 647 rabbit monoclonal to EGFR (total, #ab192982) was purchased from Abcam.

### Molecular docking to EGFR
All docking studies were performed in the Schrödinger suite (Schrödinger Release 2017-4: Maestro, Schrödinger, LLC, New York, NY, 2017). Structures of the cytoplasmic kinase domain of EGFR were downloaded from the Protein Data Bank (http://www1.rcsb.org/). The

crystal structure of EGFR wild-type was derived from the gefitinib-bound crystal structure 4WKQ, and the structure of EGFR L858R was derived from PD168393-bound crystal structure 4LQM. The structure of EGFR L858R/T790M was derived from 4-aminoindazolyl-dihydrofuro[3,4-d]pyrimidines-bound crystal structure 5EDP. The structures were prepared using the protein preparation wizard and ligands were prepared using LigPrep. Docking was done using induced fit docking with SP precision and poses with the best docking scores were manually examined. One pose per ligand was selected.

## Synthesis and characterization of HX103

The synthetic protocols of the fluorescent probe are given in Supplementary Methods. [1]H- and [13]C-NMR spectra were recorded on a Bruker AV spectrometer at 400 ([1]H) and 101 ([13]C) MHz. High-resolution mass spectra (HRMS) were recorded on a Thermo Scientific LTQ Orbitrap XL. HPLC purification was performed on a preparative LC-MS system (Agilent 1200 series) with an Agilent 6110 or 6120 mass spectrometer detector. The fluorescence spectroscopic experiments were performed on a BioTek microplate reader and Horiba Jobin Yvon-Edison Fluoromax-4.

## Cell lines and culture conditions

Cell lines including lung cancer cell lines HCC827 (CRL-2868), H1975 (CRL-5908), A549 (CCL-185), epidermoid carcinoma cell line A431 (CRL-1555), cervical carcinoma cell line Hela (CCL-2), breast cancer cell line MCF-7 (HTB-22) were purchased from the American Type Culture Collection (ATCC, Manassas, VA), lung cancer cell line PC-9 (RCB 4455) and acute T cell leukemia cell line Jurkat (RCB 3052) were obtained from RIKEN Cell Bank (Ibaraki, Japan). Cells were cultured in Dulbecco's modified Eagle medium containing 10% fetal bovine serum (Gibco), penicillin (100 U/mL), and streptomycin (100 μg/mL). Jurkat cells were cultured in RPIM 1640 (Gibco). All cells were maintained in a humidified 37 °C incubator with 5% CO$_2$. All cell lines used in our study have been authenticated and the mycoplasma tests of the cell lines were negative. Details of the cellular experiments are provided in Supplementary Methods.

## Western blot

Western blotting analysis was performed to investigate the expression of EGFR and downstream signaling pathway in various cancer cells. Cells were added at a density of $5 \times 10^5$ per well in six-well plates and serum-starved in DMEM without FBS. After 24 h of incubation, cells were treated with various concentrations of tested compounds for 2 h, followed by the stimulation with EGF (50 ng/μL, 0.5 μL, 25 ng, Sino Biological, #10605-HNAE) for 10 min. After a PBS washing step, the cells were harvested and lysed in RIPA lysis buffer (Thermo Scientific, #89900) on ice for 20 min. After sonication, cell lysates were centrifuged at 12,000×g at 4 °C for 20 min. The supernatant was collected and protein concentration was determined by a BCA protein quantitation kit (Vazyme) and normalized to 2 mg/mL. Subsequently, 20 μL of protein sample was loaded to each lane and resolved on a 10% SDS-PAGE gel. After that, proteins on the gel were transferred onto polyvinylidene difluoride (PVDF) membranes (0.45 μm, Amersham Bioscience). PVDF membranes were blocked with 5% BSA blocking solution for 2 h at room temperature, followed by incubation with different primary antibodies (4 °C, overnight), including EGFR (#4267, 1:1000), phospho-EGFR (#3777, 1:1000), Akt (#4685, 1:1000), phospho-Akt (#4060, 1:1000), ERK1/2 (#4695, 1:1000), phospho-ERK1/2 (#4370, 1:2000) and GAPDH (#5174, 1:2000). Next, the membranes were washed by TBST buffer (Tris buffer with 0.1% Tween-20) for three times, followed by incubation with anti-rabbit IgG (HRP-linked antibody, #7074, 1:5000) for 1 h at room temperature. Finally, the blots were washed in PBST and immunoreactive proteins were detected using a luminal solution with ECL enhancer and H$_2$O$_2$ by ChemiDoc MP (BioRad) with Image Lab 6.0 software. The band density of each band

was quantified and normalized to that of the corresponding GAPDH band to correct the possible differences in the content of protein loading. Each experiment was repeated 3 times for statistical analysis, and uncropped blots are in Source Data.

## Kinase inhibition assay

In general, all the compounds were initially diluted with DMSO to obtain stock dilutions, which were subsequently transferred to a 96-well plate containing 90 μL of Kinase buffer (1x) (50 mM HEPES pH 7.5, 0.01% Triton X-100). To start the experiment, 10 μL of the solution containing either recombinant human EGFR wild-type, EGFR L858R, EGFR 19del, or EGFR T790M in Kinase buffer (1x) was then added to a 384-well plate, followed by the addition of 5 μL of the tested compound dilution in 96-well plate. The obtained mixture was incubated for 10 min at room temperature. To initiate the kinase reaction, 10 μL of peptide solution containing 3 μM FAM-labeled peptide (GL Biochem, #112393) and 40 μM of ATP in Kinase buffer (1x) was added to each well in the 384-well plate. The reactions were incubated at room temperature for another 30 min and then terminated by the addition of 30 μL of stop buffer (100 mM HEPES, pH 7.5, 50 mM EDTA, 0.2% Coating Reagent #3, and 0.015% Brij-35). Finally, all samples were then subjected to analysis using a BioTek microplate reader[66]. IC$_{50}$ values were determined by plotting a log (inhibitor) vs. normalized response (Variable slope) dose-response curve generated using GraphPad Prism software.

## Confocal imaging analysis in living cells

For confocal fluorescence imaging experiments, about 10000 cells in 200 μL of medium supplemented with 10% fetal bovine serum and 1% penicillin/streptomycin were seeded in each well of an eight-well chamber (μ-Slide 8 well, Ibidi, #80826) and cultured for 24–48 h. The medium was replaced with 200 μL of serum-free medium (RPMI-1640 or DMEM) containing 5 μM fluorescent turn-on probe HX103 (0.1% DMSO). The cells were incubated with the probe at 37 °C for 1 h in a humidified incubator under 5% CO$_2$ in the air, followed by washing with PBS (three times). The cells were also co-stained with nuclear dye (Hoechst 33342, 1 μg/mL, 20 min, Invitrogen). For competition experiments, gefitinib was in situ pre-incubated for 1 h prior and then co-incubated with HX103 for another 1 h. The differential interference contrast (DIC) and fluorescence images were obtained using a Leica DMi8 confocal microscope. A white light laser was used for the light source in the confocal microscope. The fluorescence of HX103 was excited with 440 nm with emission collected at 500–550 nm. The fluorescent intensity of the confocal images was analysed by Leica LAS X software.

## Fluorescence-activated cell sorting (FACS) assay in cultured cells

**Quantification of HX103 labeling.** The FACS assay was performed according to the reported literatures[67,68]. Briefly, the cultured cells were seeded on the petri dish and counted using Count star BioMed automated cell counter. Around 10[6] live-cells were collected and were pelleted (2000×g, 3 min), followed by washing with PBS (2x). Cells were then resuspended in PBS buffer containing 0.1% BSA (1 × 10[6] cells/mL). Subsequently, 499 μL of cell suspension was transferred into a Falcon tube and treated with 0.5 μL of gefitinib (final concentration of 50 μM) or 0.5 μL of DMSO. The cells were incubated for 30 min at 37 °C. Then, 0.5 μL of HX103 (final concentration of 5 μM) or 0.5 μL DMSO was added and incubated for an additional 30 min at 37 °C. The treated cells were pelleted at (2000×g, 3 min), the supernatant was removed, and the pellet was resuspended in PBS buffer (250 μL) and washed for three times. The cell suspension was resuspended in PBS and analyzed with a FACSVerse$^{TM}$ flow cytometer (BD Biosciences, LSRFortessa). About 10,000 events per tube were analyzed and the mean fluorescence intensity was detected and analyzed.

**Quantification of EGFR expression and autophosphorylation with specific antibodies.** For the FACS assay with EGFR antibody, around $10^6$ live-cells were collected and washed with PBS (2x). The cells were fixed by 400 μL 4% paraformaldehyde for 15 min at room temperature, followed by permeabilization with 0.1% Triton X-100 for 10 min and blocking with 5% BSA for 30 min. Subsequently, the cells were centrifuged (2000×g, 3 min) and resuspended in 100 μL dilution of a specific EGFR antibody (L858R mutant, #3197, CST, 1:100; E746-A750del, #2085, CST, 1:200; phospho-EGFR, Tyr1068, #3777, CST, 1:400) in 3% BSA for 30 min at room temperature. After that, the cells were washed with 0.5% BSA/PBS (3x), followed by staining with 100 μL of secondary antibody (anti-rabbit Alexa Fluor 448 conjugate, #A11008, Invitrogen, 1:400) for 30 min at room temperature in the dark. The cells were then washed with 0.5% BSA/PBS (3x) and resuspended in 400 μL of 0.1% BSA/PBS. Notably, for the total-EGFR antibody, the Alexa Fluor 647 conjugated antibody (#ab192982, Abcam, 1:100) dilution in 3% BSA was used for one-step staining for 30 min. The resuspended cell suspension was then analyzed on a FACSVerseTM flow cytometer (BD Biosciences, LSRFortessa). Nonspecific fluorescence binding was omitted by Rabbit mAb IgG isotype control (#3900, CST, 1:100) and the corresponding mean fluorescence was subtracted from the signal measured with each primary antibody. About 10,000 events per tube were analyzed and the mean fluorescence intensity was detected and analyzed.

**Co-staining with EGFR antibody and HX103.** Briefly, the cells were collected and resuspended in 0.1% BSA/PBS buffer, followed by the addition of HX103 as described above. Subsequently, the treated cells were incubated for 30 min at 37 °C, followed by washing with PBS buffer three times. Next, the HX103-treated cells were fixed (4% paraformaldehyde, 15 min), permeabilized (0.1% Triton X-100, 10 min), and blocked (5% BSA, 30 min) at room temperature, followed by staining with total-EGFR antibody conjugated by Alexa Fluor 647 (ab192982, Abcam, 1:100). Finally, the co-stained cells were washed with 0.5% BSA/PBS for three times and resuspended in 400 μL of 0.1% BSA/PBS. The obtained cell suspension was ready for analysis. Nonspecific fluorescence binding was omitted by Rabbit mAb IgG isotype control (#3900, CST, 1:100) as described above.

**Co-staining with different EGFR antibodies.** Firstly, the collected cells were fixed (4% paraformaldehyde, 15 min), permeabilized (0.1% Triton X-100, 10 min), and blocked (5% BSA, 30 min), followed by staining with an EGFR-mutant-specific antibody [L858R mutant (#3197, CST,1:100)/E746-A750del (#2085, CST, 1:200)] or the pEGFR (#3777, CST, 1:400) antibody. Subsequently, the secondary antibody (anti-rabbit Alexa Fluor 448 conjugate, #A11008, Invitrogen, 1:400) was used as described above. Nonspecific fluorescence binding was omitted by Rabbit mAb IgG isotype control (#3900, CST, 1:100). Next, the stained cells were washed with 0.5% BSA/PBS (3x), followed by the addition of 100 μL total-EGFR antibody conjugated by Alexa Fluor 647 (1:100). After incubation for another 30 min at room temperature, the co-stained cells were washed with 0.5% BSA/PBS for three times and resuspended in 400 μL of 0.1% BSA/PBS. The obtained cell suspension was ready for analysis. Nonspecific fluorescence binding was omitted by Rabbit mAb IgG isotype control (#3900, CST, 1:100) as described above.

**Cell starvation and stimulation.** The cells were seeded on the petri dish and cultured at the normal condition with 10% FBS, and that was changed to serum-free condition for 24 h when the cells proliferated at ~80% confluence. Subsequently, the starved cells were treated with EGF (25 ng/mL) for 10 min at 37 °C, followed by washing with PBS three times. Next, the cells were co-stained with different EGFR antibodies (L858R mutant, #3197, CST, 1:100; E746-A750del, #2085, CST, 1:200; pEGFR, #3777, CST, 1:400; Alexa Fluor 647 anti-EGFR, ab192982,

Abcam, 1:100; secondary antibody (anti-rabbit Alexa Fluor 448 conjugate), #A11008, Invitrogen, 1:400) according to the experimental requirements. Nonspecific fluorescence binding was omitted by Rabbit mAb IgG isotype control (#3900, CST, 1:100) as described above.

## Multicolor fluorescence-activated cell sorting (FACS) analysis in clinical samples from NSCLC patients

The clinical specimens and biopsied tissues were obtained from NSCLC patients by qualified thoracic surgeons and pulmonary physicians, respectively, at West China Hospital in accordance with the ethical committee of the West China Hospital (No 2017.114), and written informed consent was obtained from all the patients, including for publishing clinical information potentially identifying individuals. We collected a total of 46 clinical specimens (tumors and the paired adjacent normal tissues) that had been completely removed surgically from 23 untreated NSCLC patients (from stage I to IV) who underwent lung lobe resection between January 2019 and June 2020. When a surgically resected tissue was obtained, freshly frozen tumors were sent to Sangon Biotech (Shanghai, China) for Sanger sequencing analysis.

The obtained tissues were dissociated into single-cell suspension using a standard protocol of primary cell isolation. Briefly, the obtained fresh tissue was placed in a petri dish on ice and cut into small pieces of ~2 mm. The pieces were transferred into a centrifuge tube (15 mL) containing the enzyme mixture of collagenase I (1 mg/mL, #17100017, Gibco) and IV (1 mg/mL, #17104019, Gibco), and digested for 30 min at 37 °C under gentle agitation. Subsequently, the digested sample was filtered through 70-μm and 40-μm cell strainer (Falcon), respectively, followed by the wash with serum-free medium (DMEM). The tumor cell pellet was then obtained by centrifugation at 500×g for 5 min at 4 °C for the removal of the supernatant. Then, the cell pellet was resuspended in serum-free medium and 1 mL of red blood cell lysis buffer was added and incubated for 5 min to remove the erythrocyte. The cells were pelleted and resuspended in a serum-free medium and ready for FACS analysis[67,68]. All specifications of multicolor FACS experiments and freshly isolated clinical tumor cells are provided in Supplementary Methods. Primary antibodies used in the staining include CD45 Pacific blue (#48-0459-42, Thermo Fisher, 1:200), Alexa Fluor 647 anti-EGFR (#ab192982, Abcam, 1:100), and isotype control (Alexa fluor 647 Rabbit IgG, ab199093, Abcam, 1:100). The data analysis of multicolor FACS experiments and the calculated ratios of EGFR(+)HX103(+) cells and MFI values are provided in Supplementary Methods.

## Study design and patient characteristics

An observational clinical study was performed to evaluate the correlation of HX103 labeling to response to EGFR-TKI therapy. This observational study was registered at West China Hospital and approved by the hospital ethics committee (Reference No. 2017.114), and informed written informed consent was obtained from all the patients, including for publishing clinical information potentially identifying individuals. Thirty-five patients with advanced lung cancer were recruited from West China Hospital between July 2020 and December 2021. Inclusion criteria were stage II-IV NSCLC according to the TNM classification system of the American Joint Committee on Cancer (AJCC). Sufficient biopsy samples were obtained from thirty-one patients (No.1–No.31) by the qualified pulmonary physicians for subsequent HX103-based FACS assay. Patient characteristics are provided in Supplementary Table 8. The enrolled 31 patients with sufficient biopsy samples were divided into two groups according to *EGFR* gene mutation status detected by DNA-based sequencing analysis (Fig. 6a). The first group included all consenting patients (n = 15) with *EGFR*-activating mutations (L858R or 19del), scheduled for treatment with EGFR-TKI (e.g., Afatinib and Gefitinib) according to the guidelines for the clinical use of targeted drugs in lung cancer patients. The

second group included patients (*n* = 16) with *EGFR* mutation-negative, scheduled for other treatment (e.g., immunotherapy and chemotherapy). The specific treatments for patients are provided in Supplementary Table 8. Our primary objective of this study is to identify the subset of lung cancer patients who may benefit from EGFR-TKI therapy, and hence to design appropriate treatment strategies for optimized clinical outcomes. Thus, the biopsy samples (*n* = 31) from patients of the two groups were subjected to quantitatively measure EGFR activity by HX103-based FACS, as well as to predict EGFR-TKI sensitivity in the corresponding patients. The observational Study Protocol is provided in Supplementary Notes (Observational Study Protocol).

All patients had a pretreatment tumor assessment by computer tomography scan, which was repeated to assess tumor response after ~12 weeks from the beginning of the treatment. Tumor response was evaluated by CT scan after ~2 months, with a confirmatory evaluation to be repeated in responders and in patients with stable disease at least ~4 to 6 weeks after the initial evaluation of response. Time to disease progression was assessed from the date of initiation of EGFR-TKI treatment to the date of progressive disease detection or to the date of the last contact. Tumor response was evaluated using RECIST. In general, response to EGFR-TKI treatment was categorized into the following four groups: corresponding to disappearance of all target lesions, complete response (CR); defined as a ≥30% decrease in tumor size from the baseline, partial response (PR); defined as a ≥20% increase in tumor size, progressive disease (PD); and defined as small changes that do not meet these above criteria, stable disease (SD)[62]. In this study, partial response (PR) and complete response (CR) cases were categorized as responders, and stable disease (SD) and progressive disease (PD) cases as non-responders. Of note, two patients (No. 9 and 12) did not come to us after EGFR-TKI due to personal reasons. Therefore, the CT scans of the two patients were missing and these two cases were categorized as not evaluable (NE).

### Statistical analysis
To assess the association between HX103-based FACS analysis and *EGFR*-activating mutations (Sanger sequence analysis), we performed statistical analysis with GraphPad Prism 6. Differences of the percentage values determined from FACS [EGFR(+)HX103(+), HX103(+), or EGFR(+)] between *EGFR*-activating mutations and EGFR wild-type (or between *EGFR* L858R and *EGFR* 19del) were analyzed using Mann–Whitney *U*-tests, whenever appropriate. The tests were on two sides and the statistical significance was set at $P < 0.05$. Quantitative values were expressed as means ± standard deviation (SD) or standard error of the mean (SEM) as indicated. Pearson's correlation was performed with GraphPad Prism 6 to examine the associations between probe labeling and EGFR phosphorylation among various cell lines. A receiver operating characteristic (ROC) curve analysis was computed for each labeling parameter, thereby assessing the ability of EGFR(+) HX103(+), HX103(+), or EGFR(+) to determine *EGFR*-activating mutations, and identifying the optimum cut-off values for each of them. The total area under the curve (AUC) and its 95% confidence interval (CI) were calculated. The sensitivity, specificity, and diagnostic accuracy were calculated in the prediction of *EGFR*-activating mutations and therapy response to EGFR-TKI.

## Data availability
The raw sequence data reported in this paper have been deposited in the Genome Sequence Archive in National Genomics Data Center, China National Center for Bioinformation/Beijing Institute of Genomics, Chinese Academy of Sciences (GSA-Human: HRA003138) that are publicly accessible at https://ngdc.cncb.ac.cn/gsa-human/browse/HRA003138. The authors declare that all experimental data generated in this study are provided in the paper and the supplementary information files/Source Data file. Individual de-identified participant data (including age, sex, *EGFR* mutation status, treatment, and CT imaging data) are available in Table 2, Supplementary Tables, and Figures. The study protocol is available from the authors in the supplementary information. Source data are provided with this paper.

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

## Acknowledgements

We gratefully acknowledge the voluntary participation of all the patients in this study. Dr. Ming Jiang is kindly acknowledged for performing molecular docking studies. The technical assistance of the Core Facility of West China Hospital (Li Chai, Yi Li, and Xing Xu) and West China School of Public Health and West China Fourth Hospital of Sichuan University (Xuejiao Song) is acknowledged. This work was supported by the National Natural Science Foundation of China (92159302 and 91859203 to W.L.; 22277084 to H.D.; 82104264 to Q.L.; 82100119 to C.W.), the Fundamental Research Funds for the Central Universities (SCU2022D025 to W.L.), National Major Scientific and Technological Special Project (Significant New Drugs Development, 2018ZX09201018-021 to W.L.), and the Science and Technology Project of Sichuan Province (2022ZDZX0018 to W.L.)

## Author contributions

H.D., Q.L., P.T., and W.L. designed the research. H.D., Q.L., and N.Y. synthesized compounds and carried out the characterization of the chemical probes. H.D., Q.L., C.W., Z.W., H.C., G.W., D.H., N.Y., Q.Y., M.Y., X.X., G.Z., C.C., Y.L., and F.L. performed the biological experiments, data analysis, methodology, and validation. C.W., P.T., and W.L. contributed to the clinical sample acquisition and patient management. H.D., Q.L., P.T., and W.L. wrote the paper. All authors reviewed and edited the manuscript.

## Competing interests

The authors declare no competing interests.
