## [Peer Review File · Nature Communications]

Reviewers' Comments:

Reviewer #1:

Remarks to the Author:

The authors present a very interesting use of an ex-vivo fluorescent TKI for detection of suitable tumors. The key elements of the study include the detection of functional mutations, and potential linear scale of predicting response. The primary, overall issues are twofold:

1. The authors only have a limited number of patients that were assessed in the prospective portion to determine response. This was certainly the most interesting aspect of the study. The failure to demonstrate clinical application (which is recognized in the Discussion) is the primary factor which reduces the overall impact of the study. The conjugation and in vitro binding...

2. There needs to be a cost analysis- how much for NGS vs this technique using flow and making the compound. Furthermore, it is unclear what is the additional value of information using the technique as described over conventional testing - more patients would be required in the properly suggest that one takes longer. As more technologies become available, it will be important to understand cost/benefit

Reviewer #3:

Remarks to the Author:

The exploration of a fluorescent epidermal growth factor receptor tyrosine kinase inhibitor (EGFR-TKI) by Deng et al to identify patients with activating EGFR mutations that would be responsive to EGFR-TKI therapy. The team develops HX103, a fluorogenic probe derived from gefitinib, that is sensitive to non-polar environments and increases its fluorescent quantum yield when in non-polar environments, including the hydrophobic binding pocket of EGFR. The authors first test HX103 in solvents, then in vitro assays, excised patient tissue from surgery and then tumor biopsies. The authors then used the results of activating mutation diagnosis + high HX103 fluorescence to select patients for EGFR-TKI therapy. Although this topic was exciting to investigate there are several fatal flaws in the data analysis and interpretation of the data that preclude the publication of this report in Nature Communications. As explained by the comments below, EGFR expression alone seems to have as good, if not better, correlation to TKI response than HX103 fluorescence signal. Therefore, the conclusion that HX103 is sensitive to EGFR activating mutations does not tell the whole story. In fact, it appears that really it is good at detecting EGFR concentration (which the antibodies do better in flow cytometry). Additionally, many of the results are overstated in the written text and the clinical treatment study is flawed in that treatment was only provided to those that tested positive and therefore there is no true control group.

The exploration of a fluorescent epidermal growth factor receptor tyrosine kinase inhibitor (EGFR-TKI) by Deng et al to identify patients with activating EGFR mutations that would be responsive to EGFR-TKI therapy. The team develops HX103, a fluorogenic probe derived from gefitinib, that is sensitive to non-polar environments and increases its fluorescent quantum yield when in non-polar environments, including the hydrophobic binding pocket of EGFR. The authors first test HX103 in solvents, then in vitro assays, excised patient tissue from surgery and then tumor biopsies. The authors then used the results of activating mutation diagnosis + high HX103 fluorescence to select patients for EGFR-TKI therapy. Although this topic was exciting to investigate there are several fatal flaws in the data analysis and interpretation of the data that preclude the publication of this report in *Nature Communications*. As explained by the comments below, EGFR expression alone seems to have as good, if not better, correlation to TKI response than HX103 fluorescence signal. Therefore, the conclusion that HX103 is sensitive to EGFR activating mutations does not tell the whole story. In fact, it appears that really it is good at detecting EGFR concentration (which the antibodies do better in flow cytometry). Additionally, many of the results are overstated in the written text and the clinical treatment study is flawed in that treatment was only provided to those that tested positive and therefore there is no true control group.

MAJOR COMMENTS:

HX103 in docking studies:

1. **Line 152-154** – It is unclear what is meant here. “The docking studies indicated that HX103 might be a fluorescent ligand towards EGFR with high-affinity. No strong interactions were detected in EGFR wild-type and double mutant.” These two sentences seem contradictory.
 - a. Is the first sentence referring only to L858R?
 - b. What strong interactions were not detected in wild-type and double mutant EGFR?
 - c. How much difference was measured between the binding of gefitinib and HX103?
Usually this is quantifiable by docking studies.

HX103 in solvent and *in vitro* studies (comments listed in order of importance):

2. **Figure 2C** – The fluorescence resulting from HX103 binding to wild type EGFR and L858R, and 19del mutants are equivalent. It is stated that HX103 is sensitive to EGFR activating mutations but really it is also sensitive to wild-type EGFR. **In cells, without confirming pathology, how would you be able to tell where the signal is coming from.**
3. **Figure 2E-L, 3 & S3, F** – HX103 fluorescence is correlated to EGFR expression, not mutant expression.
 - a. The differences in IC₅₀ values for HX103 and gefitinib binding wild type and mutated EGFR is overstated. The IC₅₀ values of wild-type EGFR for HX103 and gefitinib are 4.0 nM and 1.9 nM (no standard deviations reported), respectively, and is describe as being similar to gefitinib. Comparatively, the IC₅₀ values of HX103 and gefitinib with of

L858R and 19del mutant EGFR are 1.5 nM and 1.7 nM (L858R) and 1.3 and 1.2 nM (19del), respectively. These results are summarized as demonstrating “increased activity against EGFR L858R and EGFR 19del” (line 215). Without even considering the lack of standard deviation reported, it is doubtful that we can call the *nominal* change in IC₅₀ as “increased activity” and EGFR 19del is actually *increased* (again if there were any standard deviation reported these values would likely be considered the same). These claims are completely erroneous and highly misrepresented. It is even unclear how the EGFR T790M showed *~650-fold decrease* in EGFR activation – decreased compared to what? The only thing that gives a 650-fold decrease is when it is being compared to the other mutations, not wild-type EGFR which the start of the sentence states.

- b. The cell lines used have their p-EGFR/EGFR ratio determined through Western Blot and treatment with HX103. However, A431 and HeLa cells in Figure S3, which are stated to be wild-type EGFR (although this is never proven) show significant response to HX103 after stimulation with EGF. The difference in these cell lines is that the control (EGF- and HX103-) do not show high baseline activation in the absence of EGF. This is important because in Figure 3 A431 demonstrates some of the highest HX103 staining but does not have any mutant EGFR.
- c. A431 stains the highest with the EGFR antibody, second highest (tied for second?) in HX103 staining, and shows a response to gefitinib. These results are completely ignored in the human tissue scenario but appears to play a really important role in its interpretation.
- d. What is the percent of total EGFR that is mutant in the cells lines and human tissues? Typically, not all of the EGFR is mutant on a cell but a combination of the wild type and mutant. If the study is looking at activating mutations, why is staining with the antibodies for the mutations not shown? This is very important piece of this study and it is missing as far as I can tell. It is used for IHC but not for flow.
- e. Why was the K_d of HX103 not determined in A431 or HeLa cells, which do not have EGFR mutants.

4. **Figure 2A** - Autofluorescence, which in this case is the sum between the individual signals of HX103 and EGFR is a substantial portion of the total HX103 signal.

5. **Figure 2B & 2D** – Y-axis says normalized fluorescence intensity, but no explanation of normalized procedure. Also, in 2D there is $\pm 20\%$ error on the fluorescent signal. Why is the error so much, especially compared to that in 2B, which is quite low?

HX103 in human tissue sample (comments listed in order of importance):

6. **Figure 5D & 6B** – How were the thresholds selected from the receiver operating characteristic (ROC) curves? Was it the Optimal Cut-Off Point?
 - a. What decision led to selecting different thresholds for determining for the surgery samples (30.13%) and the biopsies (19.93%)? Wouldn't it make sense to either pool the

data together and make a single cut off point (more statistically accurate since you would have a higher number of samples) or just use one cut off chosen for each.

- b. Did you compare ROC curves for HX103(+), EGFR(+), and HX103(+) + EGFR(+)? Because EGFR(+) actually has better predictive value if you use a threshold cut off of 40% - this value can be used for both of the human surgery samples and biopsies. This conclusion completely changes the impact of the entire study.

Table 1. Diagnostic accuracy of *surgical tissue samples* comparing EGFR(+) to HX103(+) EGFR (+) using threshold presented in manuscript (left, white cells) and threshold for biopsy samples (right, yellow cells).

	EGFR(+)	HX103(+) EGFR (+)	HX103(+) EGFR (+)
True Positive	11	10	11
True Negative	9	9	8
False Positive	2	2	3
False Negative	1	2	1
Cut-off value	> 40%	> 30.13%	> 19.93

Table 2. Diagnostic accuracy of *biopsy samples* comparing EGFR(+) to HX103(+) EGFR (+) using threshold presented in manuscript (left, white cells) and threshold for biopsy samples (right, yellow cells).

	EGFR(+)	HX103(+) EGFR (+)	HX103(+) EGFR (+)
True Positive	6	5	6
True Negative	7	7	7
False Positive	0	0	0
False Negative	0	1	0
Cut-off value	> 40%	> 30.13%	> 19.93

- c. Again, as for the *in vitro* cells, it is unclear why the mutants aren't being quantified using flow cytometry and the antibodies against the mutant EGFR.
- d. The selection of only 6 patients to receive TKI therapy (unstated what therapy they received) does not prove the results of the team. All 13 people should have received TKI therapy and results presented. There was no negative control showing the proposed stratification actually worked.

MINOR COMMENTS:

- EGFR T790m is never introduced, as L858R and 19del mutations are.
- Significant figures – one significant figure in your standard deviation and match in average value for example $K_d = 2.73 \pm 0.39 \mu\text{M}$ should really be 2.7 ± 0.4

9. The abbreviation of EGFR-TKIs doesn't always work in context of a pluralized situation.
10. What is the level of EGFR mutation expression in the cell lines? You compare antibody binding of WT-EGFR to HX103 but not the mutations – you have antibodies for the mutated EGFR so this should be done.
11. Your definition of the percent of double positive cells is confusing. You should have one cell population (Q1, Q2, Q3) and see how many are EGFR+, how many are HX103+ and how many are double. It is unclear why the ratio of cell populations is based on different cell populations.
12. **Table 1** - The furthest right column is double stained HX103 (+) EGFR (+), but in the scatter plots (Fig. 5D and 6B) is called HX103 labeling (%), which seems like it should correspond to the HX103(+) not HX103 (+) EGFR (+).
13. For clinical patients, should have the patient number and the mutation shown in a table so we can compare, as with the initial 23 patient surgical study (as presented in Table 1).
14. In general, there are many grammatical errors.

Reviewer #4:

Remarks to the Author:

Deng et al reported the accuracy and utility of diagnosis of major EGFR mutation using fluorogenic probe.

The clinically available DNA-based testing for EGFR mutation is clinically useful, with high sensitivity and specificity.

Due to a variety of oncogenic driver gene alterations in NSCLC, we clinically need a multiplex testing with high performance, short turn-around-time, as well as low cost.

The new testing in the manuscript is only targeting major EGFR mutation, and cross-reacting WT EGFR, not as accurate as previous DNA-based technique, not along with clinical unmet needs. I think this manuscript is not suitable for Nature Communications.

Point-by-point reply for manuscript “A fluorogenic probe for predicting treatment response in non-small cell lung cancer with *EGFR*-activating mutations” The previous title is “Rapid and accurate diagnosis of activating *EGFR* mutations in non-small cell lung cancer with a new fluorogenic probe targeting *EGFR*”.

Please find below a point-by-point reply to the reviewers’ comments. We appreciate the time invested by the reviewer in evaluating our manuscript. The comments are very useful and have indeed improved our work.

For Reviewer #1

1.The authors only have a limited number of patients that were assessed in the prospective portion to determine response. this was certainly the most interesting aspect of the study. The failure to demonstrate clinical application (which is recognized in the Discussion) is the primary factor which reduces the overall impact of the study. The conjugation and in vitro binding...

In the revised version, the number of patients was increased up to 31 patients in total. According to gene sequencing analysis, 15 patients were identified with *EGFR*-activating mutations (L858R or 19del), while the rest were *EGFR* mutation-negative patients. The analysis of HX103-based FACS revealed that even although probe labeling [HX103(+) or *EGFR*(+)HX103(+)] showed imperfect correlation with *EGFR*-activating mutations, having evaluated the therapy response to *EGFR*-TKI, we found probe labeling showed higher correlation with *EGFR*-TKI response than that of gene analysis (See Figure 6 in the revised manuscript). Specifically, among the 15 patients who were identified with *EGFR*-activating mutations, 4 of them did not respond to *EGFR*-TKI treatment. This is well correlated with our results of probe labeling, demonstrating the ability of HX103-based FACS to predict *EGFR*-TKI sensitivity in *EGFR*-mutant NSCLC patients.

2.There needs to be a cost analysis- how much for NGS vs this technique using flow and making the compound. Furthermore more, it is unclear what is the additional value of information using the technique as described over conventional testing – more patients would be required in the properly suggest that one takes longer. As more technologies become available, it will be important to understand cost/benefit

We appreciate this comment by the reviewer. The cost-benefit analysis was included in the revised manuscript and the cost of HX103-based FACS for per sample was also

calculated (Table S9, see below). According to our calculation, the price for per biopsy sample was about 14\$ for HX103-based FACS, which is more than ~20 times less expensive than NGS (assumed to be ~300\$, which varies from country to country).

Currently, although the DNA-based approach is standard of practice, there are still ~20%-30% of *EGFR* mutation positive patients have no objective response or even experience disease progression after EGFR-TKI therapy. Consider this, HX103-based FACS as a function-based approach provides functional information on EGFR mutants, thereby having important value in predicting EGFR-TKI sensitivity. To make clear about the addition value of using HX103-based FACS approach, we evaluated the therapy response to EGFR-TKIs for patients who received EGFR-TKIs. According to the Response Evaluation Criteria for Solid Tumors (RECIST) criteria: progressive disease (PD), stable disease (SD), partial response (PR) and complete response (CR), we found 69.2% (9/13) of patients with *EGFR*-activating mutations demonstrated objective responses (CR or PR) to EGFR-TKIs, while 30.8% (4/13) of *EGFR*-mutant patients showed SD or PD after receiving EGFR-TKIs (3 patients showed SD and 1 patient experienced PD) in our clinical study. This suggests that the correlation between response to EGFR-TKI and *EGFR* mutations is imperfect, however, the responsiveness to EGFR-TKI was well correlated with probe labeling determined by our FACS approach, showing the important value of our method in predicting EGFR-TKI sensitivity. Next, to understand under which situation upfront *EGFR* gene mutation testing with our FACS would be economically beneficial, we have generated a cost-effectiveness decision model to visualize the decision process (Figure S39A, See below). Considering that the cost of NGS varies from different countries, we made a cost analysis with distinct ratios between NGS and FACS, as well as various percentages of patients who are *EGFR* mutation positive in a given population (See below, Figure S39B-D). The cost/benefit analysis suggests that our method may serve as cost-effective test in predicting EGFR-TKI sensitivity. The detailed discussion was shown in the main text of the revised manuscript.

Table S9. The cost of HX103-based FACS approach for per biopsy sample.

No.	Reagents	Price (\$)	Amount	Use (per biopsy sample)	Output	Price for per sample (\$)
1	CD45 antibody	250.7	100 μ L	0.5 μ L	200	1.25
2	EGFR antibody (conjugated with Alexa Fluor 647)	528.6	100 μ L	1.0 μ L	100	5.29
3	Alexa fluor 647 Rabbit IgG, monoclonal Isotype	514.3	100 μ L	0.5 μ L	200	2.57
4	Fixable viability kit	214.3	100 μ L	1.0 μ L	100	2.14
5	Fc block	111.4	100 μ L	2.0 μ L	50	2.23
6	HX103	1142.9	20 mg (3322 μ L of 10 mM stock solution was obtained)	0.5 μ L	6644	0.17
Total						13.65

Figure S39. (A) A decision process for determining under which conditions upfront *EGFR* mutation testing with HX103-based FACS would be economically preferable. Of note, 73% is the sensitivity between HX103-based FACS and *EGFR* gene mutations in biopsy sample. (B) Cost ratio between NGS and HX103-based FACS in correlation with the proportion of *EGFR* mutations in a given population. (C) Cost ratio between NGS and HX103-based FACS in correlation with the sensitivity of the HX103-based FACS assay taking into account the proportion of *EGFR* mutations in Asian population (~30%). (D) Cost ratio between NGS and HX103-based FACS in correlation with the sensitivity of the HX103-based FACS assay taking into account the proportion of *EGFR* mutations in U.S./European population (~15%).

For Reviewer #3

MAJOR COMMENTS:

HX103 in docking studies:

1. Line 152-154 – It is unclear what is meant here. “The docking studies indicated that HX103 might be a fluorescent ligand towards EGFR with high-affinity. No strong interactions were detected in EGFR wild-type and double mutant.” These two sentences seem contradictory.

a. Is the first sentence referring only to L858R?

b. What strong interactions were not detected in wild-type and double mutant EGFR?

c. How much difference was measured between the binding of gefitinib and HX103?

Usually this is quantifiable by docking studies.

(a) We apologize for the unclear statement. The sentence was corrected by “Having

calculated the binding energy of HX103 to EGFR wild-type and mutants (L858R, L858R/T790M), HX103 showed a slightly higher binding affinity toward EGFR L858R than others (Table S1).”

- (b) “No strong interactions were detected in EGFR wild-type and double mutant” has been corrected by “**Excepting for Met793, no strong interactions were detected in EGFR wild-type and L858R/T790M mutant (Lys745 and Phe795 interactions were absent, and the interaction of Pro794 in double-mutant was weak), where different orientations of the SBD group were observed**”
- (c) The binding energy difference of gefitinib and HX103 was quantified in the revised manuscript (Table S1, see below).

Table S1. Binding energy of HX103 and gefitinib to EGFR wild-type and mutants (L858R, L858R/T790M).

Target	Ligand	Vander Waal energy (kJ/mol)	Electrostatic energy (kJ/mol)	Polar solvation energy (kJ/mol)	SASA energy (kJ/mol)	Binding energy (kJ/mol)
EGFR L858R	Gefitinib	-105.7±3.6	-13.9±2.9	58.2±2.2	-10.0±0.4	-71.6±4.3
	HX103	-104.1±3.5	-9.1±3.6	56.9±2.3	-10.2±0.6	-66.6±2.5
EGFR wild-type	Gefitinib	-97.5±2.9	-15.4±3.0	60.2±2.2	-12.5±0.5	-65.2±5.0
	HX103	-102.4±3.4	-11.6±1.9	63.2±2.5	-9.0±0.6	-59.8±4.3
EGFR L858R/T790M	Gefitinib	-84.2±3.2	-15.9±2.4	58.9±2.0	-12.2±0.8	-53.3±3.1
	HX103	-88.6±2.6	-14.3±2.9	61.2±2.5	-13.1±0.4	-54.7±3.8

HX103 in solvent and in vitro studies (comments listed in order of importance):

2. Figure 2C – The fluorescence resulting from HX103 binding to wild-type EGFR and L858R, and 19del mutants are equivalent. It is stated that HX103 is sensitive to EGFR activating mutations but really it is also sensitive to wild-type EGFR. In cells, without confirming pathology, how would you be able to tell where the signal is coming from.

- We apologize for our improper statement. In the original description “A neglectable fluorescence was observed when in the presence of EGFR T790M, indicating the selectivity of HX103 towards EGFR L858R and 19del over EGFR T790M.”, we intended to compare EGFR primary mutants and T790M, but missing EGFR wild-type. In the revised manuscript, it was corrected as “**These results indicated that HX103 is selective toward EGFR wild-type and primary mutants (L858R and 19del), but less sensitive to the acquired resistance mutation EGFR T790M.**”
- In cellular experiments, we performed i) the experiments with EGFR mutant-specific antibodies and p-EGFR antibody to evaluate the levels of mutant

EGFRs and phosphorylated EGFR in these cell lines (Figure 3); ii) Western blot analysis with p-EGFR and other phosphorylated proteins in the signaling pathways to evaluate the effects of HX103 on EGFR activation and downstream signaling (Figure 2F-K, Figure S3); and iii) competition experiments with gefitinib to confirm the specific signal from HX103 (Figure S7, Figure 3A-B). Taken together, these studies provide the evidences for the specific labeling of HX103 in the cells.

3. Figure 2E-L, 3 & S3, F – HX103 fluorescence is correlated to EGFR expression, not mutant expression.

HX103 fluorescence correlated neither to mutant EGFR expression nor EGFR expression. As HX103 is designed based on EGFR-TKI, it is speculated that HX103 fluorescence would correlate to EGFR activation levels. In the revised manuscript, we performed additional experiments, and the results indeed demonstrated that HX103 fluorescence correlated to EGFR autophosphorylation (including the autophosphorylation of mutant EGFR and wild-type EGFR) (Figure 3G, Figure S12-13). The correlation of HX103 labeling with EGFR autophosphorylation provides important basis for the application of HX103 in predicting EGFR-TKI sensitivity in human tissues.

a. The differences in IC₅₀ values for HX103 and gefitinib binding wild type and mutated EGFR is overstated. The IC₅₀ values of wild-type EGFR for HX103 and gefitinib are 4.0 nM and 1.9 nM (no standard deviations reported), respectively, and is describe as being similar to gefitinib. Comparatively, the IC₅₀ values of HX103 and gefitinib with of L858R and 19del mutant EGFR are 1.5 nM and 1.7 nM (L858R) and 1.3 and 1.2 nM(19del), respectively. These results are summarized as demonstrating “increased activity against EGFR L858R and EGFR 19del” (line 215). Without even considering the lack of standard deviation reported, it is doubtful that we can call the nominal change in IC₅₀ as “increased activity” and EGFR 19del is actually increased (again if there were any standard deviation reported these values would likely be considered the same). These claims are completely erroneous and highly misrepresented. It is even unclear how the EGFR T790M showed ~650-fold decrease in EGFR activation – decreased compared to what? The only thing that gives a 650-fold decrease is when it is being compared to the other mutations, not wild-type EGFR which the start of the sentence states.

- We apologize that the standard deviations were missed in the kinase inhibition data. A table with IC₅₀ (95%CI) was included in the revised manuscript (Table S3, see below).
- The overstated description of the binding data of HX103 has been corrected as **“By contrast to EGFR wild-type, HX103 showed similar but slightly increased inhibition against EGFR L858R and EGFR 19del with IC₅₀ values of 1.5 nM (1.3-1.9 nM; 95% CI, n=3) and 1.3 nM (1.2-1.5 nM; 95% CI, n=3),**

respectively (Figure 2E and Table S3).”

- For the “**~650-fold decrease**”, we intended to compare the IC₅₀ of HX103 against EGFR T790M versus EGFR L858R. The unclear description was corrected as “**As expected, the probe’s inhibition towards the resistance mutation T790M was remarkably decreased with an IC₅₀ of 977 nM (832-1147 nM; 95% CI, n=3), ~650-fold and ~750-fold decrease when compared to the primary mutants L858R and 19del, respectively (Table S3).**”

Table S3. Inhibitory values for HX103 and gefitinib towards recombinant EGFR wild-type, EGFR L858R, EGFR 19del and EGFR T790M. Data represent average values with 95% Confidence Intervals (CI), n=3 per group.

	HX103	Gefitinib
	IC ₅₀ (95% CI) [nM]	IC ₅₀ (95% CI) [nM]
EGFR wild-type	4.0 (3.6-4.4)	1.9 (1.7-2.2)
EGFR L858R	1.5 (1.3-1.9)	1.7 (1.5-1.9)
EGFR 19del	1.3 (1.2-1.5)	1.2 (1.0-1.4)
EGFR T790M	977 (832-1147)	787 (694-893)

b. The cell lines used have their p-EGFR/EGFR ratio determined through Western Blot and treatment with HX103. However, A431 and HeLa cells in Figure S3, which are stated to be wild-type EGFR (although this is never proven) show significant response to HX103 after stimulation with EGF. The difference in these cell lines is that the control (EGF and HX103-) do not show high baseline activation in the absence of EGF. This is important because in Figure 3 A431 demonstrates some of the highest HX103 staining but does not have any mutant EGFR.

- First of all, in the revised manuscript, we demonstrated that A431 and HeLa cells were wild-type EGFR, and low levels of EGFR primary mutants were identified with mutant-specific antibodies (Figure 3F, Figure S13-14).
- Next, we demonstrated that HX103 staining was correlated with EGFR phosphorylation, but not mutant EGFR. As EGFR-activating mutations reside near the ATP cleft, resulting in the constitutive activation of EGFR in the absence of the cognate ligand. And A431 is previously reported as a carcinoma cell line with an unusually high number of EGFR inducing the activation of EGFR-pathway. With this regard, EGFR autophosphorylation (as well as HX103 labeling) (~60%, Figure 3G, Figure S10 and S12) can be observed in A431.
- Furthermore, in Figure S3 (Western blot analysis), the cells were all cultured in serum-starved condition without FBS for 24 h, followed by treatment with the probe and stimulation with EGF. As serum starvation of the cells for 24 h is a

common procedure to synchronize all cells to the same cell cycle phase, thereby removing the impact of cell cycle on cell response to the treatment. However, in Figure 3, we aimed to determine the uptake of HX103 in a natural cellular system, thus the cells were cultured physiologically, but not serum-starved. In the revised manuscript, we studied the effects of serum on EGFR activation in these cells (Figure 3H, see below). Interestingly, we found EGFR autophosphorylation was significantly reduced in A431 cells in the absence of serum, whereas the addition of serum had no effects on EGFR autophosphorylation in Hela cells. This indicates the level of EGFR autophosphorylation in A431 cells is higher than that in Hela cells.

Figure 3. (H) Bar chart showing percentage of EGFR(+)pEGFR(+) cells in the presence and absence of EGF or 10% FBS (Figure S15-16).

c. A431 stains the highest with the EGFR antibody, second highest (tied for second?) in HX103 staining, and shows a response to gefitinib. These results are completely ignored in the human tissue scenario but appears to play a really important role in its interpretation.

We appreciate this comment by the reviewer. The labeling difference of EGFR antibody and HX103 in A431 cells was applied to interpret the results in human tissue experiments. For example, the surgical sample #2-T [without *EGFR*-activating mutations (Figure S20)], showed ~98.9% of EGFR(+), ~58% of HX103(+) and 62.2% of EGFR(+)HX103(+), and also showed response to gefitinib (Figure S19). This case can be explained by A431 that EGFR autophosphorylation was also found in EGFR wild-type cells.

d. What is the percent of total EGFR that is mutant in the cells lines and human tissues? Typically, not all of the EGFR is mutant on a cell but a combination of the wild type and mutant. If the study is looking at activating mutations, why is staining with the antibodies for the mutations not shown? This is very important piece of this study and it is missing as far as I can tell. It is used for IHC but not for flow.

- In the revised manuscript, we determined the percent of total EGFR that is mutant in the cell lines with mutant-specific antibodies (Figure 3F, Figure S14). In H1975, we found ~70% of total EGFR was mutant to EGFR L858R. In HCC827 and PC-9, ~97% and ~85% of total EGFR were mutant to EGFR 19del, respectively. In EGFR wild-type (A431, A549 and HeLa) and EGFR-negative (MCF-7 and Jurkat) cells, no EGFR-activating mutants were identified.
- For human tissues, there are several technical difficulties for us to determine the percent of EGFR mutant with FACS analysis. Firstly, if we use mutant-specific antibodies (EGFR L858R and 19del) for HX103-based FACS, at least 7 single-cell suspension samples will be required for per human tissue (HX103, antibodies of L858R, 19del and CD45, live/dead cell discriminator kit, isotype control for L858R and 19del, as well as blank). However, in practice, the obtained single-cell suspensions from the human tissues were limited, and we can not ensure all the collected tissue samples (biopsy samples in particular) will be sufficient for obtaining 7 single-cell suspension samples. Furthermore, both antibodies are generated from rabbits. If we use these antibodies in the same experiment, it brings challenge for us to discriminate one from the other; and if we use the two antibodies separately, the required single-cell suspension samples and washing steps will be doubled (as too many washing steps will reduce the signal of probe labeling).
- Alternatively, we thus performed IHC experiments to determine the expression of EGFR mutants in human tissues (Figure S18). Having quantified the labeling of EGFR mutant antibodies in IHC, we found ~56% of EGFR L858R was observed in a human tumor tissue carrying *EGFR* L858R mutation, and ~50% of EGFR 19del mutant was observed in a human tumor tissue carrying *EGFR* 19del. Furthermore, the levels of EGFR phosphorylation were also determined by IHC with p-EGFR antibody (Figure 4G).

e. Why was the K_d of HX103 not determined in A431 or HeLa cells, which do not have EGFR mutants.

In the revised manuscript, the K_d values for the rest cells, including A431 and HeLa, were determined. And the results revealed that the highest K_d values of $2.1 \pm 0.4 \mu\text{M}$ and $2.9 \pm 0.5 \mu\text{M}$ were found in HCC827 and H1975, respectively, followed by PC-9, A431, HeLa and A549 (Figure S17).

4. Figure 2A - Autofluorescence, which in this case is the sum between the individual signals of HX103 and EGFR is a substantial portion of the total HX103 signal.

EGFR indeed shows some autofluorescence (~20%) in Figure 2A. However, after the autofluorescence correction, HX103 still showed >70% of fluorescence intensity. Importantly, in cell lines and human tissues, we included a blank control

(sample without probe) for each experiment, to remove the background signals, including autofluorescence. Based on these, we do not think the autofluorescence of EGFR will affect our results of HX103 labeling.

5. Figure 2B & 2D – Y-axis says normalized fluorescence intensity, but no explanation of normalized procedure. Also, in 2D there is $\pm 20\%$ error on the fluorescent signal. Why is the error so much, especially compared to that in 2B, which is quite low?

- We appreciate this comment. The normalized procedure for Figure 2B and 2D was included in the revised manuscript (in the part of Measurement of optical properties, Supporting information). Specifically, the fluorescence intensity of HX103 in the presence of EGFR wild-type was set at 100% to normalized that of the remaining samples.
- The high $\pm 20\%$ error in Figure 2D might be due to the reduced activity of recombinant EGFR kinase, as the recombinant EGFR tyrosine kinases can be degraded if it is handled improperly. Therefore, we repeated this experiment with a new stock of EGFR kinases. As shown in Figure 2D, the standard deviation was reduced to some extent, and similar to that in Figure 2B.

HX103 in human tissue sample (comments listed in order of importance):

6. Figure 5D & 6B – How were the thresholds selected from the receiver operating characteristic (ROC) curves? Was it the Optimal Cut-Off Point?

By using the software of GraphPad, we generated the ROC curves, which offered a series of sensitivity and specificity values with different thresholds. We then selected the optimal cut-off point with the best sensitivity and specificity.

a. What decision led to selecting different thresholds for determining for the surgery samples (30.13%) and the biopsies (19.93%)? Wouldn't it make sense to either pool the data together and make a single cut off point (more statistically accurate since you would have a higher number of samples) or just use one cut off chosen for each.

Previously, we considered that the single-cell suspensions obtained from biopsy samples was different from that of surgical samples, thus a different threshold for biopsy samples was used.

In the revised manuscript, to be logical, the first cut-off point from surgical samples (30.1%) was applied as the single cut-off point to predict *EGFR*-activating mutations and EGFR-TKI sensitivity (Figure 6).

b. Did you compare ROC curves for HX103(+), EGFR(+), and HX103(+) + EGFR(+)? Because EGFR(+) actually has better predictive value if you use a threshold cut off of 40% - this value can be used for both of the human surgery samples and biopsies. This conclusion completely changes the impact of the entire study.

Table 1. Diagnostic accuracy of surgical tissue samples comparing EGFR(+) to HX103(+) EGFR (+) using threshold presented in manuscript (left, white cells) and threshold for biopsy samples (right, yellow cells).

	EGFR(+)	HX103(+)EGFR (+)	HX103(+)EGFR (+)
True Positive	11	10	11
True Negative	9	9	8
False Positive	2	2	3
False Negative	1	2	1
Cut-off value	> 40%	> 30.13%	> 19.93

Table 2. Diagnostic accuracy of biopsy samples comparing EGFR(+) to HX103(+) EGFR (+) using threshold presented in manuscript (left, white cells) and threshold for biopsy samples (right, yellow cells).

	EGFR(+)	HX103(+)EGFR (+)	HX103(+)EGFR (+)
True Positive	6	5	6
True Negative	7	7	7
False Positive	0	0	0
False Negative	0	1	0
Cut-off value	> 40%	> 30.13%	> 19.93

In the revised manuscript, we compared the ROC curves for all three labeling parameters from HX103-based FACS [HX103(+), EGFR(+), and HX103(+)EGFR(+)] (Figure 5, Figure S22-24, S29). Indeed, EGFR(+) showed better correlations with *EGFR*-activating mutations in surgical samples. However, our aim is to predict EGFR-TKI sensitivity rather than *EGFR*-activating mutations. Although *EGFR*-activating mutations were found to be linked to an increased sensitivity to EGFR-TKIs, there are still ~20-30% of patients carrying *EGFR*-activating mutations showed stable or progress disease on EGFR-TKIs underlining the imperfect correlation. In this regard, we analyzed the CT imaging data (pre-TKI versus post-TKI) to examine the therapeutic response to EGFR-TKIs in *EGFR*-mutant NSCLC patients. In the revised manuscript, we found that 69.2% (9/13) of patients with *EGFR*-activating mutations demonstrated objective responses (CR or PR) to EGFR-TKIs, while 30.8% (4/13) of patients carrying *EGFR*-activating mutations showed SD or experienced PD after receiving EGFR-TKIs. (Table 2-3 and Table S8). Therefore, our clinical results also supported that *EGFR* gene mutation status was not equal to EGFR-TKI response.

Furthermore, having compared the labeling parameters of HX103-based FACS, we found the labeling of EGFR(+)HX103(+) and HX103(+) showed better correlation with response to EGFR-TKI than that of EGFR(+) (Figure 6F, Figure S30B, see

below). For example, Patient 3 carrying *EGFR* L858R mutation who showed stable disease after EGFR-TKI, had a high labeling of EGFR(+) (64.5% > the cut-off point of 36.3%), but low in EGFR(+)HX103(+) (~26.2% < the cut-off point of 30.1%)(Figure 6G, see below). This case suggests that EGFR(+)HX103(+) has better predictive value for the effectiveness of EGFR-TKI therapy than EGFR(+).

Overall, regarding therapy response to EGFR-TKI, the conclusion that EGFR-TKI-based probe may have high potential in prediction of EGFR-TKI sensitivity has not been changed.

Figure 6G. CT images for Patient 3.

c. Again, as for the in vitro cells, it is unclear why the mutants aren't being quantified using flow cytometry and the antibodies against the mutant EGFR.

As we explained before, there are several technical difficulties to quantify the EGFR mutant with FACS analysis (see the answer for Question 3d). The main issue is that a large amount of single-cell suspensions are required for human tissues when mutant-specific antibodies are applied in FACS. However, in practice, the collected human tissues are not big enough to obtain at least 7 single-cell suspensions. Considering our final aim is to study the correlation between probe labeling and

therapy response to EGFR-TKI, we thus determined the maximum response from baseline in target lesion size to EGFR-TKI treatment and studied the correlation between therapy response to EGFR-TKI and our FACS analysis.

d. The selection of only 6 patients to receive TKI therapy (unstated what therapy they received) does not prove the results of the team. All 13 people should have received TKI therapy and results presented. There was no negative control showing the proposed stratification actually worked.

Because our clinical study was a non-interventional study, and the treatment of the patients should fall within the current practice. According to the guidelines for the clinical use of targeted drugs in lung cancer patients, only the patients carrying *EGFR*-activating mutations were assigned to EGFR-TKI therapy. Therefore, the 15 patients (the number of patients was increased up to 31 in the revised manuscript) carrying *EGFR*-activating mutations were selected for EGFR-TKI (the specific therapy for each patient was shown in Table 2). The rest 16 patients showing *EGFR* gene mutation negative were assigned to other treatments according to the guidelines (Table S8).

MINOR COMMENTS:

7. EGFR T790m is never introduced, as L858R and 19del mutations are.

The following sentence was added in the revised manuscript to briefly introduce EGFR T790M mutation. “Furthermore, we applied HX103 to the “gatekeeper” point mutation T790M, the most common resistance mutation^{49,50},”

8. Significant figures – one significant figure in your standard deviation and match in average value for example $K_d = 2.73 \pm 0.39 \mu\text{M}$ should really be 2.7 ± 0.4

We appreciate the comment and the significant figures were all corrected as the reviewer suggested in the revised manuscript.

9. The abbreviation of EGFR-TKIs doesn't always work in context of a pluralized situation.

The use of EGFR-TKIs or EGFR-TKI was corrected in the revised manuscript.

10. What is the level of EGFR mutation expression in the cell lines? You compare antibody binding of WT-EGFR to HX103 but not the mutations – you have antibodies for the mutated EGFR so this should be done.

The expression of EGFR mutants in cell lines with mutant-specific antibodies was determined (Figure S13-14). Among them, HCC827 (~97%) showed high level of

EGFR 19del expression, followed by PC-9 (~85%), and the remaining cell lines showed minimal expression of EGFR 19del. In addition, H1975 was found with the highest expression of EGFR L858R (~70%) among these cells.

11. Your definition of the percent of double positive cells is confusing. You should have one cell population (Q1, Q2, Q3) and see how many are EGFR+, how many are HX103+ and how many are double. It is unclear why the ratio of cell populations is based on different cell populations.

In principle, one cell population (Q1, Q2 or Q3) should be observed in tumor cells, just like the case in culture cancer cell lines (e.g., H1975, HCC827, PC-9, etc.). However, lung cancer is a highly heterogeneous disease, and different levels of heterogeneity have been recognized in cancer, including interpatient, intratumor and intertumor. Thus, in some of our cases (e.g., #20-T, #22-T, #23-T and biopsy samples No.2-6), we observed several different cell populations in one sample. This can be explained by tumor heterogeneity that EGFR expression and kinase activity may be different in different cell populations.

Furthermore, to make clear about the percent of double positive cells (Q2), EGFR+ cells (Q1+Q2) and HX103+ cells (Q2+Q3), we included a table for per sample showing the calculations of the ratios for each sample (Figure S23-24 for surgical sample, Figure S31-34 for biopsy sample).

12. Table 1 - The furthest right column is double stained HX103 (+) EGFR (+), but in the scatter plots (Fig. 5D and 6B) is called HX103 labeling (%), which seems like it should correspond to the HX103(+) not HX103 (+) EGFR (+).

We apologize for the unclear description. We corrected “HX103 labeling (%)” by “the percentage value of HX103(+)” in our revised manuscript.

13. For clinical patients, should have the patient number and the mutation shown in a table so we can compare, as with the initial 23 patient surgical study (as presented in Table 1).

In the revised manuscript, we included a table (Table 2), which has the patient number, the mutation, and the percentage values of EGFR(+)HX103(+), HX103(+) and EGFR(+).

14. In general, there are many grammatical errors.

We have checked the manuscript carefully, and tried our best to avoid the grammatical errors.

For Reviewer #4

The clinically available DNA-based testing for EGFR mutation is clinically useful, with high sensitivity and specificity. Due to a variety of oncogenic driver gene alterations in NSCLC, we clinically need a multiplex testing with high performance, short turn-around-time, as well as low cost.

The new testing in the manuscript is only targeting major EGFR mutation, and cross-reacting WT EGFR, not as accurate as previous DNA-based technique, not along with clinical unmet needs. I think this manuscript is not suitable for Nature Communications.

We agree with the review that DNA-based testing for *EGFR* mutation is clinically useful. However, as for therapy response to EGFR-TKI, there are approximately 70%-80% (depending on the trials) of *EGFR* mutation-positive NSCLC patients who respond to EGFR-TKI treatment, and ~20%-30% of patients carrying *EGFR*-activating mutations do not show objective response when treated with EGFR-TKIs.

Furthermore, in our revised manuscript, 15 patients were identified with *EGFR*-activating mutation positive (L858R or 19del) among 31 enrolled patients, and 69.2% (9/13) of patients with *EGFR*-activating mutations demonstrated objective responses (CR or PR) to EGFR-TKIs, while 30.8% (4/13) of patients carrying *EGFR*-activating mutations showed SD or experienced PD after receiving EGFR-TKIs (CT scans of two patients after EGFR-TKIs were not obtained). These results further demonstrate that *EGFR*-activating mutation status does not always correlate with response to EGFR-TKI. These observations indicate that the correlation between *EGFR* mutation status and TKI therapy response is imperfect, and to predict EGFR-TKI response, *EGFR* gene mutations may not be the sole determinants.

Despite there are a variety of oncogenic driver gene alterations in NSCLC, L858R and 19del are the most common mutations, accounting for ~85% among all EGFR mutations. Furthermore, although with such a large number of oncogenic driver gene alterations in NSCLC, the pharmacophore of the current EGFR-TKIs is similar (4-anilinoquinazoline; although the third-generation EGFR-TKIs has a different pharmacophore, but they only target the acquired resistance mutation EGFR T790M). Importantly, in clinical, the aim to determine *EGFR* mutation status is to predict EGFR-TKI sensitivity, and to study the correlation with EGFR-TKI therapy response will be better than only with *EGFR* gene mutations.

Since our probe HX103 is based on the pharmacophore of EGFR-TKI, it has the inherent advantage to “on-target” predict EGFR-TKI sensitivity by quantifying the amount of functional EGFR in tumors. According to our results, although the imperfect correlation was found with *EGFR* mutations status, it showed high correlation with response to EGFR-TKI (Figure 6, see below), highlighting the ability of EGFR-TKI-based probe in predicting the sensitivity of EGFR-TKI. Based on this, we think HX103-based assay will have potential in predicting the sensitivity and efficacy of EGFR-TKI in clinical trials.

Figure 6C and E. (C) Analysis of the prediction for *EGFR*-activating mutations by the percentage of *EGFR*(+)*HX103*(+) in 31 biopsy samples from NSCLC patients. (E) Comparison of responsiveness to *EGFR*-TKI therapy with the percentage of *EGFR*(+)*HX103*(+) in 15 NSCLC patients carrying *EGFR*-activating mutations. Of note, therapy responses of 2 patients (9 and 12) were evaluated.

Furthermore, comparing with currently DNA-based methods (e.g., NGS), *HX103*-based FACS approach takes much less time (~6 hour, Table R1 see below). Importantly, having calculated the total cost of *HX103*-based FACS analysis (~14\$ per biopsy sample, Table S9 in supporting information), we found the cost of *HX103*-based FACS approach is much less expensive than gene analysis. For example, NGS usually costs at least ~300\$ per biopsy sample (although it differs from country to country). The cost/benefit analysis for *HX103*-based FACS was also provided in the revised manuscript. Besides, although the probe cross-reacts with *EGFR* wild-type (this is mainly due to the pharmacophore of *EGFR*-TKIs), it would not be an issue in predicting the effectiveness of *EGFR*-TKIs in clinical. Because *EGFR*-TKI drugs also cross-react with *EGFR* wild-type (For example, gefitinib showed inhibition towards *EGFR* wild-type with an IC_{50} of 1.9 nM, shown Figure S4).

Taken together, with regarding the cost, turn-around time, and the correlation with therapy response to *EGFR*-TKI, we think *EGFR*-TKI-based probe would have high value in predicting the effectiveness of *EGFR*-TKI therapy in clinical. Particularly, when in the combination with DNA-based technique, it would have high potential to reduce the false-positive results of gene analysis, thereby improving patients' outcomes.

Table R1. Time required for *HX103*-based FACS approach.

Procedures	Time (hour)	Total time for per sample
Single-cell isolation	3	~6 hour
HX103 incubation	0.5	
Antibody cocktail	0.5	
Blocking	0.5	
Washing	0.5	
FACS	0.5	
Others	0.5	

Reviewers' Comments:

Reviewer #1:

Remarks to the Author:

Appropriate revision

Reviewer #3:

Remarks to the Author:

Point-by-point reply for manuscript “A fluorogenic probe for predicting treatment response in non-small cell lung cancer with *EGFR*-activating mutations” The previous title is “Rapid and accurate diagnosis of activating *EGFR* mutations in non-small cell lung cancer with a new fluorogenic probe targeting *EGFR*”.

Please find below a point-by-point reply to the reviewers’ comments. We appreciate the time invested by the reviewer in evaluating our manuscript. The comments are very useful and have indeed improved our work.

For Reviewer #1

1.The authors only have a limited number of patients that were assessed in the prospective portion to determine response. this was certainly the most interesting aspect of the study. The failure to demonstrate clinical application (which is recognized in the Discussion) is the primary factor which reduces the overall impact of the study. The conjugation and in vitro binding...

In the revised version, the number of patients was increased up to 31 patients in total. According to gene sequencing analysis, 15 patients were identified with *EGFR*-activating mutations (L858R or 19del), while the rest were *EGFR* mutation-negative patients. The analysis of HX103-based FACS revealed that even although probe labeling [HX103(+) or *EGFR*(+)HX103(+)] showed imperfect correlation with *EGFR*-activating mutations, having evaluated the therapy response to *EGFR*-TKI, we found probe labeling showed higher correlation with *EGFR*-TKI response than that of gene analysis (See Figure 6 in the revised manuscript). Specifically, among the 15 patients who were identified with *EGFR*-activating mutations, 4 of them did not respond to *EGFR*-TKI treatment. This is well correlated with our results of probe labeling, demonstrating the ability of HX103-based FACS to predict *EGFR*-TKI sensitivity in *EGFR*-mutant NSCLC patients.

2.There needs to be a cost analysis- how much for NGS vs this technique using flow and making the compound. Furthermore more, it is unclear what is the additional value of information using the technique as described over conventional testing – more patients would be required in the properly suggest that one takes longer. As more technologies become available, it will be important to understand cost/benefit

We appreciate this comment by the reviewer. The cost-benefit analysis was included in the revised manuscript and the cost of HX103-based FACS for per sample was also calculated (Table S9, see below). According to our calculation, the price for per biopsy

sample was about 14\$ for HX103-based FACS, which is more than ~20 times less expensive than NGS (assumed to be ~300\$, which varies from country to country).

Currently, although the DNA-based approach is standard of practice, there are still ~20%-30% of *EGFR* mutation positive patients have no objective response or even experience disease progression after EGFR-TKI therapy. Consider this, HX103-based FACS as a function-based approach provides functional information on EGFR mutants, thereby having important value in predicting EGFR-TKI sensitivity. To make clear about the addition value of using HX103-based FACS approach, we evaluated the therapy response to EGFR-TKIs for patients who received EGFR-TKIs. According to the Response Evaluation Criteria for Solid Tumors (RECIST) criteria: progressive disease (PD), stable disease (SD), partial response (PR) and complete response (CR), we found 69.2% (9/13) of patients with *EGFR*-activating mutations demonstrated objective responses (CR or PR) to EGFR-TKIs, while 30.8% (4/13) of *EGFR*-mutant patients showed SD or PD after receiving EGFR-TKIs (3 patients showed SD and 1 patient experienced PD) in our clinical study. This suggests that the correlation between response to EGFR-TKI and *EGFR* mutations is imperfect, however, the responsiveness to EGFR-TKI was well correlated with probe labeling determined by our FACS approach, showing the important value of our method in predicting EGFR-TKI sensitivity. Next, to understand under which situation upfront *EGFR* gene mutation testing with our FACS would be economically beneficial, we have generated a cost-effectiveness decision model to visualize the decision process (Figure S39A, See below). Considering that the cost of NGS varies from different countries, we made a cost analysis with distinct ratios between NGS and FACS, as well as various percentages of patients who are *EGFR* mutation positive in a given population (See below, Figure S39B-D). The cost/benefit analysis suggests that our method may serve as cost-effective test in predicting EGFR-TKI sensitivity. The detailed discussion was shown in the main text of the revised manuscript.

Table S9. The cost of HX103-based FACS approach for per biopsy sample.

No.	Reagents	Price (\$)	Amount	Use (per biopsy sample)	Output	Price for per sample (\$)
1	CD45 antibody	250.7	100 μ L	0.5 μ L	200	1.25
2	EGFR antibody (conjugated with Alexa Fluor 647)	528.6	100 μ L	1.0 μ L	100	5.29
3	Alexa fluor 647 Rabbit IgG, monoclonal Isotype	514.3	100 μ L	0.5 μ L	200	2.57
4	Fixable viability kit	214.3	100 μ L	1.0 μ L	100	2.14
5	Fc block	111.4	100 μ L	2.0 μ L	50	2.23
6	HX103	1142.9	20 mg (3322 μ L of 10 mM stock solution was obtained)	0.5 μ L	6644	0.17
Total						13.65

Figure S39. (A) A decision process for determining under which conditions upfront *EGFR* mutation testing with HX103-based FACS would be economically preferable. Of note, 73% is the sensitivity between HX103-based FACS and *EGFR* gene mutations in biopsy sample. (B) Cost ratio between NGS and HX103-based FACS in correlation with the proportion of *EGFR* mutations in a given population. (C) Cost ratio between NGS and HX103-based FACS in correlation with the sensitivity of the HX103-based FACS assay taking into account the proportion of *EGFR* mutations in Asian population (~30%). (D) Cost ratio between NGS and HX103-based FACS in correlation with the sensitivity of the HX103-based FACS assay taking into account the proportion of *EGFR* mutations in U.S./European population (~15%).

For Reviewer #3

Individual response to comments are listed below in red. In general, the response to the review seemed thorough although there is a significant lack of validation of any claimed 'correlation' using a defined correlation test. All correlations appear to be qualitative observations. This needs to be remedied prior to publication and language adjusted accordingly.

MAJOR COMMENTS:

HX103 in docking studies:

1. Line 152-154 – It is unclear what is meant here. “The docking studies indicated that HX103 might be a fluorescent ligand towards EGFR with high-affinity. No strong interactions were detected in EGFR wild-type and double mutant.” These two sentences seem contradictory.

a. Is the first sentence referring only to L858R?

b. What strong interactions were not detected in wild-type and double mutant

EGFR?

c. How much difference was measured between the binding of gefitinib and HX103?

Usually this is quantifiable by docking studies.

- (a) We apologize for the unclear statement. The sentence was corrected by **“Having calculated the binding energy of HX103 to EGFR wild-type and mutants (L858R, L858R/T790M), HX103 showed a slightly higher binding affinity toward EGFR L858R than others (Table S1).”**

Thank you for clarifying. However, there are still some remaining questions. What is the error being reported here? Are these binding energies statistically different? They have overlapping errors (standard deviations?), which suggests they are not different, and you cannot make any statement about one being “slightly higher”.

- (b) “No strong interactions were detected in EGFR wild-type and double mutant” has been corrected by **“Excepting for Met793, no strong interactions were detected in EGFR wild-type and L858R/T790M mutant (Lys745 and Phe795 interactions were absent, and the interaction of Pro794 in double-mutant was weak), where different orientations of the SBD group were observed”**

The details of this study are very vague, what consists of a strong interaction and what is a weak interaction? There are no definitions or quantitations.

- (c) The binding energy difference of gefitinib and HX103 was quantified in the revised manuscript (Table S1, see below).

Thank you for the addition of this table. What are the values and errors reported? Mean \pm standard deviation? You should only have one significant figure in the error and report mean to that number. For instance **-71.6 \pm 4.3 should be written -72 ± 4 .**

Table S1. Binding energy of HX103 and gefitinib to EGFR wild-type and mutants (L858R, L858R/T790M).

Target	Ligand	Vander Waal energy (kJ/mol)	Electrostatic energy (kJ/mol)	Polar solvation energy (kJ/mol)	SASA energy (kJ/mol)	Binding energy (kJ/mol)
EGFR L858R	Gefitinib	-105.7 \pm 3.6	-13.9 \pm 2.9	58.2 \pm 2.2	-10.0 \pm 0.4	-71.6\pm4.3
	HX103	-104.1 \pm 3.5	-9.1 \pm 3.6	56.9 \pm 2.3	-10.2 \pm 0.6	-66.6\pm2.5
EGFR wild-type	Gefitinib	-97.5 \pm 2.9	-15.4 \pm 3.0	60.2 \pm 2.2	-12.5 \pm 0.5	-65.2\pm5.0
	HX103	-102.4 \pm 3.4	-11.6 \pm 1.9	63.2 \pm 2.5	-9.0 \pm 0.6	-59.8\pm4.3
EGFR L858R/T790M	Gefitinib	-84.2 \pm 3.2	-15.9 \pm 2.4	58.9 \pm 2.0	-12.2 \pm 0.8	-53.3\pm3.1
	HX103	-88.6 \pm 2.6	-14.3 \pm 2.9	61.2 \pm 2.5	-13.1 \pm 0.4	-54.7\pm3.8

HX103 in solvent and in vitro studies (comments listed in order of importance):

2. Figure 2C – The fluorescence resulting from HX103 binding to wild-type EGFR and L858R, and 19del mutants are equivalent. It is stated that HX103 is sensitive to EGFR activating mutations but really it is also sensitive to wild-type EGFR. In cells, without confirming pathology, how would you be able to tell where the signal is coming from.

- We apologize for our improper statement. In the original description “A neglectable fluorescence was observed when in the presence of EGFR T790M, indicating the selectivity of HX103 towards EGFR L858R and 19del over EGFR T790M.”, we intended to compare EGFR primary mutants and T790M, but missing EGFR wild-type. In the revised manuscript, it was corrected as **“These results indicated that HX103 is selective toward EGFR wild-type and primary mutants (L858R and 19del), but less sensitive to the acquired resistance mutation EGFR T790M.”**

Thank you for this clarification.

- In cellular experiments, we performed i) the experiments with EGFR mutant-specific antibodies and p-EGFR antibody to evaluate the levels of mutant EGFRs and phosphorylated EGFR in these cell lines (Figure 3); ii) Western blot analysis with p-EGFR and other phosphorylated proteins in the signaling pathways to evaluate the effects of HX103 on EGFR activation and downstream signaling (Figure 2F-K, Figure S3); and iii) competition experiments with gefitinib to confirm the specific signal from HX103 (Figure S7, Figure 3A-B). Taken together, these studies provide the evidences for the specific labeling of HX103 in the cells.

The antibodies used for mutant specific staining are not described anywhere that I can see. The additional data significantly helps.

3. Figure 2E-L, 3 & S3, F – HX103 fluorescence is correlated to EGFR expression, not mutant expression.

HX103 fluorescence correlated neither to mutant EGFR expression nor EGFR expression. As HX103 is designed based on EGFR-TKI, it is speculated that HX103 fluorescence would correlate to EGFR activation levels. In the revised manuscript, we performed additional experiments, and the results indeed demonstrated that HX103 fluorescence correlated to EGFR autophosphorylation (including the autophosphorylation of mutant EGFR and wild-type EGFR) (Figure 3G, Figure S12-13). The correlation of HX103 labeling with EGFR autophosphorylation provides important basis for the application of HX103 in predicting EGFR-TKI sensitivity in human tissues.

You have performed no tests for correlation. Your conclusion that “HX103 correlates” to activation is a qualification of the results. You need to perform a Pearson or

Spearman correlation, or another relevant correlation test, to substantiate these claims.

a. The differences in IC_{50} values for HX103 and gefitinib binding wild type and mutated EGFR is overstated. The IC_{50} values of wild-type EGFR for HX103 and gefitinib are 4.0 nM and 1.9 nM (no standard deviations reported), respectively, and is describe as being similar to gefitinib. Comparatively, the IC_{50} values of HX103 and gefitinib with of L858R and 19del mutant EGFR are 1.5 nM and 1.7 nM (L858R) and 1.3 and 1.2 nM(19del), respectively. These results are summarized as demonstrating “increased activity against EGFR L858R and EGFR 19del” (line 215). Without even considering the lack of standard deviation reported, it is doubtful that we can call the nominal change in IC_{50} as “increased activity” and EGFR 19del is actually increased (again if there were any standard deviation reported these values would likely be considered the same). These claims are completely erroneous and highly misrepresented. It is even unclear how the EGFR T790M showed ~650-fold decrease in EGFR activation – decreased compared to what? The only thing that gives a 650-fold decrease is when it is being compared to the other mutations, not wild-type EGFR which the start of the sentence states.

- We apologize that the standard deviations were missed in the kinase inhibition data. A table with IC_{50} (95%CI) was included in the revised manuscript (Table S3, see below).

Thank you for this addition

- The overstated description of the binding data of HX103 has been corrected as “**By contrast to EGFR wild-type, HX103 showed similar but slightly increased inhibition against EGFR L858R and EGFR 19del with IC_{50} values of 1.5 nM (1.3-1.9 nM; 95% CI, n=3) and 1.3 nM (1.2-1.5 nM; 95% CI, n=3), respectively (Figure 2E and Table S3).**”

Again, have you statistically tested this statement? If they are not statistically different you cannot comment on any increased inhibition. This difference seems too small to make any conclusion about it.

- For the “**~650-fold decrease**”, we intended to compare the IC_{50} of HX103 against EGFR T790M versus EGFR L858R. The unclear description was corrected as “**As expected, the probe’s inhibition towards the resistance mutation T790M was remarkably decreased with an IC_{50} of 977 nM (832-1147 nM; 95% CI, n=3), ~650-fold and ~750-fold decrease when compared to the primary mutants L858R and 19del, respectively (Table S3).**”

Thank you for the clarification.

Table S3. Inhibitory values for HX103 and gefitinib towards recombinant EGFR wild-type, EGFR L858R, EGFR 19del and EGFR T790M. Data represent average values with 95%

Confidence Intervals (CI), n=3 per group.

	HX103	Gefitinib
	IC ₅₀ (95% CI) [nM]	IC ₅₀ (95% CI) [nM]
EGFR wild-type	4.0 (3.6-4.4)	1.9 (1.7-2.2)
EGFR L858R	1.5 (1.3-1.9)	1.7 (1.5-1.9)
EGFR 19del	1.3 (1.2-1.5)	1.2 (1.0-1.4)
EGFR T790M	977 (832-1147)	787 (694-893)

b. The cell lines used have their p-EGFR/EGFR ratio determined through Western Blot and treatment with HX103. However, A431 and HeLa cells in Figure S3, which are stated to be wild-type EGFR (although this is never proven) show significant response to HX103 after stimulation with EGF. The difference in these cell lines is that the control (EGF and HX103-) do not show high baseline activation in the absence of EGF. This is important because in Figure 3 A431 demonstrates some of the highest HX103 staining but does not have any mutant EGFR.

- First of all, in the revised manuscript, we demonstrated that A431 and HeLa cells were wild-type EGFR, and low levels of EGFR primary mutants were identified with mutant-specific antibodies (Figure 3F, Figure S13-14).
- Thank you for this addition, but please describe the antibodies that you used as this is lacking from the revised manuscript.
- Next, we demonstrated that HX103 staining was correlated with EGFR phosphorylation, but not mutant EGFR. As EGFR-activating mutations reside near the ATP cleft, resulting in the constitutive activation of EGFR in the absence of the cognate ligand. And A431 is previously reported as a carcinoma cell line with an unusually high number of EGFR inducing the activation of EGFR-pathway. With this regard, EGFR autophosphorylation (as well as HX103 labeling) (~60%, Figure 3G, Figure S10 and S12) can be observed in A431.
- Again, you have no substantiative claim on correlation without performing a correlation test, otherwise you are commenting on qualitative trends. You go as far to say in the text that you observe “perfect correlation between HX103 labeling and EGFR activation” (lines 318-319). Observational or quantitative? Please validate.
- There appears to be several FACS experiments performed but they are not well described. How are you staining for p-EGFR when there is no description of permeabilization in your methods. Your supplementary figures have no descriptive text or call out in the materials and methods.

- Furthermore, in Figure S3 (Western blot analysis), the cells were all cultured in serum-starved condition without FBS for 24 h, followed by treatment with the probe and stimulation with EGF. As serum starvation of the cells for 24 h is a common procedure to synchronize all cells to the same cell cycle phase, thereby removing the impact of cell cycle on cell response to the treatment. However, in Figure 3, we aimed to determine the uptake of HX103 in a natural cellular system, thus the cells were cultured physiologically, but not serum-starved. In the revised manuscript, we studied the effects of serum on EGFR activation in these cells (Figure 3H, see below). Interestingly, we found EGFR autophosphorylation was significantly reduced in A431 cells in the absence of serum, whereas the addition of serum had no effects on EGFR autophosphorylation in HeLa cells. This indicates the level of EGFR autophosphorylation in A431 cells is higher than that in HeLa cells.

Interesting result.

Figure 3. (H) Bar chart showing percentage of EGFR(+)pEGFR(+) cells in the presence and absence of EGF or 10% FBS (Figure S15-16).

c. A431 stains the highest with the EGFR antibody, second highest (tied for second?) in HX103 staining, and shows a response to gefitinib. These results are completely ignored in the human tissue scenario but appears to play a really important role in its interpretation.

We appreciate this comment by the reviewer. The labeling difference of EGFR antibody and HX103 in A431 cells was applied to interpret the results in human tissue experiments. For example, the surgical sample #2-T [without *EGFR*-activating mutations (Figure S20)], showed ~98.9% of EGFR(+), ~58% of HX103(+) and 62.2% of EGFR(+)HX103(+), and also showed response to gefitinib (Figure S19). This case can be explained by A431 that EGFR autophosphorylation was also found in EGFR wild-type cells.

Thank you for this clarification.

d. What is the percent of total EGFR that is mutant in the cells lines and human

tissues? Typically, not all of the EGFR is mutant on a cell but a combination of the wild type and mutant. If the study is looking at activating mutations, why is staining with the antibodies for the mutations not shown? This is very important piece of this study and it is missing as far as I can tell. It is used for IHC but not for flow.

- In the revised manuscript, we determined the percent of total EGFR that is mutant in the cell lines with mutant-specific antibodies (Figure 3F, Figure S14). In H1975, we found ~70% of total EGFR was mutant to EGFR L858R. In HCC827 and PC-9, ~97% and ~85% of total EGFR were mutant to EGFR 19del, respectively. In EGFR wild-type (A431, A549 and HeLa) and EGFR-negative (MCF-7 and Jurkat) cells, no EGFR-activating mutants were identified.

Thank you for the addition.

- For human tissues, there are several technical difficulties for us to determine the percent of EGFR mutant with FACS analysis. Firstly, if we use mutant-specific antibodies (EGFR L858R and 19del) for HX103-based FACS, at least 7 single-cell suspension samples will be required for per human tissue (HX103, antibodies of L858R, 19del and CD45, live/dead cell discriminator kit, isotype control for L858R and 19del, as well as blank). However, in practice, the obtained single-cell suspensions from the human tissues were limited, and we can not ensure all the collected tissue samples (biopsy samples in particular) will be sufficient for obtaining 7 single-cell suspension samples. Furthermore, both antibodies are generated from rabbits. If we use these antibodies in the same experiment, it brings challenge for us to discriminate one from the other; and if we use the two antibodies separately, the required single-cell suspension samples and washing steps will be doubled (as too many washing steps will reduce the signal of probe labeling).
- Alternatively, we thus performed IHC experiments to determine the expression of EGFR mutants in human tissues (Figure S18). Having quantified the labeling of EGFR mutant antibodies in IHC, we found ~56% of EGFR L858R was observed in a human tumor tissue carrying *EGFR* L858R mutation, and ~50% of EGFR 19del mutant was observed in a human tumor tissue carrying *EGFR* 19del. Furthermore, the levels of EGFR phosphorylation were also determined by IHC with p-EGFR antibody (Figure 4G).

Thank you for this clarification.

e. Why was the K_d of HX103 not determined in A431 or HeLa cells, which do not have EGFR mutants.

In the revised manuscript, the K_d values for the rest cells, including A431 and HeLa, were determined. And the results revealed that the highest K_d values of $2.1 \pm 0.4 \mu\text{M}$ and $2.9 \pm 0.5 \mu\text{M}$ were found in HCC827 and H1975, respectively,

followed by PC-9, A431, HeLa and A549 (Figure S17).

Thank you.

4. Figure 2A - Autofluorescence, which in this case is the sum between the individual signals of HX103 and EGFR is a substantial portion of the total HX103 signal.

EGFR indeed shows some autofluorescence (~20%) in Figure 2A. However, after the autofluorescence correction, HX103 still showed >70% of fluorescence intensity. Importantly, in cell lines and human tissues, we included a blank control (sample without probe) for each experiment, to remove the background signals, including autofluorescence. Based on these, we do not think the autofluorescence of EGFR will affect our results of HX103 labeling.

Thank you.

5. Figure 2B & 2D – Y-axis says normalized fluorescence intensity, but no explanation of normalized procedure. Also, in 2D there is $\pm 20\%$ error on the fluorescent signal. Why is the error so much, especially compared to that in 2B, which is quite low?

- We appreciate this comment. The normalized procedure for Figure 2B and 2D was included in the revised manuscript (in the part of Measurement of optical properties, Supporting information). Specifically, the fluorescence intensity of HX103 in the presence of EGFR wild-type was set at 100% to normalized that of the remaining samples.
- The high $\pm 20\%$ error in Figure 2D might be due to the reduced activity of recombinant EGFR kinase, as the recombinant EGFR tyrosine kinases can be degraded if it is handled improperly. Therefore, we repeated this experiment with a new stock of EGFR kinases. As shown in Figure 2D, the standard deviation was reduced to some extent, and similar to that in Figure 2B.

Thank you

HX103 in human tissue sample (comments listed in order of importance):

6. Figure 5D & 6B – How were the thresholds selected from the receiver operating characteristic (ROC) curves? Was it the Optimal Cut-Off Point?

By using the software of GraphPad, we generated the ROC curves, which offered a series of sensitivity and specificity values with different thresholds. We then selected the optimal cut-off point with the best sensitivity and specificity.

Thank you.

a. What decision led to selecting different thresholds for determining for the surgery samples (30.13%) and the biopsies (19.93%)? Wouldn't it make sense to either pool the data together and make a single cut off point (more statistically accurate since you would have a higher number of samples) or just use one cut off chosen for each.

Previously, we considered that the single-cell suspensions obtained from biopsy samples was different from that of surgical samples, thus a different threshold for biopsy samples was used.

In the revised manuscript, to be logical, the first cut-off point from surgical samples (30.1%) was applied as the single cut-off point to predict *EGFR*-activating mutations and *EGFR*-TKI sensitivity (Figure 6).

Thank you.

b. Did you compare ROC curves for HX103(+), *EGFR*(+), and HX103(+) + *EGFR*(+)? Because *EGFR*(+) actually has better predictive value if you use a threshold cut off of 40% - this value can be used for both of the human surgery samples and biopsies. This conclusion completely changes the impact of the entire study.

Table 1. Diagnostic accuracy of surgical tissue samples comparing *EGFR*(+) to HX103(+) *EGFR* (+) using threshold presented in manuscript (left, white cells) and threshold for biopsy samples (right, yellow cells).

	EGFR (+)	HX103(+) EGFR (+)	HX103(+) EGFR (+)
True Positive	11	10	11
True Negative	9	9	8
False Positive	2	2	3
False Negative	1	2	1
Cut-off value	> 40%	> 30.13%	> 19.93

Table 2. Diagnostic accuracy of biopsy samples comparing *EGFR*(+) to HX103(+) *EGFR* (+) using threshold presented in manuscript (left, white cells) and threshold for biopsy samples (right, yellow cells).

	EGFR (+)	HX103(+) EGFR (+)	HX103(+) EGFR (+)
True Positive	6	5	6
True Negative	7	7	7
False Positive	0	0	0
False Negative	0	1	0
Cut-off value	> 40%	> 30.13%	> 19.93

In the revised manuscript, we compared the ROC curves for all three labeling parameters from HX103-based FACS [HX103(+), *EGFR*(+), and HX103(+) *EGFR*(+)] (Figure 5, Figure S22-24, S29). Indeed, *EGFR*(+) showed better correlations with *EGFR*-activating mutations in surgical samples. However, our aim is to predict *EGFR*-TKI sensitivity rather than *EGFR*-activating mutations. Although *EGFR*-activating mutations were found to be linked to an increased sensitivity to

EGFR-TKIs, there are still ~20-30% of patients carrying *EGFR*-activating mutations showed stable or progress disease on EGFR-TKIs underlining the imperfect correlation. In this regard, we analyzed the CT imaging data (pre-TKI versus post-TKI) to examine the therapeutic response to EGFR-TKIs in *EGFR*-mutant NSCLC patients. In the revised manuscript, we found that 69.2% (9/13) of patients with *EGFR*-activating mutations demonstrated objective responses (CR or PR) to EGFR-TKIs, while 30.8% (4/13) of patients carrying *EGFR*-activating mutations showed SD or experienced PD after receiving EGFR-TKIs. (Table 2-3 and Table S8). Therefore, our clinical results also supported that *EGFR* gene mutation status was not equal to EGFR-TKI response.

Furthermore, having compared the labeling parameters of HX103-based FACS, we found the labeling of EGFR(+)HX103(+) and HX103(+) showed better correlation with response to EGFR-TKI than that of EGFR(+) (Figure 6F, Figure S30B, see below). For example, Patient 3 carrying *EGFR* L858R mutation who showed stable disease after EGFR-TKI, had a high labeling of EGFR(+) (64.5% > the cut-off point of 36.3%), but low in EGFR(+)HX103(+) (~26.2% < the cut-off point of 30.1%)(Figure 6G, see below). This case suggests that EGFR(+)HX103(+) has better predictive value for the effectiveness of EGFR-TKI therapy than EGFR(+).

Overall, regarding therapy response to EGFR-TKI, the conclusion that EGFR-TKI-based probe may have high potential in prediction of EGFR-TKI sensitivity has not been changed.

Figure 6F. Box plot of the percentage of EGFR(+)HX103(+), EGFR(+) or HX103(+) with responsiveness to EGFR-TKI. CR or PR cases were categorized as responders, and SD or PD cases, as non-responders. Data are average values \pm SEM. Statistics was performed using Mann-Whitney test (**P < 0.01).

Figure 6G. CT images for Patient 3.

c. Again, as for the in vitro cells, it is unclear why the mutants aren't being quantified using flow cytometry and the antibodies against the mutant EGFR.

As we explained before, there are several technical difficulties to quantify the EGFR mutant with FACS analysis (see the answer for Question 3d). The main issue is that a large amount of single-cell suspensions are required for human tissues when mutant-specific antibodies are applied in FACS. However, in practice, the collected human tissues are not big enough to obtain at least 7 single-cell suspensions. Considering our final aim is to study the correlation between probe labeling and therapy response to EGFR-TKI, we thus determined the maximum response from baseline in target lesion size to EGFR-TKI treatment and studied the correlation between therapy response to EGFR-TKI and our FACS analysis.

d. The selection of only 6 patients to receive TKI therapy (unstated what therapy they received) does not prove the results of the team. All 13 people should have received TKI therapy and results presented. There was no negative control showing the proposed stratification actually worked.

Because our clinical study was a non-interventional study, and the treatment of the patients should fall within the current practice. According to the guidelines for the clinical use of targeted drugs in lung cancer patients, only the patients carrying *EGFR*-activating mutations were assigned to EGFR-TKI therapy. Therefore, the 15 patients (the number of patients was increased up to 31 in the revised manuscript) carrying *EGFR*-activating mutations were selected for EGFR-TKI (the specific therapy for each patient was shown in Table 2). The rest 16 patients showing *EGFR* gene mutation negative were assigned to other treatments according to the guidelines (Table S8).

Thank you for this clarification.

MINOR COMMENTS:

All corrections are suitable.

7. EGFR T790M is never introduced, as L858R and 19del mutations are.

The following sentence was added in the revised manuscript to briefly introduce EGFR T790M mutation. "Furthermore, we applied HX103 to the "gatekeeper" point mutation T790M, the most common resistance mutation^{49,50},"

8. Significant figures – one significant figure in your standard deviation and match in average value for example $K_d = 2.73 \pm 0.39 \mu\text{M}$ should really be $2.7 \pm$

0.4

We appreciate the comment and the significant figures were all corrected as the reviewer suggested in the revised manuscript.

9. The abbreviation of EGFR-TKIs doesn't always work in context of a pluralized situation.

The use of EGFR-TKIs or EGFR-TKI was corrected in the revised manuscript.

10. What is the level of EGFR mutation expression in the cell lines? You compare antibody binding of WT-EGFR to HX103 but not the mutations – you have antibodies for the mutated EGFR so this should be done.

The expression of EGFR mutants in cell lines with mutant-specific antibodies was determined (Figure S13-14). Among them, HCC827 (~97%) showed high level of EGFR 19del expression, followed by PC-9 (~85%), and the remaining cell lines showed minimal expression of EGFR 19del. In addition, H1975 was found with the highest expression of EGFR L858R (~70%) among these cells.

11. Your definition of the percent of double positive cells is confusing. You should have one cell population (Q1, Q2, Q3) and see how many are EGFR+, how many are HX103+ and how many are double. It is unclear why the ratio of cell populations is based on different cell populations.

In principle, one cell population (Q1, Q2 or Q3) should be observed in tumor cells, just like the case in culture cancer cell lines (e.g., H1975, HCC827, PC-9, etc.). However, lung cancer is a highly heterogeneous disease, and different levels of heterogeneity have been recognized in cancer, including interpatient, intratumor and intertumor. Thus, in some of our cases (e.g., #20-T, #22-T, #23-T and biopsy samples No.2-6), we observed several different cell populations in one sample. This can be explained by tumor heterogeneity that EGFR expression and kinase activity may be different in different cell populations.

Furthermore, to make clear about the percent of double positive cells (Q2), EGFR+ cells (Q1+Q2) and HX103+ cells (Q2+Q3), we included a table for per sample showing the calculations of the ratios for each sample (Figure S23-24 for surgical sample, Figure S31-34 for biopsy sample).

12. Table 1 - The furthest right column is double stained HX103 (+) EGFR (+), but in the scatter plots (Fig. 5D and 6B) is called HX103 labeling (%), which seems like it should correspond to the HX103(+) not HX103 (+) EGFR (+).

We apologize for the unclear description. We corrected "HX103 labeling (%)" by "the percentage value of HX103(+)" in our revised manuscript.

13. For clinical patients, should have the patient number and the mutation shown in a table so we can compare, as with the initial 23 patient surgical study (as presented in Table 1).

In the revised manuscript, we included a table (Table 2), which has the patient number, the mutation, and the percentage values of EGFR(+)HX103(+), HX103(+) and EGFR(+).

14. In general, there are many grammatical errors.

We have checked the manuscript carefully, and tried our best to avoid the grammatical errors.

For Reviewer #4

The clinically available DNA-based testing for EGFR mutation is clinically useful, with high sensitivity and specificity. Due to a variety of oncogenic driver gene alterations in NSCLC, we clinically need a multiplex testing with high performance, short turn-around-time, as well as low cost.

The new testing in the manuscript is only targeting major EGFR mutation, and cross-reacting WT EGFR, not as accurate as previous DNA-based technique, not along with clinical unmet needs. I think this manuscript is not suitable for Nature Communications.

We agree with the review that DNA-based testing for *EGFR* mutation is clinically useful. However, as for therapy response to EGFR-TKI, there are approximately 70%-80% (depending on the trials) of *EGFR* mutation-positive NSCLC patients who respond to EGFR-TKI treatment, and ~20%-30% of patients carrying *EGFR*-activating mutations do not show objective response when treated with EGFR-TKIs.

Furthermore, in our revised manuscript, 15 patients were identified with *EGFR*-activating mutation positive (L858R or 19del) among 31 enrolled patients, and 69.2% (9/13) of patients with *EGFR*-activating mutations demonstrated objective responses (CR or PR) to EGFR-TKIs, while 30.8% (4/13) of patients carrying *EGFR*-activating mutations showed SD or experienced PD after receiving EGFR-TKIs (CT scans of two patients after EGFR-TKIs were not obtained). These results further demonstrate that *EGFR*-activating mutation status does not always correlate with response to EGFR-TKI. These observations indicate that the correlation between *EGFR* mutation status and TKI therapy response is imperfect, and to predict EGFR-TKI response, *EGFR* gene mutations may not be the sole determinants.

Despite there are a variety of oncogenic driver gene alterations in NSCLC, L858R and 19del are the most common mutations, accounting for ~85% among all EGFR mutations. Furthermore, although with such a large number of oncogenic driver gene

alterations in NSCLC, the pharmacophore of the current EGFR-TKIs is similar (4-anilinoquinazoline; although the third-generation EGFR-TKIs has a different pharmacophore, but they only target the acquired resistance mutation EGFR T790M). Importantly, in clinical, the aim to determine *EGFR* mutation status is to predict EGFR-TKI sensitivity, and to study the correlation with EGFR-TKI therapy response will be better than only with *EGFR* gene mutations.

Since our probe HX103 is based on the pharmacophore of EGFR-TKI, it has the inherent advantage to “on-target” predict EGFR-TKI sensitivity by quantifying the amount of functional EGFR in tumors. According to our results, although the imperfect correlation was found with *EGFR* mutations status, it showed high correlation with response to EGFR-TKI (Figure 6, see below), highlighting the ability of EGFR-TKI-based probe in predicting the sensitivity of EGFR-TKI. Based on this, we think HX103-based assay will have potential in predicting the sensitivity and efficacy of EGFR-TKI in clinical trials.

Figure 6C and E. (C) Analysis of the prediction for *EGFR*-activating mutations by the percentage of EGFR(+)/HX103(+) in 31 biopsy samples from NSCLC patients. (E) Comparison of responsiveness to EGFR-TKI therapy with the percentage of EGFR(+)/HX103(+) in 15 NSCLC patients carrying *EGFR*-activating mutations. Of note, therapy responses of 2 patients (9 and 12) were evaluated.

Furthermore, comparing with currently DNA-based methods (e.g., NGS), HX103-based FACS approach takes much less time (~6 hour, Table R1 see below). Importantly, having calculated the total cost of HX103-based FACS analysis (~14\$ per biopsy sample, Table S9 in supporting information), we found the cost of HX103-based FACS approach is much less expensive than gene analysis. For example, NGS usually costs at least ~300\$ per biopsy sample (although it differs from country to country). The cost/benefit analysis for HX103-based FACS was also provided in the revised manuscript. Besides, although the probe cross-reacts with EGFR wild-type (this is mainly due to the pharmacophore of EGFR-TKIs), it would not be an issue in predicting the effectiveness of EGFR-TKIs in clinical. Because EGFR-TKI drugs also cross-react with EGFR wild-type (For example, gefitinib showed inhibition towards EGFR wild-type with an IC_{50} of 1.9 nM, shown Figure S4).

Taken together, with regarding the cost, turn-around time, and the correlation with therapy response to EGFR-TKI, we think EGFR-TKI-based probe would have high value in predicting the effectiveness of EGFR-TKI therapy in clinical. Particularly, when in the combination with DNA-based technique, it would have high potential to

reduce the false-positive results of gene analysis, thereby improving patients' outcomes.

Table R1. Time required for HX103-based FACS approach.

Procedures	Time (hour)	Total time for per sample
Single-cell isolation	3	~6 hour
HX103 incubation	0.5	
Antibody cocktail	0.5	
Blocking	0.5	
Washing	0.5	
FACS	0.5	
Others	0.5	

Reviewer #4:

Remarks to the Author:

In the clinical point of view, clinician needs to diagnose multiple oncogenic drivers, not only EGFR major mutations, but also EGFR minor mutations (e.g exon 20 ins, de novo T790M) to deliver precise targeted therapy for patients with advanced/recurrent NSCLC. The diagnostic tool for only targeting major EGFR mutation (ex19 deletion and L858R) is not sufficient in clinical practice. As the authors mention in the manuscript, the presence of EGFR mutation does not perfectly predict response to EGFR-TKI, however, most of these are reportedly explained by co-existing mutation in other genes (TP53 mutation, RTK gene amplification, cell cycle gene alterations, etc).

Furthermore, the standard of care for EGFR mutation-positive NSCLC is now 3rd generation-EGFR-TKI, Osimertinib, which can target T790M with sparing effect against WT-EGFR.

The authors claim the clinical utility of fluorogenic probe-diagnosis to predict sensitivity to EGFR-TKI treatment, when in combined with traditional DNA-based diagnosis. However, the diagnostic accuracy of HX103 for EGFR mutation was moderate (< 90%), which was quite similar with WT-EGFR probe. In addition, the cohort evaluated in this study contains very limited number of patients with NSCLC expressing WT-EGFR, which may also mislead the diagnostic yields of EGFR probe as predictive marker of EGFR-TKI. As the authors mention that HX103 cross-react with WT-EGFR, the significance of predictive biomarker of EGFR-TKI should be evaluated in consecutive cohort containing patients with EGFR overexpression/amplification, independent of DNA-based EGFR mutation diagnosis.

Even if HX103 will be used alone or in combination with DNA-based diagnosis, it is very hard to handle this technique in the clinical practice, because HX103-based analysis requires fresh sample. We need to obtain fresh sample for HX103-based analysis for all patients suspicious of advanced/recurrent NSCLC, in addition to sample for pathological diagnosis, otherwise, we need re-biopsy for HX103-based analysis to determine the indication of EGFR-TKI.

I don't think this technique is more useful, feasible, and cost-effective compared with traditional companion diagnostics.

Collectively, I still think this manuscript is not suitable for the publication in Nature Communications.

Point-by-point reply for manuscript “A fluorogenic probe for predicting treatment response in non-small cell lung cancer with *EGFR*-activating mutations”

Please find below a point-by-point reply to the reviewers' comments (blue text). We again appreciate the time and patience invested by the reviewers in evaluating our manuscript. The comments are very useful and have improved our manuscript.

For Reviewer #3

Individual response to comments are listed below in red. In general, the response to the review seemed thorough although there is a significant lack of validation of any claimed „correlation” using a defined correlation test. All correlations appear to be qualitative observations. This needs to be remedied prior to publication and language adjusted accordingly.

We sincerely thank the reviewer for carefully evaluating our manuscript. Pearson's correlation analysis has been performed to validate the correlation, and other modifications have also been made according to the comments.

HX103 in docking studies:

1. Line 152-154 – It is unclear what is meant here. “The docking studies indicated that HX103 might be a fluorescent ligand towards EGFR with high-affinity. No strong interactions were detected in EGFR wild-type and double mutant.” These two sentences seem contradictory.

a. Is the first sentence referring only to L858R?

b. What strong interactions were not detected in wild-type and double mutant EGFR?

c. How much difference was measured between the binding of gefitinib and HX103?

Usually this is quantifiable by docking studies.

(a) We apologize for the unclear statement. The sentence was corrected by “**Overall, the docking studies suggest that the fluorescent probe HX103 may show strong binding affinity towards EGFR L858R.**”

Thank you for clarifying. However, there are still some remaining questions. What is the error being reported here? Are these binding energies statistically different? They have overlapping errors (standard deviations?), which suggests they are not different, and you cannot make any statement about one being “slightly higher”.

We apologize for the lack of the error description in Supplementary Table 1, the error being reported here is the standard error of mean (SEM).

We agree with the review that the statement of “HX103 may show strong binding affinity towards EGFR L858R” is not accurate by being “slightly higher” binding energy, we corrected the statement by following description:

“Furthermore, the binding energies of the fluorescent probe HX103 and gefitinib to EGFR wild-type and mutants were calculated (Supplementary Table 1). These results provide the theoretical basis for the binding affinity of HX103 toward EGFR wild-type and the mutants.”

- (b) “No strong interactions were detected in EGFR wild-type and double mutant” has been corrected by **“Excepting for Met793, no strong interactions were detected in EGFR wild-type and L858R/T790M mutant (Lys 745 and Phe795 interactions were absent, and the interaction of Pro794 in double-mutant was weak), where different orientations of the SBD group were observed”**
The details of this study are very vague, what consists of a strong interaction and what is a weak interaction? There are no definitions or quantitations.

The details of the interactions between gefitinib/HX103 and EGFRs were added in the revised Supplementary Fig.1a and b, where the ionic bonding, H-bond and pi interactions were specified. The description has also been corrected as follows:

“As for EGFR wild-type and L858R/T790M double-mutant, the orientations of the SBD group of HX103 were different from that of EGFR L858R. As shown in Fig. 1c and Supplementary Fig. 1, only one H-bond interaction was found between HX103 and Met 793 in EGFR wild-type, while gefitinib formed another ionic bonding with Asp800 in EGFR wild-type. Despite an additional H-bond was found between HX103 and Pro794 in EGFR double-mutant, the interaction was weaker when comparing with the interaction between gefitinib and Asp800 in double-mutant (the ionic bonding and H-bond interaction were both observed, Supplementary Fig. 1).”

- (c) The binding energy difference of gefitinib and HX103 was quantified in the revised manuscript (Table S1, see below).

Thank you for the addition of this table. What are the values and errors reported? Mean \pm standard deviation? You should only have one significant figure in the error and report mean to that number. For instance, -71.6 ± 4.3 should be written -72 ± 4 .

The values and errors reported in Supplementary Table 1 are Mean \pm standard error of mean (SEM). The significant figures in the table were corrected in the revised manuscript (see below).

Supplementary Table 1. Binding energy of HX103 and gefitinib to EGFR wild-type and mutants (L858R, L858R/T790M). Data represent average values with the standard error of mean (SEM).

Target	Ligand	Vander Waal energy (kJ/mol)	Electrostatic energy (kJ/mol)	Polar solvation energy (kJ/mol)	SASA energy (kJ/mol)	Binding energy (kJ/mol)
EGFR L858R	Gefitinib	-106±4	-14±3	58±2	-10±0	-72±4
	HX103	-104±4	-9±4	57±2	-10±1	-67±3
EGFR wild-type	Gefitinib	-98±3	-15±3	60±2	-13±1	-65±5
	HX103	-102±4	-12±2	63±3	-9±1	-60±4
EGFR L858R/T790M	Gefitinib	-84±3	-16±2	59±2	-12±1	-53±3
	HX103	-89±3	-14±3	61±3	-13±0	-55±4

HX103 in solvent and in vitro studies (comments listed in order of importance):

2. Figure 2C – The fluorescence resulting from HX103 binding to wild-type EGFR and L858R, and 19del mutants are equivalent. It is stated that HX103 is sensitive to EGFR activating mutations but really it is also sensitive to wild-type EGFR. In cells, without confirming pathology, how would you be able to tell where the signal is coming from.

- In cellular experiments, we performed i) experiments with EGFR mutant-specific antibodies and p-EGFR antibody to evaluate the levels of mutant EGFRs and phosphorylated EGFR in these cell lines (Figure 3); ii) Western blot analysis with p-EGFR and other phosphorylated proteins in the signaling pathways to evaluate the effects of HX103 on EGFR activation and downstream signaling (Figure 2F-K, Figure S3); and iii) competition experiments with gefitinib to confirm the specific signal from HX103 (Figure S7, Figure 3A-B). Taken together, these studies provide the evidences for the specific labeling of HX103 in the cells.

The antibodies used for mutant specific staining are not described anywhere that I can see. The additional data significantly helps.

We apologize for the lack of the specific information about the antibodies used for the additional experiments. The experimental details and antibody information were added in the Supplementary Methods (the part of “Fluorescence-activated cell sorting (FACS) assay in cultured cells”, see below).

Fluorescence-activated cell sorting (FACS) assay in cultured cells

Quantification of EGFR expression and auto-phosphorylation with specific antibodies. For FACS assay with EGFR antibody, around 106 live-cells were collected and washed with PBS (2x). The cells were fixed by 400 µL 4% paraformaldehyde for 15 min at room temperature, followed by permeabilization with 0.1% tritonX-100 for 10 min and blocking with 5% BSA for 30 min.

Subsequently, the cells were centrifuged (2000 g, 3 min) and re-suspended in 100 μ L dilution of a **specific EGFR antibody (L858R mutant, #3197, CST; E746-A750del, #2085, CST; phospho-EGFR, Tyr1068, #3777, CST)** in 3% BSA for 30 min at room temperature. After that, the cells were washed with 0.5% BSA/PBS (3x), followed by staining with 100 μ L of **secondary antibody (anti-rabbit Alexa Fluor 448 conjugate, cat #A11008, Invitrogen)** for 30 min at room temperature in the dark. The cells were then washed with 0.5% BSA/PBS (3x) and resuspended in 400 μ L of 0.1% BSA/PBS. Notably, **for total-EGFR antibody, the Alexa Fluor 647 conjugated antibody (#1929882, abcam)** dilution in 3% BSA was used for one-step staining for 30 min. The resuspended cell suspension was then analyzed on a FACSVerseTM flow cytometer (BD Biosciences, LSRFortessa). Nonspecific fluorescence binding was omitted by isotype control (#3900, CST) and the corresponding mean fluorescence was subtracted from the signal measured with each primary antibody. 10000 events per tube were analyzed and the mean fluorescence intensity was detected and analyzed.

3. Figure 2E-L, 3 & S3, F – HX103 fluorescence is correlated to EGFR expression, not mutant expression.

HX103 fluorescence correlated neither to mutant EGFR expression nor EGFR expression. As HX103 is designed based on EGFR-TKI, it is speculated that HX103 fluorescence would correlate to EGFR activation levels. In the revised manuscript, we performed additional experiments, and the results indeed demonstrated that HX103 fluorescence correlated to EGFR autophosphorylation (including the autophosphorylation of mutant EGFR and wild-type EGFR) (Figure 3G, Figure S12-13). The correlation of HX103 labeling with EGFR autophosphorylation provides important basis for the application of HX103 in predicting EGFR-TKI sensitivity in human tissues.

You have performed no tests for correlation. Your conclusion that “HX103 correlates” to activation is a qualification of the results. You need to perform a Pearson or Spearman correlation, or another relevant correlation test, to substantiate these claims.

We appreciate the comment by the reviewer. Pearson’s correlation analysis has been performed to substantiate the correlation between HX103 labeling and EGFR activation (Supplementary Fig. 13, see below).

Supplementary Fig. 13. Pearson's correlation analysis between HX103 labeling and EGFR expression\ EGFR activation in a panel of cancer cells (H1975, HCC827, A549, HeLa, PC-9, A431, MCF-7 and Jurkat). **a** Correlation of the percentage values between HX103(+) and EGFR(+). **b** Correlation of the percentage values between HX103(+) and pEGFR(+). **c** Correlation of the percentage values between EGFR(+)HX103(+) and EGFR(+)pEGFR(+). **d** Correlation of the percentage values between pEGFR(+) and EGFR(+). Data represent average values \pm SEM, $n = 3$ per group. The statistical P values were calculated by the two-tailed Student's test, and r represents Pearson's correlation coefficient.

a. The differences in IC_{50} values for HX103 and gefitinib binding wild type and mutated EGFR is overstated. The IC_{50} values of wild-type EGFR for HX103 and gefitinib are 4.0 nM and 1.9 nM (no standard deviations reported), respectively, and is describe as being similar to gefitinib. Comparatively, the IC_{50} values of HX103 and gefitinib with of L858R and 19del mutant EGFR are 1.5 nM and 1.7 nM (L858R) and 1.3 and 1.2 nM(19del), respectively. These results are summarized as demonstrating “increased activity against EGFR L858R and EGFR 19del” (line 215). Without even considering the lack of standard deviation reported, it is doubtful that we can call the nominal change in IC_{50} as “increased activity” and EGFR 19del is actually increased (again if there were any standard deviation reported these values would likely be considered the same). These claims are completely erroneous and highly misrepresented. It is even unclear how the EGFR T790M showed ~650-fold decrease in EGFR activation – decreased compared to what? The only thing that gives a 650-fold decrease is when it is being compared to the other mutations, not wild-type EGFR which the start of the sentence states.

- The overstated description of the binding data of HX103 has been corrected as “By contrast to EGFR wild-type, HX103 showed similar but slightly increased

inhibition against EGFR L858R and EGFR 19del with IC_{50} values of 1.5 nM (1.3-1.9 nM; 95% CI, n=3) and 1.3 nM (1.2-1.5 nM; 95% CI, n=3), respectively (Figure 2E and Table S3).”

Again, have you statistically tested this statement? If they are not statistically different you cannot comment on any increased inhibition. This difference seems too small to make any conclusion about it.

We have corrected the original statement by the following sentence:

“As for EGFR L858R and EGFR 19del, HX103 inhibited the kinase activities with IC_{50} values of 1.5 nM (1.3-1.9 nM; 95% CI, n=3) and 1.3 nM (1.2-1.5 nM; 95% CI, n=3), respectively (Fig. 2e and Supplementary Table 3).”

b. The cell lines used have their p-EGFR/EGFR ratio determined through Western Blot and treatment with HX103. However, A431 and HeLa cells in Figure S3, which are stated to be wild-type EGFR (although this is never proven) show significant response to HX103 after stimulation with EGF. The difference in these cell lines is that the control (EGF and HX103-) do not show high baseline activation in the absence of EGF. This is important because in Figure 3 A431 demonstrates some of the highest HX103 staining but does not have any mutant EGFR.

- First of all, in the revised manuscript, we demonstrated that A431 and HeLa cells were wild-type EGFR, and low levels of EGFR primary mutants were identified with mutant-specific antibodies (Figure 3F, Figure S13-14).

Thank you for this addition, but please describe the antibodies that you used as this is lacking from the revised manuscript.

The antibody information and experimental details were added in the revised supplementary information (see the part of “Fluorescence-activated cell sorting (FACS) assay in cultured cells - *Quantification of EGFR expression and auto-phosphorylation with specific antibodies*”).

- Next, we demonstrated that HX103 staining was correlated with EGFR phosphorylation, but not mutant EGFR. As EGFR-activating mutations reside near the ATP cleft, resulting in the constitutive activation of EGFR in the absence of the cognate ligand. And A431 is previously reported as a carcinoma cell line with an unusually high number of EGFR inducing the activation of EGFR-pathway. With this regard, EGFR autophosphorylation (as well as HX103 labeling) (~60%, Figure 3G, Figure S10 and S12) can be observed in A431.

Again, you have no substantiative claim on correlation without performing a correlation test, otherwise you are commenting on qualitative trends. You go as far to say in the text that you observe “perfect correlation between HX103 labeling and EGFR activation” (lines 318-319). Observational or quantitative? Please

validate.

Thank you for the suggestion, and Pearson's correlation analysis has been performed in the revised manuscript. The above text has been modified as follows,

“The results reveal that the levels of EGFR activation show a similar pattern as the accumulation of HX103. Furthermore, we found the percentage of EGFR antibody and HX103 double-positive cells [EGFR(+)HX103(+)] showed similar percentage values as that of EGFR(+)pEGFR(+) (Fig. 3g, Supplementary Fig. 9-12), again suggesting that HX103 labeling may correlate with EGFR activation. We subsequently substantiated this by using Pearson's correlation analysis [HX103(+) v.s. pEGFR(+), $r = 0.9835$; $P < 0.0001$; EGFR(+)HX103(+) v.s. EGFR(+)pEGFR(+), $r = 0.9928$; $P < 0.0001$] (Supplementary Fig. 13).”

There appears to be several FACS experiments performed but they are not well described. How are you staining for p-EGFR when there is no description of permeabilization in your methods. Your supplementary figures have no descriptive text or call out in the materials and methods.

The experimental details of the additional FACS experiments were provided in the revised supplementary methods (the part of “Fluorescence-activated cell sorting (FACS) assay in cultured cells”, see below).

Co-staining with EGFR antibody and HX103. Briefly, the cells were collected and resuspended in 0.1% BSA/PBS buffer, followed by the addition of HX103 as described above. Subsequently, the treated cells were incubated for 30 min at 37 °C, followed by washing with PBS buffer for three times. Next, the HX103-treated cells were fixed (4% paraformaldehyde, 15 min), permeabilized (0.1% tritonX-100, 10 min) and blocked (5% BSA, 30 min) at room temperature, followed by staining with total-EGFR antibody conjugated by Alexa Fluor 647. Finally, the co-stained cells were washed with 0.5% BSA/PBS for three times and re-suspended in 400 μ L of 0.1% BSA/PBS. The obtained cell suspension was ready for analysis.

Co-staining with different EGFR antibodies. Firstly, the collected cells were fixed (4% paraformaldehyde, 15 min), permeabilized (0.1% tritonX-100, 10 min) and blocked (5% BSA, 30 min), followed by staining with an EGFR mutant-specific antibody (L858R mutant/E746-A750del) or the pEGFR antibody as described above. Next, the stained cells were washed with 0.5% BSA/PBS (3x), followed by the addition of 100 μ L total-EGFR antibody conjugated by Alexa Fluor 647. After incubation for another 30 min at room temperature, the co-stained cells were washed with 0.5% BSA/PBS for three times and re-suspended in 400 μ L of 0.1% BSA/PBS. The obtained cell suspension was ready for analysis.

Cell starvation and stimulation. The cells were seeded on the petri-dish and cultured at the normal condition with 10% FBS, and that was changed to serum-free condition for 24 hours when the cells proliferated at ~80% confluence.

Subsequently, the starved cells were treated with EGF (25 ng/mL) for 10 min at 37 °C, followed by washing with PBS for three times. Next, the cells were co-stained with different EGFR antibodies as described above for the subsequent analysis.

Reviewer #4 (Remarks to the Author):

In the clinical point of view, clinician needs to diagnose multiple oncogenic drivers, not only EGFR major mutations, but also EGFR minor mutations (e.g exon 20 ins, de novo T790M) to deliver precise targeted therapy for patients with advanced/recurrent NSCLC. The diagnostic tool for only targeting major EGFR mutation (ex19 deletion and L858R) is not sufficient in clinical practice. As the authors mention in the manuscript, the presence of EGFR mutation does not perfectly predict response to EGFR-TKI, however, most of these are reportedly explained by co-existing mutation in other genes (TP53 mutation, RTK gene amplification, cell cycle gene alterations, etc). Furthermore, the standard of care for EGFR mutation-positive NSCLC is now 3rd generation-EGFR-TKI, Osimertinib, which can target T790M with sparing effect against WT-EGFR. The authors claim the clinical utility of fluorogenic probe-diagnosis to predict sensitivity to EGFR-TKI treatment, when in combined with traditional DNA-based diagnosis. However, the diagnostic accuracy of HX103 for EGFR mutation was moderate (< 90%), which was quite similar with WT-EGFR probe. In addition, the cohort evaluated in this study contains very limited number of patients with NSCLC expressing WT-EGFR, which may also mislead the diagnostic yields of EGFR probe as predictive marker of EGFR-TKI. As the authors mention that HX103 cross-react with WT-EGFR, the significance of predictive biomarker of EGFR-TKI should be evaluated in consecutive cohort containing patients with EGFR overexpression/amplification, independent of DNA-based EGFR mutation diagnosis. Even if HX103 will be used alone or in combination with DNA-based diagnosis, it is very hard to handle this technique in the clinical practice, because HX103-based analysis requires fresh sample. We need to obtain fresh sample for HX103-based analysis for all patients suspicious of advanced/recurrent NSCLC, in addition to sample for pathological diagnosis, otherwise, we need re-biopsy for HX103-based analysis to determine the indication of EGFR-TKI. I don't think this technique is more useful, feasible, and cost-effective compared with traditional companion diagnostics. Collectively, I still think this manuscript is not suitable for the publication in Nature Communications.

Thank you for the time and patience invested by the reviewer in evaluating our manuscript.

We agree with the review that methods detecting multiple oncogenic drivers is quite useful for patients with NSCLC. As we mentioned before, such methods only provide information on genetic changes, which do not always translate to the protein level. However, the actual target of EGFR-TKI is the active site of EGFR kinase domain. Importantly, EGFR-TKI only targets EGFR kinase with functional activity, thereby exerting their therapeutic effect. Thus, screening with genetic methods only may

produce false-positive cases that would not benefit from EGFR-TKI treatment. In this situation, to improve clinical trial design for NSCLC patients, alternative methods are necessary, particularly structure-function-based methods enabling the determination of EGFR activation. Therefore, based on the pharmacophore of EGFR-TKI, we developed a chemical probe, which mimics the binding of EGFR-TKI and detects EGFR activity.

Indeed, as the reviewer mentioned, the imperfect correlation between EGFR mutation and EGFR-TKI therapy response was reportedly explained by co-existing mutation in other genes (e.g., TP53 mutation, RTK gene amplification etc.). However, this is only the partial reason for EGFR mutation cases that do not respond to EGFR-TKI, but cannot explain all cases. For example, in our study, patient 11 with only EGFR L858R mutation identified but no other co-existing mutation, also showed SD after receiving EGFR-TKI. Thus, we think other mechanisms may also play a role to regulate the kinase activity of EGFR, which determines the efficacy of EGFR-TKI binding. Besides, we agree with the review that the diagnostic accuracy of our assay for EGFR mutation was moderate (<90%), and this is mainly due to four patients (3, 8, 11 and 15) with EGFR mutations but low probe labeling. Considering these patients did not respond to EGFR-TKI treatment as well, we speculated that although the patients carrying EGFR mutations, the level of kinase activity of the mutant EGFR could be low and EGFR-TKIs may show minimal binding on EGFR mutant without kinase activity.

Besides, the receptor overexpression is considered as another mechanism of EGFR activation in cancer cells (e.g., A431), however, the overexpression/amplification of EGFR is not exclusively accompanied by EGFR activation. Therefore, to design effective adjuvant therapies in patients with EGFR overexpression/amplification, it requires the evaluation of not only protein overexpression but also its functional status. Considering our probe cross-reacts with EGFR wild-type, HX103 may also have the potential to predict the sensitivity of EGFR-TKI in patients with EGFR overexpression/amplification. In the revised manuscript, the utility of our probe in patients with EGFR overexpression/amplification has been analyzed in the Discussion section (See below the text).

“Furthermore, it is noteworthy that amplified wild-type *EGFR* was also associated with EGFR-TKI responses in some of patients (~11%)^{55, 61}, and HX103-based FACS measures EGFR activation not only in EGFR mutant cells but also wild-type EGFR cells. As such, this method may also have utility in patients with EGFR overexpression/amplification to select individuals who may respond to EGFR-TKI. Further studies are therefore required to evaluate the predictive value of this method and to determine whether it can predict EGFR-TKI sensitivity in patients with EGFR overexpression/amplification, independent of DNA-based *EGFR* mutation diagnosis.”

In addition, for the sample in HX103-based FACS assay, it is not necessary to be fresh, and the frozen tissue sample (e.g., stored in liquid nitrogen) is also possible according to our experience, as well as literature reports [ref. DOI 10.1186/s13048-017-0337-0]. Furthermore, we have to emphasize that our aim is to show whether a

function-activity-based approach could be a biomarker in prediction of EGFR-TKI sensitivity, thereby giving a clue to develop alternative tools or methods to precisely personalize EGFR-TKI therapy and improve clinical trial design. In future, we believe that the more diagnostic tools or methods are developed (exon-based, activity-based, etc.), the better clinical trial choices can be made. Projecting forward, new clinical feasible, easy-used techniques that based on EGFR kinase activity are still need to be developed.

Reviewers' Comments:

Reviewer #4:

Remarks to the Author:

I agree with the possibility of your probe (HX103) to select candidate patients for EGFR-TKI more precisely, in combination with traditional EGFR mutational analysis or alone in the future.

It would be useful that this technique can be utilized with stored samples.

Besides that, I also checked again the relationship of response to TKI with HX103-positivity, two patients who were not evaluable (NE) for efficacy of TKI, but positive for HX103 were excluded from evaluation of HX103 prediction value. These NE patients are usually classified as not responding population (PD/NE) in clinical trials. I recommend to add the reason why these two patients were not evaluable (No measurable target or no follow up etc). Current evaluation using the cohort (15 patients, excluding 2 NE patients) is too small, and misleading to claim the utility of this probe for predicting EGFR-TKI. Therefore, I suggest you evaluate it with larger numbers of retrospective cohort or another prospective validation cohort.

I still think the current version of manuscript is not enough for publication in Nature Communications.

Reviewer #5:

Remarks to the Author:

It is an interesting collaborative study involving EGFR-TKI sensitivity prediction, which could have significant implications for personalized and precision medicine.

As part of the revision process, the authors and review3 have been interactive with each other. Reviewer 3's questions were answered fairly by the authors, and the suggested edits have been made. As a result, some of their claims are much more clearly supported. It was surprising, however, that the authors did not clearly describe how the SBD activates when it binds to the EGFR. Please describe the changes in the SBD environment. Furthermore, despite several revisions, there is no proper description and reference to environment-sensitive fluorophores in the introduction and results.

Point-by-point reply for manuscript “A fluorogenic probe for predicting treatment response in non-small cell lung cancer with *EGFR*-activating mutations”

Please find below a point-by-point reply to the reviewers' comments (blue text). We again appreciate the time and patience invested by the reviewers in evaluating our manuscript. The comments are very useful and have improved our manuscript.

Reviewer #4:

I agree with the possibility of your probe (HX103) to select candidate patients for EGFR-TKI more precisely, in combination with traditional EGFR mutational analysis or alone in the future.

It would be useful that this technique can be utilized with stored samples.

Besides that, I also checked again the relationship of response to TKI with HX103-positivity, two patients who were not evaluable (NE) for efficacy of TKI, but positive for HX103 were excluded from evaluation of HX103 prediction value. These NE patients are usually classified as not responding population (PD/NE) in clinical trials. I recommend to add the reason why these two patients were not evaluable (No measurable target or no follow up etc). Current evaluation using the cohort (15 patients, excluding 2 NE patients) is too small, and misleading to claim the utility of this probe for predicting EGFR-TKI. Therefore, I suggest you evaluate it with larger numbers of retrospective cohort or another prospective validation cohort.

I still think the current version of manuscript is not enough for publication in Nature Communications.

Thank you for the time and patience invested by the reviewer in evaluating our manuscript once again.

The following sentences were added in the Methods section (part of Evaluation of EGFR-TKI efficacy) of the revised manuscript to explain why these two patients were not evaluable (NE):

“Of note, two patients (9 and 12) did not come to us after EGFR-TKI due to personal reasons. Therefore, the CT scans of the two patients were missing and these two cases were categorized as not evaluable (NE).”

We agree with the review that in some cases the NE patients are classified as not responding population (PD/NE) in clinical trials. However, it really depends on the specific scenarios, for example, the NE cases containing non-measurable target or the patients pass away should be classified as PD. In our study, we cannot obtain the CT scans of two cases (Patient 9 and 12) after EGFR-TKIs, which mainly due to the personal reasons of the patients (patient 9 moved to another city due to the job transfer; patient 12 moved to the place where his daughter is living), but not any health reasons.

In general, missing data in clinical trials could be classified into 3 scenarios i) missing completely at random (because of the personal reasons, the patient cannot be followed up), ii) missing at random (the effect of previous treatment is unsatisfied, causing the drop-out) and iii) missing not at random. The approach to handle the missing data varies due to different circumstances. In our study, CT scans of the two cases after EGFR-TKIs are missing because of their personal reasons. According to literature reports [1-2], this kind of missing data can be classified as missing completely at random. To obtain an unbiased analysis, the missing data belong to missing completely at random are often not included during the analysis, and we thus excluded the two cases in our study of the relationship between TKI response and the probe labeling.

References:

1. Roderick J. Little et al. The Prevention and Treatment of Missing Data in Clinical Trials, *N Engl J Med*, 367(14):1355-60, (2012).
2. Melanie L Bell et al. Handling missing data in RCTs; a review of the top medical journals, *BMC Med Res Methodol*, 14:118, (2014).

Reviewer #5:

It is an interesting collaborative study involving EGFR-TKI sensitivity prediction, which could have significant implications for personalized and precision medicine.

As part of the revision process, the authors and review3 have been interactive with each other. Reviewer 3's questions were answered fairly by the authors, and the suggested edits have been made. As a result, some of their claims are much more clearly supported. It was surprising, however, that the authors did not clearly describe how the SBD activates when it binds to the EGFR. Please describe the changes in the SBD environment. Furthermore, despite several revisions, there is no proper description and reference to environment-sensitive fluorophores in the introduction and results.

We sincerely thank the reviewer for carefully evaluating our manuscript, and appreciate the comment by the reviewer. The description of how the SBD activates when it binds to EGFR has been added in the revised manuscript. Furthermore, the description and relevant references to environment-sensitive fluorophores was also added in the introduction part. (See below the added paragraphs).

“Among them, environment-sensitive fluorophores such as 4-sulfonamidebenzoxadiazole (SBD) have unique emission properties that are highly sensitive to the immediate environment. Generally, very weak fluorescence is observed with these fluorophores in polar and protic environment, whereas fluorescence turn-on can be activated when they are in hydrophobic surroundings. As most of the binding sites in proteins are hydrophobic, fluorescent turn-on probes thus

can be achieved by incorporating an environment-sensitive fluorophore with a protein-specific ligand (e.g., small-molecule inhibitors like EGFR-TKIs).”

“It is expected that the binding of EGFR-TKI to the hydrophobic domain of EGFR would bring the environment-sensitive fluorophore closer to the hydrophobic pocket, thereby causing the fluorophore to emit stronger fluorescence. By contrast, in the absence of target protein, the EGFR-TKI-based probe would remain in the aqueous buffer and show very weak fluorescence.”